# GPTNT💥: Benchmarking Real-Time Collaboration Between Multimodal Agents on *Keep Talking And Nobody Explodes*

## Abstract

Multimodal models are increasingly deployed to solve tasks collaboratively with humans or other artificial agents. While existing benchmarks show that they possess the fundamental capabilities, the various conditions that coincide when collaborating—time pressure, information asymmetry, and imperfect communication—have traditionally been studied in isolation.

To address this gap, we introduce GPTNT, a benchmark built on the cooperative video game *Keep Talking and Nobody Explodes*, in which two agents must coordinate to defuse procedurally generated bomb puzzles against a live countdown. One agent has access to the bomb but not the instructions for defusing it; the other holds the instructions but cannot see or manipulate the bomb. Neither agent can succeed alone: the task requires contributions from both, and is solvable only through effective, efficient communication. We remove turn-taking proxies or simplifications, instead requiring agents to act asynchronously and communicate in real time.

GPTNT is designed to separate collaboration from relying on memorised solutions: the instruction manual, the partner, or both, can optionally be withheld to isolate what a model derives in the moment from what it already knows. We demonstrate that GPTNT poses a considerable challenge to the state-of-the-art: not one of the closed- and open-source models we test defuses a single bomb in real time, a bar that human players clear. In a range of controlled experiments, we explore where capabilities break down, identifying critical weaknesses in state tracking, efficient acting within the time budget, handling ambiguity, and error recovery.

We release GPTNT as a means for testing the collaborative performance that current benchmarks leave unmeasured. Since it runs on the real game, GPTNT benefits from procedural generation and inherits a living modding community: as models improve, the benchmark can be evolved to remain challenging, rather than being solved once and retired.

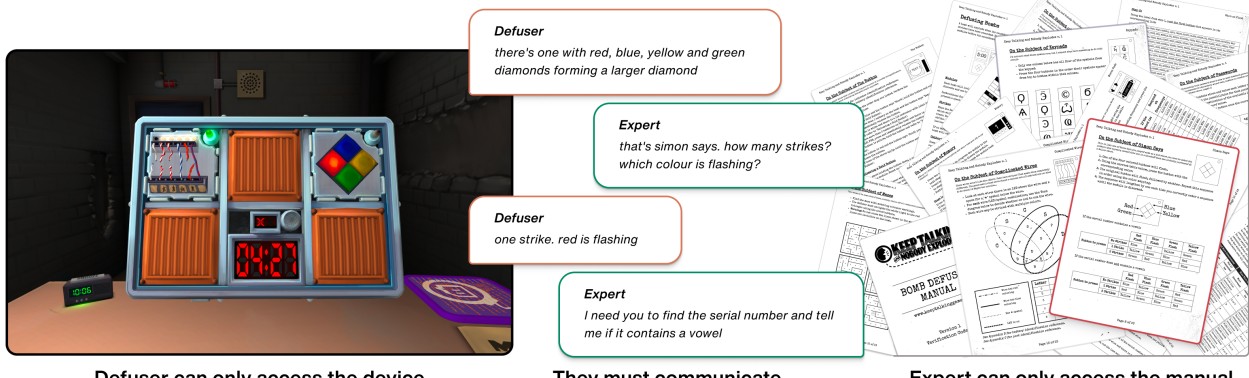

**Defuser can only access the device**   **They must communicate**   **Expert can only access the manual**

Figure 1: High-level overview of the interaction in *Keep Talking and Nobody Explodes*. Two agents must coordinate under asymmetric information and real-time pressure to solve a series of puzzles: the Defuser can see the bomb but has no manual, while the Expert has the manual but cannot see the bomb.

# 1   Introduction

Multimodal large language models (MLLMs) can read documents (Ma et al., 2024b; Zhu et al., 2026), interpret complex visual scenes (Li et al., 2024a; 2025b; Wang et al., 2025d), and reason over long contexts (Kuratov et al., 2024; Wang et al., 2024a; 2025d)—and they are being deployed to do so collaboratively, alongside humans and other agents (Zou et al., 2025). However, the conditions that define real-time collaboration—time pressure on decision-making and action execution, information asymmetry between partners, and the ensuing need to communicate effectively and efficiently across a visually complex and dynamic environment (Gergle et al., 2004; Kraut et al., 2003)—are not typically tested together.

Generalist agentic benchmarks evaluate models across a broad range of interactive tasks in single-agent, static environments where communication is not required (Liu et al., 2024a;b; Ma et al., 2024a). Game-based environments offer richer visual complexity and clearer success criteria but remain fundamentally single-agent (Paglieri et al., 2024; Zhang et al., 2025a); cooperative settings exist but either remove the information asymmetry that makes communication necessary (Gong et al., 2024), bypass visual grounding (Li et al., 2024b; Shridhar et al., 2021), or operate in environments that only change when agents act (Padmakumar et al., 2022; Puig et al., 2020; Shridhar et al., 2021). No existing benchmark approximates real-world conditions by simultaneously demanding cooperative coordination under asymmetric information, real-time asynchronous action, visual grounding in a dynamic environment, and extended multi-turn communication.

The video game *Keep Talking and Nobody Explodes* (KTANE; Steel Crate Games, 2015)[1] provides an ideal environment for testing these capabilities in an integrated fashion. As illustrated by Figure 1, two players with strictly asymmetric information must collaborate in real time to defuse a procedurally generated bomb composed of a number of puzzle modules—one player can see the bomb but not the instruction manual, the other the manual but not the bomb. The rules in the instruction manual for defusing the bomb are arbitrary; therefore, possessing commonsense or prior knowledge about defusing real-world bombs or relevant domains such as engineering does not translate into an advantage for playing KTANE. Fundamentally, neither player can succeed alone, and communication is the only mechanism through which the task can be solved.

We introduce GPTNT, a framework and corresponding benchmark that builds on top of KTANE to provide a rigorous evaluation environment for multimodal models. Across the five state-of-the-art model families we test, **no model successfully defuses a single entry-level bomb in real time**—a bar that nine out of ten human player pairs can clear. Through systematic ablations that reduce time and task complexity, we reveal that models struggle to complete even the simplest setting: a bomb with a single puzzle module, and unlimited deliberation and generation time. We investigate how models perform zero-shot coordination in *self-play* and *other-play* (Hu et al., 2020; Stone et al., 2010) and find that models dedicate a larger portion of the game to communication when collaborating with a different model.

Targeted offline evaluations and qualitative analyses confirm several well-documented low-level weaknesses of current models: models hallucinate visual and textual information (Asadi et al., 2026), fail to identify when information provided by the other player is implausible (Sharma et al., 2023) or ambiguous (Wang et al., 2025c), lose track of state across turns (Laban et al., 2026), and rarely recover from mistakes (Zhang et al., 2024). Beyond this, the benchmark aims to distinguish communicative competence from prior exposure to the game during training (Magar & Schwartz, 2022). To this end, we release evaluation components that serve both as diagnostic tools *and* contamination checks, allowing us to systematically surface individual agents' reliance on memorisation shortcuts.

GPTNT is built to remain challenging as models improve. **New missions can be procedurally generated and flexibly configured** by adjusting the time limit or varying the count, type, or placement of modules. In addition, new module types released by KTANE's modding community can be introduced without rebuilding the framework—presenting a countermeasure to saturation and keeping the task space ahead of any fixed training distribution (ARC Prize Foundation, 2026).

---

[1] https://keeptalkinggame.com

Our core contributions are as follows:

- **A benchmark that unifies the conditions for real-time collaboration.** GPTNT is, to our knowledge, the first benchmark to simultaneously require coordination under asymmetric information, real-time asynchronous action, visual grounding in a dynamic environment, and sustained multi-turn communication.

- **Comprehensive evaluation of frontier models.** In addition to evaluating how frontier models perform on the GPTNT benchmark against a baseline of ten human player pairs (§5), we systematically reduce complexity to localise where model performance breaks down through simplified missions (§6), and conduct in-depth analyses of how models collaborate (§7), and how they fail at a higher level (§8) and a lower level (§9).

- **Diagnostic tools that separate collaboration from memorisation.** We devise dedicated single-agent evaluations to isolate the effect of collaboration, and remove access to the instruction manual to expose contamination (§10).

## 2 Related Work

Table 1: Comparison with prior interactive MLLM benchmarks that require both vision and natural language.

| | Coordination | | | Environment | | Context | | |
| --- | --- | --- | --- | --- | --- | --- | --- | --- |
| | Multi-Agent[*] | Information Asymmetry | Asynchronous Action | Dynamic | Real-Time | Multi-Turn Communication | Image Sequence | Long Context |
| VisualWebArena (2024b) | ✗ | ✗ | ✗ | ✗ | ✗ | ✗ | ✓ | ✗ |
| OSWorld (2024b) | ✗ | ✗ | ✗ | ✗ | ✗ | ✗ | ✓ | ✓ |
| MineDojo (2022) | ✗ | ✗ | ✗ | ✓ | ✗ | ✗ | ✓ | ✗ |
| HAZARD (2024) | ✗ | ✗ | ✗ | ✓ | ✗ | ✗ | ✗ | ✗ |
| BALROG (2024) | ✗ | ✗ | ✗ | ✓ | ✗ | ✗ | ✓ | ✓ |
| VideoGameBench (2025a) | ✗ | ✗ | ✗ | ✓ | ✓ | ✗ | ✓ | ✓ |
| WatchAndHelp (2018; 2020) | Co-op | ✗ | ✓ | ✗ | ✗ | ✗ | ✓ | ✗ |
| TEACh (2022) | Co-op | ✓ | ✗ | ✗ | ✗ | ✓ | ✗ | ✗ |
| Alexa Arena (2023) | Co-op | ✓ | ✗ | ✗ | ✗ | ✗ | ✓ | ✗ |
| WhodunitBench (2024a) | Comp | ✓ | ✗ | ✗ | ✗ | ✓ | ✗ | ✓ |
| Multimodal CLEM (2025) | Co-op | ✓ | ✗ | ✗ | ✗ | ✓ | ✓ | ✗ |
| COMMA (2025) | Co-op | ✓ | ✗ | ✗ | ✗ | ✓ | ✗ | ✗ |
| VS-Bench (2026) | Mixed | ✓ | ✓ | ✓ | ✗ | ✗ | ✗ | ✗ |
| **GPTNT** | Co-op | ✓ | ✓ | ✓ | ✓ | ✓ | ✓ | ✓ |

[*] Includes co-operative (Co-op), competitive (Comp), and mixed-motive (Mixed) games. We provide definitions for each column in Appendix A.

Benchmarks for evaluating MLLM agents range from static perception and reasoning tasks (Chen et al., 2024; Lu et al., 2024; Yue et al., 2025) to increasingly interactive settings that additionally demand planning, memory, and coordination. Table 1 compares GPTNT with existing interactive benchmarks that involve both vision and language, organised by the dimensions that distinguish them. We define each dimension and provide an extended literature review in Appendix A.

Interactive GUI benchmarks such as OSWorld (Xie et al., 2024b) and VisualWebArena (Koh et al., 2024b) evaluate visually-grounded decision-making for computer tasks, and progress in GUI agents has translated to video game interfaces (OpenAI, 2025a; Wang et al., 2025a;e). In these non-dynamic environments, the state *only* changes when the agent acts. Certain game-based (e.g., BALROG, Paglieri et al., 2024; MineDojo, Fan et al., 2022) and embodied benchmarks (e.g., HAZARD, Zhou et al., 2024) introduce dynamic environments that change independently of the agent. VideoGameBench (Zhang et al., 2025a) extends this further to real-time evaluation, which imposes latency constraints. All of these, however, test single agents.

Multi-agent settings with information asymmetry add a distinct challenge: when each agent holds knowledge the other lacks and neither can succeed alone, language becomes a necessary coordination channel (Gao et al.,

2023; 2022; Padmakumar et al., 2022; Xie et al., 2024a). Cooperative environments often instantiate this asymmetry as an instructor–follower setup, but typically agents are not required to integrate information over a sequence of images, and traces stay short enough to avoid the long context that extended collaboration produces (Laban et al., 2026). Benchmarks, such as COMMA (Ossowski et al., 2025) and Multimodal CLEM (Hakimov et al., 2025), focus on pragmatic communication in cooperative settings. COMMA takes direct inspiration from KTANE puzzles but removes the multi-module structure, time pressure, and three-dimensional, dynamic environment to isolate communication as the main object of study.

As Table 1 shows, no existing benchmark instantiates all of these properties at once within a single task, which limits how well they capture real collaborative interaction (Paolo et al., 2024). VS-Bench (Xu et al., 2026) and VideoGameBench (Zhang et al., 2025a) come closest in coverage, with VS-Bench being one of the few environments that support asynchronous coordination between agents—acting in parallel rather than in alternating turns. However, both distribute their properties across heterogeneous game collections, instead of a single task to test these properties simultaneously. GPTNT is the only benchmark where asynchronous multi-agent coordination under real-time pressure and extended multimodal context all hold at once—not because each is engineered in, but because KTANE itself demands them.

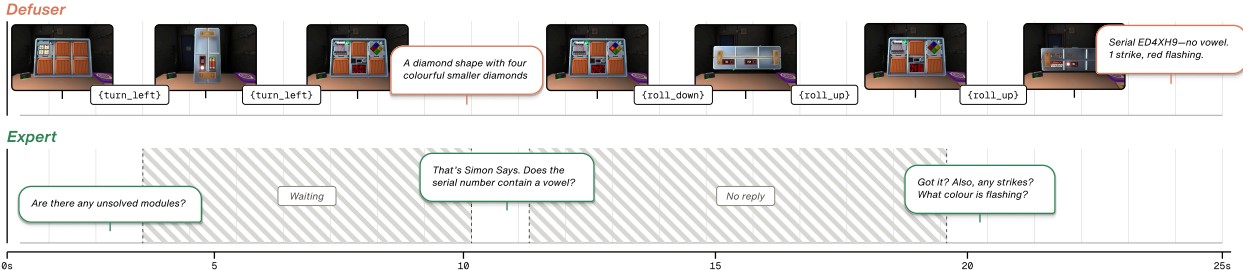

Figure 2: Rollout of two agents interacting in real time. The Defuser gathers and relays information about modules and widgets incrementally by rotating the three-dimensional bomb. The Expert—who cannot see the bomb—identifies the module type from the Defuser's description and requests additional details on which the solution for the module depends (Figure 4). Every action consumes game time, and agents do not share the same context: messages are the only channel between them.

## 3  The Game: What is *Keep Talking and Nobody Explodes?*

*Keep Talking and Nobody Explodes* (KTANE) is a popular cooperative bomb defusal video game where one player (the *Defuser*) can see the bomb but lacks the knowledge needed to defuse it, while the other (the *Expert*) has access to the instructions but cannot see the bomb. The two players must bridge this difference through communication to defuse a bomb by solving a number of puzzle modules before the countdown timer runs out, and without exceeding the allowed number of strikes. Crucially, neither player can directly apply any pre-existing knowledge about bombs, as the solutions to the puzzles in KTANE are specific to the game and are not applicable in any other setting. Additionally, there is a strict limit to how many random guesses can be attempted, as each failed attempt leads to a strike. Together, these factors make KTANE a challenging test-bed for studying multimodal agentic collaboration under time pressure.

**The Expert.**  Experts are limited to communicating with the other player or waiting to receive new information from them. Since the Expert cannot see the bomb, it must identify the correct instructions based exclusively on what the Defuser describes, making effective communication critical.

**The Defuser.**  Where the Expert can only communicate and wait, the Defuser can additionally explore and manipulate the bomb directly. However, their actions ultimately depend on the instructions they receive from the Expert. Due to the three-dimensional nature of the bomb,[2] the Defuser's view is partial—only one side of the bomb is visible at a time. To inspect any other side, the Defuser must rotate the bomb vertically or horizontally (see Figure 2).

---

[2]For an exploded view of an example bomb, see Figure B.1.

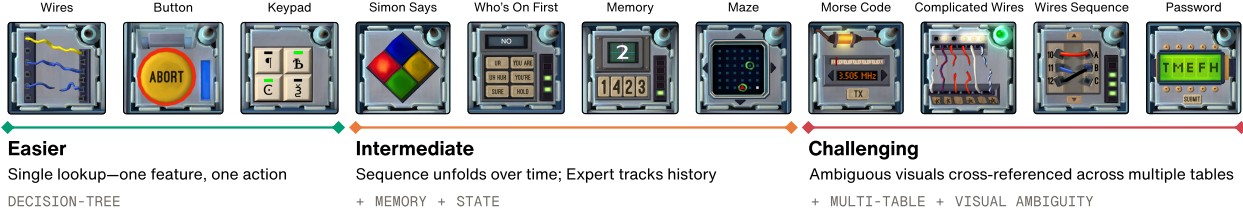

| Wires | Button | Keypad | Simon Says | Who's On First | Memory | Maze | Morse Code | Complicated Wires | Wires Sequence | Password |

**Easier**
Single lookup—one feature, one action
DECISION-TREE

**Intermediate**
Sequence unfolds over time; Expert tracks history
+ MEMORY + STATE

**Challenging**
Ambiguous visuals cross-referenced across multiple tables
+ MULTI-TABLE + VISUAL AMBIGUITY

Figure 3: Overview of the default puzzle module types in KTANE, grouped by their intended difficulty. Solving any module involves following conditional rules to a single correct action. Harder tiers increase difficulty by layering additional demands, introducing new perception, reasoning, and planning challenges.

**Modules.** At a minimum, a bomb will contain the countdown timer and one puzzle module. At full capacity, a bomb can hold up to eleven modules across its front and back. All modules on a given bomb are independent of one another and can be solved in any order. There are eleven default[3] module *types* in KTANE, ranging from easier to challenging (as illustrated in Figure 3), each of which poses a distinct challenge for the players. Individual instances of the same module type are sampled randomly and so, besides appearing visually distinct, they may require entirely different solutions. We provide a description of each module type in Appendix B.2.

**Widgets.** The four sides of a bomb—left, right, top, and bottom—can carry a number of widgets: non-interactive components that are sampled randomly, and can include a serial number, various types of batteries, ports, and indicators. The correct solution to several module types depends on these values, for instance, whether an indicator is lit, the number of batteries, the presence of a specific type of port, or whether the last digit of the serial number is odd (e.g., Figure 4).

**Missions.** In KTANE, a mission is a set of constraints used to procedurally generate similar bomb scenarios. These constraints include parameters such as the types and number of puzzle modules, the strike limit, and the time limit, and are used to achieve seven levels of increasing difficulty. Each of the seven difficulty levels is further broken down into sub-levels: for instance, level 2.1 introduces new modules, while level 2.4 tests this under a stricter time limit. By default, each bomb scenario instantiated from the same mission definition will vary. For the purpose of the benchmark, our mission definitions include a pre-defined seed. This guarantees identical bomb scenarios for every player pair.

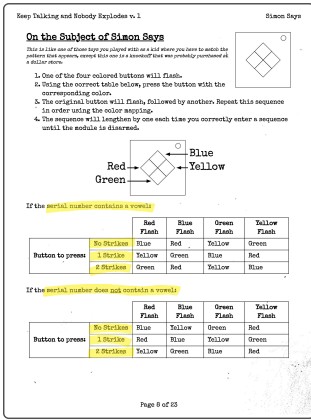

Figure 4: Manual page for ◈ *Simon Says*; widget references highlighted.

**Outcomes.** The bomb is defused, and a mission is considered successful when all puzzle modules on the bomb have been solved. Module-level progress can be tracked by observing an LED that lights up green when the module has been solved (see 🖼 *Complicated Wires*; Figure 3). In some cases, modules provide additional partial progress signals (e.g., 🔢 *Memory*; Figure 3). A failed mission ends in one of two terminal states: a *strikeout* occurs when the Defuser makes three mistakes, leading to three strikes, or a *timeout* when the timer expires before all modules are solved. The two states are diagnostically distinct: strikeout reflects acting on a wrong belief, timeout reflects failing to communicate and/or act efficiently enough.

## 4  The Benchmark: What is GPTNT?

GPTNT turns KTANE into a zero-shot benchmark for evaluating real-time collaboration between MLLMs. Two model instances—one playing as the Defuser, and one as the Expert—coordinate to complete procedurally generated missions from the game. The model instances never share the same context: every piece of information can only be communicated in natural language (§4.2). As KTANE provides no native API, visual interface hook, or tooling for programmatic control, we develop this from scratch. We release

---

[3]We exclude *Needy Modules* from our experiments—a separate class of modules that require repeated attention at random intervals and cannot be evaluated in isolation. They rely on audio cues to signal when they require attention, even while the Defuser is focused elsewhere. We leave this to future work.

the full infrastructure required to run the benchmark: the game mod and microservice framework, the fixed suite of mission configurations, the processed manual, and the role-specific system prompts.[4]

## 4.1 Handling Time

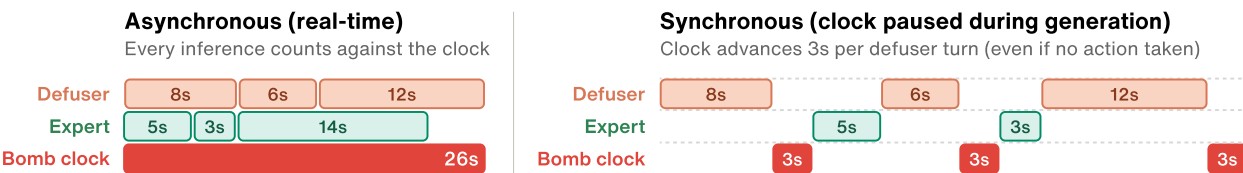

Figure 5: Example showing how the generation time for each request relates to the in-game time for asynchronous and synchronous gameplay settings. In asynchronous mode (left), generation time per request counts against the bomb clock, simulating constraints of real-time decision-making. In synchronous mode (right), the game is paused during generation, decoupling task difficulty from generation latency.

**Async**hronous mode.  In the default setting of the benchmark, models play in *asynchronous* real-time, mimicking conditions human players would encounter: the bomb clock advances in wall-clock time and does not pause while models are generating. This departs from the turn-based protocol commonly used in agentic benchmarks (Table 1), where the environment waits for each model to finish before the world state advances. We run the two perception-action loops in parallel; neither blocks the other, meaning the Expert can reason and speak while the Defuser acts (Figure 5, left). Latency from generating tokens becomes part of the task, rather than an artefact to be controlled for—every generated token consumes real game time, penalising models that think exhaustively and communicate verbosely.[5]

**Sync**hronous mode.  The benchmark additionally supports gameplay in a controlled *synchronous* setting, in which the clock is paused during generation and instead advances a fixed increment per Defuser turn (Figure 5, right). While this relaxes the real-time pressure of the asynchronous mode, models still need to complete missions within a fixed number of steps. This allows for separating the ability to reason accurately from the ability to reason quickly and succinctly, and lowers the barrier to entry by matching the turn-based interaction that current models typically assume. In practice, we set the fixed increment to 3 s, allowing the transition animation after an action to complete before retrieving the updated visual state (Appendix B.4).

## 4.2 Players

We instantiate an independent agent process per role, each with its own perception-action loop, either from the *same* model (self-play), or two *different* models (other-play). We provide full implementation details, including the complete action schema, image resolutions per model, and the manual format, in Appendix C.

**System prompt.**  The Defuser and Expert receive separate system prompts built around their distinct role-specific responsibilities, action and observation spaces. Additionally, models are made aware of the specifics of time progression through their system prompts: in the synchronous setting, each action is specified to consume a fixed amount of game time, whereas in the asynchronous setting, models are informed that the clock advances in real time, proportional to the generation duration. The full prompts are provided in Appendix H.

### 4.2.1 Observation space

On each turn, the Defuser receives the current game frame (a screenshot from the game) alongside the immediately preceding frame. Including the preceding frame is necessary as it contains the only reliable signal for inferring that an action produced a visible change in game state—such as receiving a strike, completing a

---

[4]We do not provide access to KTANE itself, as that goes against the license. We discuss this further in §12.

[5]We cap the maximum number of new tokens per response as the reasoning models are systematically biased towards producing long reasoning chains that add little and consume disproportionately more generation time (Chen et al., 2025; Sui et al., 2025).

sub-goal, or solving a module. By comparing the game time in the clocks between two frames, the model can also infer how much game time passes between actions. For ▤ *Morse Code* and ◈ *Simon Says*, where information is encoded in flashing sequences that unfold over time, the Defuser instead receives a buffer of 16 frames captured at 0.25 s intervals—four seconds of gameplay, sized to contain at least one complete cycle of the flashing pattern (the longest letter for ▤ *Morse Code*; see Appendix C.1.4). The Expert receives each page of the bomb defusal manual as interleaved text and images, delivered in its first incoming message and protected from context truncation regardless of game length. As the Expert cannot perceive the bomb, it entirely depends on the Defuser's descriptions to identify modules, request missing information, and reason over what to do. Messages are free-form text and are delivered atomically to the recipient on its next forward pass; no information is dropped, summarised, or withheld in transit.

**Set-of-Marks.**    While GPTNT supports coordinate-based interaction, models using direct coordinate output perform substantially worse (see Appendix G.10 for prelimi- nary results). We therefore adopt a *set-of-marks* (SoM) approach (Yang et al., 2023), following its use in recent agentic settings (Koh et al., 2024a; Liu et al., 2024b; Xie et al., 2024b; Zhang et al., 2025b), to avoid obscuring agent capabilities with sub-par low-level execution. We overlay segmentation-mask labels on each frame, using letters rather than numbers to avoid ambiguity with numeric module content. Labels are positioned consistently relative to the interactive layout, and their colours match element properties where colour carries meaning (Figure 6). The full implementation details can be found in Appendix C.4.2.

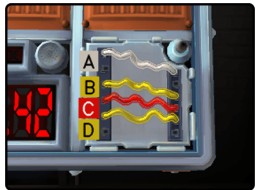

Figure 6: SoM on *Wires*

### 4.2.2   Action space

The Expert's action space is *strictly* limited to sending a message or doing nothing—its sole contribution is providing instructions and guidance to the Defuser using natural language. Beyond messaging and waiting, the Defuser can click, hold, or release interactive elements, zoom into or out of modules, as well as rotate or flip the bomb. For both players, doing nothing is a first-class action. KTANE contains modules where, in fact, waiting is the correct action for the Defuser—for instance, ▣ *Button* requires releasing on a specific timer digit, and ▤ *Morse Code* and ◈ *Simon Says* require observing flashing sequences across multiple turns. The Expert may choose to wait while the Defuser is exploring and manipulating the bomb.[6]

**ReAct-style framework.**    Following recent work on agentic models (Zhang et al., 2025a), all models are prompted using a ReAct formulation (Yao et al., 2022). Beyond the established benefits of chain-of-thought reasoning (Wei et al., 2022b), the reasoning trace can act as an implicit working memory, allowing the Defuser to accumulate a representation of the game state from partial observations without having to provide every prior image as part of its context. Specifically, models are prompted to output an explicit reasoning trace followed by a single action, delimited by XML tags, within a combined 1,000-token budget.[7] While the benchmark supports extended reasoning—as seen in large reasoning models (Sui et al., 2025; Xu et al., 2025)—we effectively disable it for all evaluated models. In preliminary experiments, models routinely consume over 4,000 reasoning tokens per turn before producing a response—equal to approximately one minute of wall-clock time. In addition, thinking budgets are typically implemented at the API level, with no equivalent for locally served open-source models, and disabling thinking ensures configurations are comparable across models (full rationale in Appendix C.7).

### 4.3   Evaluation Protocol

GPTNT is a zero-shot benchmark. The main evaluation comprises ten missions, configured at levels 2 and 3 out of the seven difficulty levels used in the game (full details in Appendix B.3). The missions differ with respect to the number, types, and placement of modules, as well as the total time limit. Unlike evaluating pass@$k$ on a single mission—where each additional attempt only provides evidence for how solvable that particular

---

[6]Waiting is also the result of the model causing any type of error, which we detail in Appendix C.7.2.
[7]We do not take advantage of structured decoding or tool calling, and provide our rationale in Appendix C.7.

configuration is—sampling several missions and attempting each once (pass@1) allows us to estimate overall model competence across the wider task space at a lower compute cost (Bouthillier et al., 2021).

Besides mission success, we report partial success as the number of modules solved in those missions, and the cause of failure, distinguishing between accumulating too many mistakes or running out of time. While asynchronous gameplay is the default evaluation mode, we additionally report results in the synchronous setting (in §5.2) to separate capability from response speed. Finally, for transparency, we report single-agent performance without manual access (in §10.2) to disentangle collaborative capability from parametric knowledge of the game.

## 5 Benchmarking Models on GPTNT

We evaluate five recent MLLMs regarded as strong vision-language models, selected based on their prevalence and performance on existing benchmarks (Chollet et al., 2025; Rein et al., 2024; Xie et al., 2025; Yue et al., 2025). These include ⑨ GPT-5.2 (OpenAI, 2025b), ✳ Claude Sonnet 4.6 (Anthropic, 2026b), ◆ Gemini 3 Flash (Google, 2025), 🐦 Qwen3.5 (27B; Qwen Team, 2026), and 🎋 InternVL 3.5 (38B; Wang et al., 2025b). The open-source models are selected at the largest size that can be accommodated within our available compute. While considerably smaller than the evaluated closed-source ones, open-source models serve as reproducible and accessible baselines for the academic community. Models without sufficient context length and native multi-image inputs cannot support the task and are consequently not considered—see Appendix E.1 for more details. Further details on model selection and configuration are provided in Appendix E.

To contextualise model performance, we additionally collect a human baseline: ten pairs of human players (20 participants, nine of whom have never played KTANE) play the same ten multi-module missions under real-time conditions. Since no pairing consists of two experienced players, we treat this as a soft upper bound on non-expert human performance. Full recruitment details, protocol, and results are in Appendix F. This baseline serves as a reference point rather than a controlled comparison with model performance, as participants communicate by voice instead of text.[8] Each setup has structural advantages: humans benefit from the incrementality inherent to spoken dialogue, while models suffer no working-memory constraints, can access the full manual and exact dialogue history, and benefit from the precision of text when communicating symbols from various scripts, sidestepping the homophone ambiguities that arise in speech.

### 5.1 Asynchronous Gameplay Results

We first evaluate each model in asynchronous self-play—where time progresses in real time and without turn-taking constraints, and where both roles are filled by an instance of the same model—on ten missions with multiple puzzle modules. This setting combines the most demanding conditions we test. Table 2 shows the game outcomes for these missions ordered by increasing difficulty.

**Models fail to solve a single mission.** No model fully completes a single mission in our full setting, though all models solve at least one module. The larger, closed-source models solve more individual modules than the smaller, open-source models, with Sonnet solving the most (5) and InternVL solving the fewest (1). Model performance falls far behind the human baselines: even the worst-performing human pair, while not completing any of the missions, solves more modules (18) than all models combined (15), whereas the best-performing human player pair solves twice as many modules (30), including six full missions. On average, humans manage to complete 2.8 out of 10 missions, and 20.8 out of 34 modules.

**Both models and human players run out of time.** The dominant failure mode for both models and humans is timeout. Apart from Sonnet, which has more strikeouts than timeouts, models time out on anywhere between 50% and 90% of missions (detailed in Appendix G.4). Humans fail by running out of time 74% of the time across all games and player pairs (detailed in Appendix F.4). However, interpreting timeouts

---

[8]Human players *can* succeed using only text messages, confirming the task is not speech-dependent. However, models produce typed text significantly faster than most humans can type. In this iteration of the benchmark, we refrain from using implementations that rely on speech-to-text to avoid introducing transcription errors and other confounds.

Table 2: Game outcomes and number of modules solved for self-play across ten distinct missions in asynchronous mode, ordered by increasing difficulty (Appendix B.3). In the asynchronous setting, we do not stop the game clock while models are generating. Each mission is attempted once (pass@1). We also show the best and worst human pairings as baselines.

| | Mission (ordered by difficulty) | | | | | | | | | | |
| | 1 | 2 | 3 | 4 | 5 | 6 | 7 | 8 | 9 | 10 | Overall |
|---|---|---|---|---|---|---|---|---|---|---|---|
| ✳ Sonnet 4.6 | ●∘∘ ✗ | ∘∘∘∘ ✗ | ●∘∘∘ ⏱ | ●∘∘ ⏱ | ∘∘∘ ✗ | ∘∘∘ ✗ | ●∘∘ ✗ | ●∘∘ ⏱ | ∘∘∘∘ ⏱ | ∘∘∘∘ ✗ | 0 |
| ✦ Gemini 3 Flash | ∘∘∘ ✗ | ∘∘∘∘ ⏱ | ∘∘∘∘ ✗ | ∘∘∘ ✗ | ∘∘∘ ⏱ | ●∘∘ ⏱ | ●∘∘ ✗ | ●∘∘ ⏱ | ∘∘∘∘ ⏱ | ∘∘∘∘ ✗ | 0 |
| ⬡ GPT-5.2 | ●∘∘ ⏱ | ∘∘∘∘ ⏱ | ●●∘∘ ⏱ | ●∘∘ ⏱ | ∘∘∘ ⏱ | ∘∘∘ ⏱ | ∘∘∘ ✗ | ∘∘∘ ⏱ | ∘∘∘∘ ⏱ | ∘∘∘∘ ⏱ | 0 |
| 〽 InternVL 3.5 (38B) | ∘∘∘ ✗ | ∘∘∘∘ ⏱ | ∘∘∘∘ ⏱ | ∘∘∘ ✗ | ∘∘∘ ⏱ | ∘∘∘ ⏱ | ∘∘∘ ✗ | ●∘∘ ⏱ | ∘∘∘∘ ⏱ | ∘∘∘∘ ⏱ | 0 |
| ⟁ Qwen3.5 (27B) | ∘∘∘ ⏱ | ∘∘∘∘ ⏱ | ∘∘∘∘ ⏱ | ●∘∘ ⏱ | ●∘∘ ⏱ | ∘∘∘ ✗ | ∘∘∘ ✗ | ∘∘∘ ⏱ | ∘∘∘∘ ⏱ | ∘∘∘∘ ⏱ | 0 |
| *Best Human Pairing* | ●●● ✓ | ●●●● ✓ | ●●●● ✓ | ●●∘ ⏱ | ●●● ✓ | ●●● ✓ | ●●∘ ⏱ | ●●● ✓ | ●●●∘ ⏱ | ●●●∘ ⏱ | 6 |
| *Worst Human Pairing* | ●●∘ ✗ | ●●∘∘ ⏱ | ●●●∘ ✗ | ●●∘ ⏱ | ●∘∘ ⏱ | ∘∘∘ ⏱ | ●●∘ ⏱ | ●∘∘ ✗ | ●∘∘∘ ✗ | ●●∘∘ ⏱ | 0 |

Icons represent mission outcome: ✓ solved, ✗ strikeout, ⏱ timeout.
Each bullet represents one module: ● solved, ∘ unsolved.

in isolation is problematic: they conflate task difficulty with model-specific factors such as generation latency and verbosity, which leaves little room to understand the cause of failure.

## 5.2 Synchronous Gameplay Results

In the asynchronous setting—where the game runs in real time—total generation time becomes both a bottleneck and a confounder: a model pair that would otherwise succeed may simply time out. For the purpose of this benchmark, we view total generation time as something shaped by model behaviour: a model that deliberates extensively or communicates verbosely will consume more of the time budget regardless of token throughput. Some modules—such as 🔲 *Button*—require that generated actions are *aligned* with the time shown on the game clock; a response that arrives too late or too early will fail regardless of its correctness, making it impossible to attribute failure to capability, deliberation cost, timing, or speed alone.

We address this by running the same set of experiments in a *synchronous mode*: between each Defuser turn, we freeze the game clock to give each model unlimited processing time.[9] In this case, models take turns: the Defuser acts, the game advances, the Expert acts, and so on. From this, we establish a capability ceiling: synchronous performance reflects an upper bound of what a model pair can achieve without putting time pressure on reasoning and communication. This helps to reveal collaborative abilities that may otherwise be hidden by end-to-end inference latency.[10]

**The synchronous setting is only marginally less challenging for models.** Table 3 shows that in this setting, which reduces *time complexity*, models continue to struggle to solve our realistically complex missions. Only Sonnet and GPT, the models that solve the most and second-most individual modules in the asynchronous setting, complete a mission. While all models solve more individual modules than in the asynchronous setting, they still fail without solving a single module on at best half (Sonnet, GPT), and at worst seven (Qwen) of the ten missions.

**Synchronous play shifts failures toward striking out.** Across all models, strikeouts increase from 34% in the asynchronous setting to half of total failures, matching the number of timeouts. This is especially evident for the second-best performing model (GPT), for which 67% of failures occur due to striking out,

---

[9] We set Unity's internal time scale to zero to freeze the game in place.
[10] We calibrate each step to a real-time equivalent at three seconds per action; more details in Appendix B.4.

Table 3: Game outcomes and number of modules solved for self-play across ten distinct missions in synchronous mode, ordered by increasing difficulty (Appendix B.3). In the synchronous setting, we stop the game clock while models take turns generating. Each mission is attempted once (pass@1).

| | Mission (ordered by difficulty) | | | | | | | | | | |
|---|---|---|---|---|---|---|---|---|---|---|---|
| | 1 | 2 | 3 | 4 | 5 | 6 | 7 | 8 | 9 | 10 | |
| FRONT: / BACK: | | | | | | | | | | | Overall |
| ✳ Sonnet 4.6 | ○○○ ✗ | ●●●● ✓ | ○○○○ ✗ | ●●○ ⏱ | ●○○ ⏱ | ●●○ ⏱ | ○○○ ✗ | ●○○ ✗ | ○○○○ ✗ | ○○○○ ✗ | 1 |
| ✦ Gemini 3 Flash | ●○○ ✗ | ○○○○ ⏱ | ●○○○ ⏱ | ○○○ ⏱ | ○○○ ⏱ | ○○○ ✗ | ●●○ ⏱ | ●○○ ⏱ | ○○○○ ✗ | ○○○○ ✗ | 0 |
| Ⓢ GPT-5.2 | ○○○ ⏱ | ●○○○ ✗ | ○○○○ ✗ | ●●● ✓ | ●○○ ⏱ | ●○○ ✗ | ○○○ ✗ | ●○○ ✗ | ○○○○ ✗ | ○○○○ ✗ | 1 |
| ✺ InternVL 3.5 (38B) | ●○○ ⏱ | ○○○○ ⏱ | ●○○○ ⏱ | ○○○ ⏱ | ○○○ ⏱ | ○○○ ✗ | ○○○ ✗ | ●○○ ⏱ | ○○○○ ⏱ | ●○○○ ✗ | 0 |
| ✹ Qwen3.5 (27B) | ○○○ ✗ | ●●○○ ⏱ | ○○○○ ✗ | ●○○ ⏱ | ○○○ ⏱ | ○○○ ⏱ | ○○○ ⏱ | ●○○ ✗ | ○○○○ ⏱ | ○○○○ ✗ | 0 |

Icons represent mission outcome: ✓ solved, ✗ strikeout, ⏱ timeout.
Each bullet represents one module: ● solved, ○ unsolved.

compared with only 10% in the asynchronous setting (see Appendix G.4 for the full comparison). However, models continue to exceed the time limit, with all models timing out on Mission #5. This suggests that timeouts may also result from the specific combination of module difficulty and available game time.[11]

**Module difficulty shapes mission difficulty.** Within the scope of the three- and four-module missions we test, module difficulty appears to be the most influential factor for overall success. Models struggle most on the two hardest missions (#9 and #10), which each contain three of the more challenging module types. The effect of module difficulty is inextricably intertwined with other characteristics of multi-module missions. Every additional module is a potential visual distractor and source of ambiguity, and models must understand and utilise incremental feedback as each module is solved.

## 6 Evaluating Models on Simplified Missions

Single-module missions reduce the planning and perceptual demands of multi-module settings and allow us to evaluate model performance on each module type in isolation. Critically, however, this isolation does not reduce the task to triviality: a single-module bomb still constitutes a sequential decision-making problem, requiring multiple perception-action loops between agents to resolve. For each module type (illustrated in Figure 3), we sample ten sufficiently distinct missions.[12] In line with §5, we consider the pass@1 across all ten missions of a given module type.

### 6.1 Asynchronous Single-Module Gameplay Results

**Models achieve limited success on single-module missions.** As shown in Table 4, all models are able to solve several single-module missions with the game running in real time. This is in contrast with models' performance on the original missions with multiple modules (Table 2), where models are unable to solve *any* mission. Gemini is the best-performing model in this setting, with an average success rate of 14.5%, solving at least one mission on around half of the module types, while in the original multi-module missions, the model solves fewer individual modules than the other closed-source models. This suggests a disconnect between how reliably different models can solve individual modules *in isolation* and in the presence of distractors.

**In simplified missions, module difficulty begins to emerge.** The single-module results broadly reflect the intended difficulty level for the different module types (see Figure 3), but reveal additional nuances within the difficulty tiers. Models are most successful on 🔲 *Wires*, with all models solving at least two missions,

---

[11]Note that Missions #4-6 have a reduced time limit, corresponding to level 2.4 in the game. See Appendix B.3 for further details.
[12]We describe the protocol for determining the seeds used to instantiate the missions in Appendix D.1.

Table 4: Success rates (%) for self-play on single-module missions in asynchronous mode, where the game clock continues to run down while models are generating. We sample ten distinct missions per module type. Each mission is attempted once (pass@1).

| | | | | | | | | | | | | Avg. |
|---|---|---|---|---|---|---|---|---|---|---|---|---|
| Sonnet 4.6 | 60.0 | 30.0 | 30.0 | 0.0 | 0.0 | 0.0 | 0.0 | 0.0 | 0.0 | 0.0 | 20.0 | 12.7 |
| Gemini 3 Flash | 20.0 | 10.0 | 40.0 | 0.0 | 20.0 | 50.0 | 0.0 | 20.0 | 0.0 | 0.0 | 0.0 | 14.5 |
| GPT-5.2 | 30.0 | 60.0 | 0.0 | 0.0 | 0.0 | 0.0 | 0.0 | 0.0 | 0.0 | 0.0 | 0.0 | 8.2 |
| InternVL 3.5 (38B) | 30.0 | 10.0 | 10.0 | 0.0 | 0.0 | 0.0 | 0.0 | 0.0 | 0.0 | 0.0 | 0.0 | 4.5 |
| Qwen3.5 (27B) | 30.0 | 40.0 | 10.0 | 0.0 | 0.0 | 0.0 | 0.0 | 0.0 | 0.0 | 0.0 | 0.0 | 7.3 |
| *Average* | 34.0 | 30.0 | 18.0 | 0.0 | 4.0 | 10.0 | 0.0 | 4.0 | 0.0 | 0.0 | 4.0 | 9.5 |

followed by ▣ *Button* and ▦ *Keypad*, for which almost all models are able to solve at least a single mission. Only Gemini succeeds on ▤ *Who's On First* and ▣ *Memory* missions, which correspond to the intermediate module types. Models struggle with some of the module types in the most challenging tier—no model is able to solve a single mission for ▦ *Complicated Wires* and ✕ *Wire Sequence*. Observed difficulty deviates from the intended tiers for four modules: ◈ *Simon Says* and ▦ *Maze* prove harder than other intermediate modules—no model solves either—while ▤ *Morse Code* and ▦ *Passwords* prove easier than other challenging ones.

## 6.2 Synchronous Single-Module Gameplay Results

While the results in §6.1 show that models can solve *some* missions with a single module, the real-time setting conflates problem-solving capability with time pressure: a failure may be the result of a missing skill or simply insufficient time to apply it. We therefore repeat the single-module evaluation with the synchronous setting, freezing the game clock while models take turns generating. Table 5 reports success rates per module type, and illustrates the change in performance compared to the asynchronous setting in Table 4.

Table 5: Success rates (%) for self-play on single-module missions in synchronous mode, where we stop the game clock while models take turns generating. We sample ten distinct missions per module type. Each mission is attempted once (pass@1).

| | | | | | | | | | | | | *Avg.* |
|---|---|---|---|---|---|---|---|---|---|---|---|---|
| Sonnet 4.6 | 80.0▲ | 50.0▲ | 60.0▲ | 0.0 | 20.0▲ | 80.0▲ | 10.0▲ | 10.0▲ | 20.0▲ | 0.0 | 100.0▲ | 39.1▲ |
| Gemini 3 Flash | 20.0 | 30.0▲ | 40.0 | 0.0 | 50.0▲ | 60.0▲ | 10.0▲ | 0.0▼ | 0.0 | 10.0▲ | 10.0▲ | 20.9▲ |
| GPT-5.2 | 60.0▲ | 80.0▲ | 30.0▲ | 30.0▲ | 40.0▲ | 80.0▲ | 0.0 | 10.0▲ | 10.0▲ | 10.0▲ | 70.0▲ | 38.2▲ |
| InternVL 3.5 (38B) | 40.0▲ | 40.0▲ | 20.0▲ | 0.0 | 10.0▲ | 0.0 | 0.0 | 0.0 | 30.0▲ | 0.0 | 0.0 | 12.7▲ |
| Qwen3.5 (27B) | 40.0▲ | 40.0 | 30.0▲ | 0.0 | 0.0 | 0.0 | 0.0 | 0.0 | 0.0 | 0.0 | 10.0▲ | 10.9▲ |
| *Average* | 48.0▲ | 48.0▲ | 36.0▲ | 6.0▲ | 24.0▲ | 44.0▲ | 4.0▲ | 4.0 | 12.0▲ | 4.0▲ | 38.0▲ | 24.4▲ |

Indicators compare success against the asynchronous setting (Table 4).

**Collaborative gameplay itself is a bottleneck.** While stopping the game clock during generation and introducing turn-taking provides a notable boost across all models and module types, we reveal that the real-time constraints are not the only bottleneck. Even in the simplest setting, where we simplify missions to single modules *and* grant models unlimited time to generate, only one-in-four missions are solved (Table 5). Both open- and closed-source models fail entirely on several module types, and even the strongest closed-source model cannot surpass an overall success rate of 39.1%. This shows that models struggle with the core demands of our collaborative gameplay setting, and not solely the complexities introduced by real-time pressure or missions with multiple modules.

**The gap between larger and smaller models becomes more apparent.** Sonnet and GPT achieve the highest average success rates (39.1% and 38.2% respectively), with Gemini performing around average at 20.9%. The closed-source models perform substantially better than the open-source models, solving around 3× as many missions as InternVL (12.7%) and Qwen (10.9%). This gap between open- and closed-source models likely reflects model scale, which is consistent with broader findings in the MLLM literature that more complex tasks benefit from scale (Duan et al., 2024; Lee et al., 2025; Li et al., 2025a; Wei et al., 2022a).

**Some module types become easier than the intended difficulty.** Performance in the synchronous setting (Table 5) generally reflects the intended difficulty (Figure 3)—with two notable exceptions. For models, the difficulty of the intermediate 🀫 *Memory* aligns with the easier tier, which is plausible given every observation the Defuser makes is relayed to the Expert in text, converting what is a demanding working-memory task for humans into a context-retrieval task that models handle well. A similar shift becomes evident for 🀫 *Passwords*, which is categorised as a hard module[13] but presents as much less challenging. This module requires cycling through letter slots to find a valid word, a process that is procedurally straightforward once the time pressure is removed, with no implicit state to construct and no perceptual ambiguity.[14]

**Poor performance on certain module types reveals weaknesses in state tracking.** The modules that models struggle most with share a common structure: success requires constructing an *implicit* representation of the game state across turns, whether a growing colour sequence (◈ *Simon Says*), a cumulative wire-colour count (◻ *Wire Sequence*), a decoded letter sequence (🀫 *Morse Code*[15]), or a spatial map of a 6×6 grid (🀫 *Maze*). These point to two distinct aspects of state tracking: losing detail when observations are not preserved in the dialogue (as in 🀫 *Maze* and ◻ *Wire Sequence*), and struggling to integrate multiple observations within a single turn (as in ◈ *Simon Says* and 🀫 *Morse Code*).

## 7 Investigating Collaboration Patterns

The preceding sections evaluate models under self-play conditions, where both roles are filled by the same model. This establishes per-model performance on the benchmark, but real-world collaboration often requires adapting to different partners with varying capability levels and communication approaches (Hu et al., 2020; Stone et al., 2010). In this section, we expand the evaluation to all pairwise model combinations to determine how performance patterns vary with each model's role and partner. This also allows us to explore the relationship between communication rate—the proportion of the game that models spend communicating—and model pairings. For our pairwise evaluations, we use the simplified single-module missions in synchronous mode to allow us to surface nuances in behaviour.

### 7.1 Influence of Role and Partner

Table 6 reports the success rate of each of the 25 pairings on single-module missions in synchronous mode. Success rates across pairings vary widely, ranging from 6.4% to 40.9%, indicating that performance is highly dependent on the exact model combination.

**Performance varies more across Defusers than across Experts.** The spread of average performance across Defusers (22.6%) is wider than that across Experts (13.5%), confirming the intuition that the Defuser has a greater influence on overall success. A strong Expert cannot compensate for a weak Defuser: Qwen's poor performance as a Defuser improves somewhat with stronger Expert partners, but never approaches the stronger Defusers'. No Expert can make up for InternVL's shortcomings, pointing to a fundamental limitation in its Defuser capabilities. We return to the nature of this bottleneck in §9.2, where targeted offline evaluations isolate perception and reasoning failures.

---

[13]The asynchronous results in Table 4 corroborate this assessment for real-time gameplay.

[14]To solve 🀫 *Passwords* missions, Sonnet performs on average 44.3 click actions in synchronous mode, while managing to complete an average of 34.1 clicks within the time limit in the asynchronous mode.

[15]We observe models getting stuck in runaway repetition of the visual signal (e.g., "dot dash dash dot...").

Table 6: Success rate (%) of all model pairings for single-module missions in synchronous mode.

| | | Expert | | | | | |
| | | ✳ Sonnet 4.6★ | ✦ Gemini 3 Flash | ⟲ GPT-5.2 | 🗲 InternVL 3.5 | 🗲 Qwen3.5 | *Average* |
|---|---|---|---|---|---|---|---|
| **Defuser** | ✳ Sonnet 4.6★ | 39.1 | 40.9 | 32.7 | 20.9 | 23.6 | 31.5 |
| | ✦ Gemini 3 Flash | 37.3 | 20.9 | 13.6 | 19.1 | 11.8 | 20.5 |
| | ⟲ GPT-5.2 | 36.4 | 33.6 | 38.2 | 22.7 | 22.7 | 30.7 |
| | 🗲 InternVL 3.5 | 10.9 | 6.4 | 6.4 | 12.7 | 8.2 | 8.9 |
| | 🗲 Qwen3.5 | 16.4 | 11.8 | 12.7 | 11.8 | 10.9 | 12.7 |
| | *Average* | 28.0 | 22.7 | 20.7 | 17.5 | 15.5 | 20.9 |

★Highest average success rate for the role, taken over all pairings (*Average* row and column). Highest success rate of any single pairing underlined.

**Models differ in their suitability for each role.** Comparing each model's average performance as Defuser and as Expert reveals an asymmetric role affinity. While Sonnet is the strongest player in both roles, the remaining model rankings reorder across roles. For instance, Gemini is stronger as an Expert (22.7%) than as a Defuser (20.5%), while GPT shows the reverse pattern, averaging 30.7% as Defuser against 20.7% as Expert. The direction of this asymmetry differs across models, pointing to role-specific suitability rather than a uniform difference in role difficulty.

**Partner compatibility shapes performance.** Gemini illustrates compatibility sensitivity most evidently: peak performance overall is achieved by Sonnet as Defuser paired with Gemini as Expert (40.9%), a result not predicted by Gemini's overall Expert average. Yet with GPT as Expert, Gemini drops to 13.6%, while with InternVL as Expert, it performs nearly as well as with itself, despite InternVL being among the weakest Experts. These results suggest that collaborative performance is driven by compatibility rather than partner capability alone. One plausible cause is divergent communication styles shaped during post-training, though we leave this to future work.

## 7.2 Communication Rates Across Model Pairings

At any given point in a mission, models choose whether to send a message, wait, or in the case of the Defuser, interact with the bomb. We analyse the proportion of game steps involving a message from either of the players to reveal how models approach collaboration.[16]

**Communication rates differ in self-play and other-play.** As Table 7 summarises, on average, models dedicate more game time to communication in other-play, that is, when paired with a partner from a different model class. For some models, this holds regardless of the role—whether acting as the Defuser or the Expert, they consistently communicate more frequently when paired with another model. We specifically observe this for Sonnet. Other models only exhibit this behaviour in other-play when acting as the Defuser (GPT and InternVL) or as the Expert (Gemini and Qwen). Models collaborate in an ad hoc teamwork fashion (Hu et al., 2020; Stone et al., 2010): they are unaware who they are partnered with, and any adaptation is derived purely from the content and style of the messages they receive.

Table 7: Communication rates (% game steps with messages) for self- vs. other-play.

| Communicate with | Self | Other | |
|---|---|---|---|
| | | as Defuser | as Expert |
| ✳ Sonnet 4.6 | 22.5 | 34.0 | 26.3 |
| ✦ Gemini 3 Flash | 25.5 | 36.6 | 24.4 |
| ⟲ GPT-5.2 | 35.5 | 27.1 | 38.7 |
| 🗲 InternVL 3.5 | 34.3 | 30.6 | 41.0 |
| 🗲 Qwen3.5 | 30.0 | 32.6 | 30.4 |
| *Average* | 29.6 | 32.2 | 32.2 |

---

[16]While we focus on the self- vs. other-play differences, we compare communication rates across all model pairings in Appendix G.6.

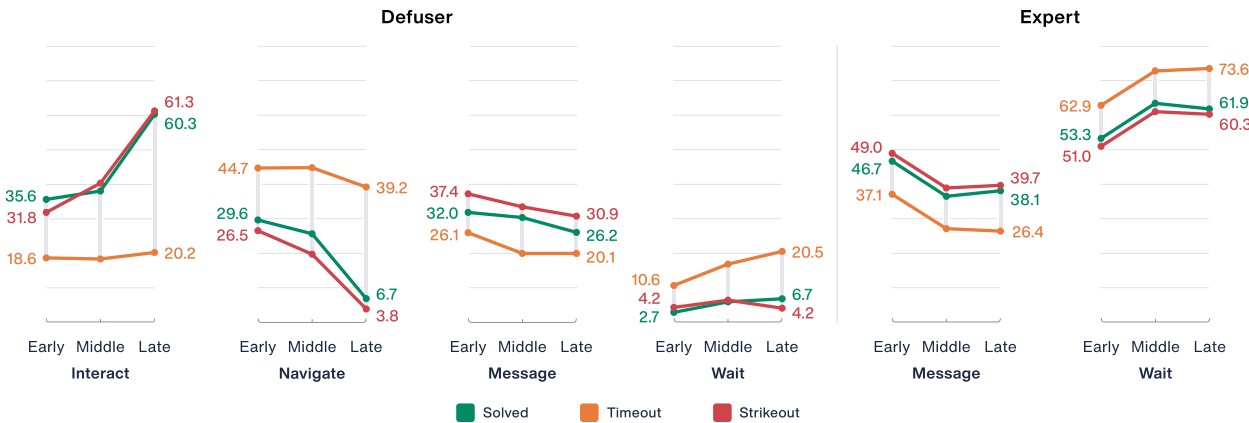

Figure 7: Distribution of action types (% of game steps) by role and mission outcome across the early, middle, and late phases of games played by all possible model pairings on single-module missions in synchronous mode.

## 8 Analysing How Models Fail

A mission succeeds only if every module is solved before the bomb reaches three strikes or runs out of time. We track strikeouts and timeouts separately, and examine what behavioural patterns distinguish them.

**Timeouts and strikeouts have distinct signatures.** Failed games are characterised by structurally different behavioural patterns depending on the outcome (Figure 7). Games ending in a timeout have a distinctive shape: Defusers spend roughly twice as much of the time played navigating around the bomb as those that succeed or strike out, and interactions with modules are almost half as frequent. Experts mirror this behaviour by choosing to wait increasingly across game phases. Neither role fills the gap with communication, indicating a mutual disengagement. Games ending in a strikeout look mostly similar to solved games. The one consistent difference is messaging: both roles communicate more on games with a strikeout than on either solved or timed-out ones, suggesting that the players are unable to reach sufficient common ground.

**Recovery is possible, but rare.** In 56% of synchronous, single-module missions, the players accumulate at least one strike. Figure 8 traces how those missions unfold. The single most common event following a strike is another strike, suggesting that models continue on the same course rather than correcting. While a small share of missions with one or even two strikes are still completed, demonstrating that recovery is possible, receiving a strike is a stronger predictor of further failure than any other outcome. A considerable share of missions with one or two strikes ultimately end in timeouts, indicating that models may respond to strikes by disengaging entirely. Whether these patterns reflect compounding errors from a corrupted reasoning context, an inability to recognise the strike as a meaningful signal and identify an alternative strategy, or a combination, cannot be disentangled without further systematic tests—we leave this to future work.

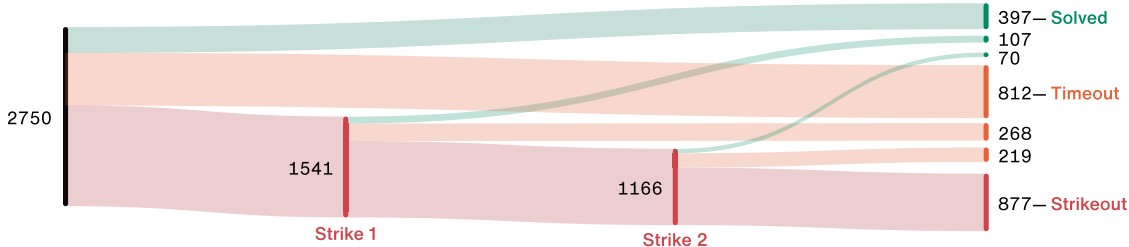

Figure 8: Frequency of game outcomes depending on strikes across games played by all possible model pairings on single-module missions in synchronous mode.

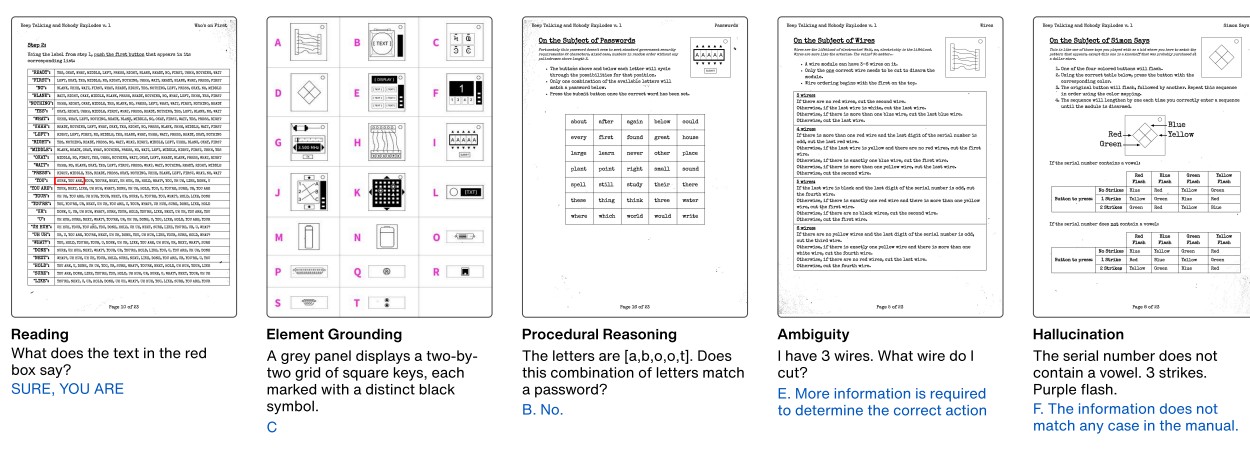

Figure 9: Examples for each task in the offline Manual VQA.

# 9 Isolating Underlying Capabilities Offline

Interactive performance requires, and consequently conflates, many individual capabilities. To isolate these from the multi-turn demands of gameplay, we conduct targeted single-step offline evaluations for each role.

## 9.1 Manual VQA

Given the manual and descriptions of what the Defuser sees, the Expert must identify the correct module type and corresponding page(s), determine what additional information they need from the Defuser, and use all available information to arrive at the correct instructions using deductive reasoning. While solutions are deterministic, the path to them can be derailed by ambiguity or inaccurate information supplied by the Defuser.

We isolate five distinct capabilities that a strong Expert should possess—reading comprehension, element grounding, procedural reasoning, ambiguity detection, and hallucination detection—and generate 884 targeted multiple-choice questions to assess each one independently. These tasks are illustrated in Figure 9, and full construction details are provided in Appendix D.2.

Table 8: Accuracy (%) on offline Manual VQA tasks by model and capability measured.

|  | Reading | Element Grounding | Procedural Reasoning | Ambiguity | Hallucination | *Average* |
|---|---|---|---|---|---|---|
| Sonnet 4.6 | 75.9 | 73.3 | 67.1 | 51.6 | 62.9 | 65.3 |
| Gemini 3 Flash | 81.2 | 88.3 | 64.1 | 51.6 | 50.0 | 69.3 |
| GPT-5.2 | 65.4 | 90.0 | 61.5 | 59.4 | 61.4 | 64.4 |
| InternVL 3.5 (38B) | 29.3 | 38.3 | 44.5 | 23.4 | 25.7 | 36.1 |
| Qwen3.5 (27B) | 68.6 | 55.0 | 53.7 | 46.9 | 48.6 | 56.0 |
| *Average* | 64.1 | 69.0 | 58.2 | 46.6 | 49.7 | 58.2 |

**Models exhibit distinct deficiencies in individual capabilities.** Results in Table 8 show that the Manual VQA tasks are non-trivial, with models achieving between 36.1 and 69.3% overall accuracy. Despite comparable overall performance, the larger closed-source models show divergent capability profiles. Gemini's overall performance is driven by strong reading comprehension and element grounding, while GPT peaks on element grounding. Sonnet, by contrast, is similarly capable on perceptual and reasoning tasks. However, no model achieves a perfect accuracy on any task. Therefore, if any deficiency manifests at a single step, it can compound across multiple turns, leading to further deviations and errors.

**Assessing information reliably is the crux.** Both ambiguity and hallucination detection lag well behind perceptual and reasoning performance. GPT leads in detecting incomplete information, while Sonnet is the most capable at identifying hallucinated inputs that do not match the manual descriptions. This is consistent with qualitative observations, where Sonnet more readily withholds instructions and requests clarification rather than reasoning over unreliable inputs.[17] Whether these failures reflect a reasoning limitation or a tendency toward sycophancy—where models are more likely to leave incorrect beliefs or information unchallenged (Ibrahim et al., 2026)—remains an open question.

**Manual VQA trends do not predict gameplay trends.** For some of the models, we observe a notable disconnect between their offline, single-step performance and the interactive setting. The superior performance of Gemini across these isolated core capabilities does not consistently translate to competitive gameplay (Table 6). InternVL is the weakest model on Manual VQA by a large margin, but performs comparably to Qwen in the interactive setting. This suggests that strong perceptual and reasoning capabilities can only ever be the basis for effective collaboration.

## 9.2 Simulator VQA

The Expert's contribution is only as useful as the information the Defuser supplies—if the Defuser supplies the Expert with inaccurate or insufficient information, the instructions the Expert provides will in turn be incorrect. As the Defuser, a player must operate directly from visuals—reading characters, counting, and identifying elements based on their attributes—to describe the game state with enough precision for the Expert to reason over. We probe relevant abilities by generating 640 multiple-choice questions spanning basic perception (reading, colour, counting) and robustness probes for ambiguity and hallucination detection, examples for which are illustrated in Figure 10. Full dataset construction details are provided in Appendix D.3.

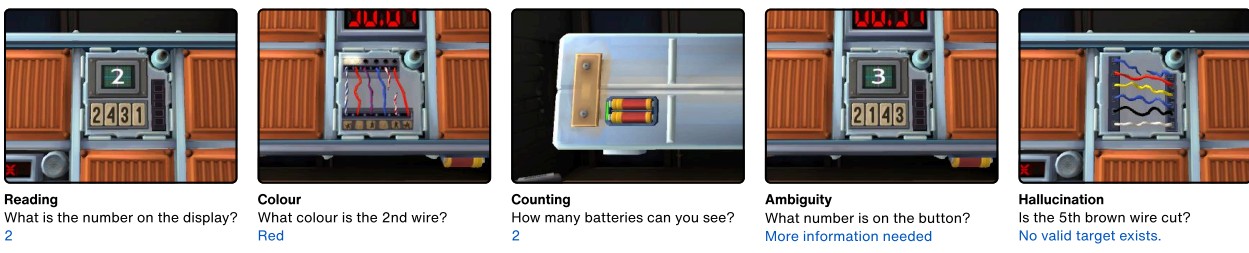

Figure 10: Examples from the offline Simulator VQA evaluation tasks.

Table 9: Accuracy (%) on offline Simulator VQA tasks by model and capability measured.

|  | Reading | Colour | Counting | Ambiguity | Hallucination | *Average* |
|---|---|---|---|---|---|---|
| ✳ Sonnet 4.6 | 96.7 | 82.0 | 47.8 | 6.6 | 77.9 | 63.0 |
| ✦ Gemini 3 Flash | 98.7 | 83.6 | 61.5 | 12.1 | 84.0 | 70.0 |
| ⬡ GPT-5.2 | 82.9 | 83.6 | 46.8 | 7.7 | 74.0 | 58.9 |
| 〰 InternVL 3.5 (38B) | 78.3 | 52.5 | 35.1 | 1.1 | 38.2 | 42.8 |
| ⬡ Qwen3.5 (27B) | 88.2 | 86.9 | 39.0 | 8.8 | 67.9 | 56.9 |
| *Average* | 88.9 | 77.7 | 46.0 | 7.3 | 68.4 | 58.3 |

**Reading is easy, counting is hard.** With an average performance of 88.9%, reading alphanumeric characters on bomb elements is a seemingly straightforward task for models, confirming that Defusers are given a considerably simpler visual parsing task compared with the text-heavy manual pages the Expert must process (see Table 8). Colour recognition is relatively more challenging (77.7%), and all models perform worst

---

[17]See examples in Figures G.4 and G.9.

on counting (46.0%). This is not surprising: while cardinality is trivial for humans, it remains a persistent weakness for large models (Paiss et al., 2023; Vo et al., 2025). The harder the task, the more apparent the gap between the strongest and weakest model becomes; on counting tasks, the strongest model's accuracy is almost double that of the weakest (61.5% vs. 35.1%). The capabilities we measure are likely not simply scale-dependent: on the easier tasks, Qwen's performance is on par with that of the closed-source models.

**Detecting ambiguity is a bottleneck.** Models can detect when they are asked or instructed about elements that do not exist in the image, averaging 68.4%. However, all models struggle to detect ambiguous, underspecified questions. Even Gemini, the strongest model on most tasks, only achieves 12.1%, and InternVL fails almost entirely, scoring only 1.1%. This suggests that models can gauge when a question or instruction refers to an element that is clearly absent, but struggle when a description fits several elements. This hints at an important downstream bottleneck in communication between two players: even if the Expert and Defuser were maximally capable, the information asymmetry and time pressure make it almost impossible to avoid ambiguous exchanges entirely (Lemon, 2022; Piantadosi et al., 2012), and knowing when to ask for clarification rather than proceeding under uncertainty becomes a critical skill (Chiyah-Garcia et al., 2024).[18]

**Low-level skills only partly explain in-game performance.** Echoing the findings in §9.1, Gemini is the best-performing model across the Simulator VQA tasks, outperforming the other models by a large margin on the more difficult counting task, despite middling in-game performance as Defuser. On the other hand, InternVL's consistently poor performance on Simulator VQA tasks suggests that its in-game performance as Defuser is constrained by inadequate low-level capabilities.

### 9.3 Simulator Localisation Results

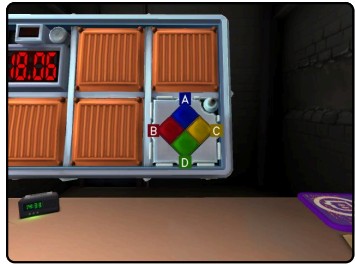

(a) *Click on the green button.*

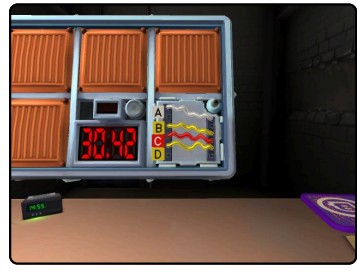

(b) *Click on the first wire.*

Figure 11: SoM localisation.

In KTANE, the Defuser must not only perceive, but also act on what it perceives. To interact with UI elements, the Defuser must localise the correct element according to the Expert's instructions. As discussed in §4, we facilitate localisation via *set-of-marks* (SoM; Yang et al., 2023). Labelled segmentation masks are overlaid on the image, reducing the action space to selecting the correct label. We distil this into an offline task consisting of 665 concise instructions, as illustrated by the examples in Figure 11, and report the results in Table 10. We also provide a comparison between SoM and direct coordinate prediction in Appendix G.10.

**Fine-grained localisation remains challenging.** A clear pattern is the prevalence of perfect accuracy across multiple modules, indicating that models are broadly capable of correctly identifying and targeting relevant elements given direct localisation instructions. This corresponds to modules with sparse and visually distinct interaction targets, such as 🔲 *Button*, 🔷 *Simon Says* and ▦ *Maze*, suggesting that any downstream failures on these modules are unlikely to be grounding failures. By contrast, modules that require interaction with smaller or more densely-packed elements—such as ▤ *Wires*, ▤ *Who's On First*, ✕ *Wire Sequence* and ▦ *Passwords*—prove more challenging across models. This aligns with the intuition that models struggle when targets are smaller or when the screen contains many closely-clustered elements, making precise selection more difficult.

**Localisation failure acts as a hard ceiling on downstream success.** This is especially prominent for InternVL, which fails to demonstrate sufficient localisation capabilities on the majority of modules. This fundamental gap is further evident in InternVL's in-game behaviour: 28.4% of its attempted interactions with modules fail because the model hallucinates a location marker. Conversely, we observe a number of cases where the open-source models mistake the location marker for a part of the UI, despite extensive guidance

---

[18]We include an example where the Expert fails to detect implausible information from the Defuser in Figure G.18.

Table 10: Accuracy (%) on offline Simulator Localisation tasks, and in-game localisation errors.

| | M1 | M2 | M3 | M4 | M5 | M6 | M7 | M8 | M9 | M10 | M11 | *Avg.* | Errors[*] |
|---|---|---|---|---|---|---|---|---|---|---|---|---|---|
| ☀ Sonnet 4.6 | 64.3 | 100.0 | 71.7 | 100.0 | 77.8 | 95.5 | 100.0 | 100.0 | 74.0 | 60.9 | 76.5 | 80.6 | 0.4 |
| ◆ Gemini 3 Flash | 64.3 | 100.0 | 98.1 | 100.0 | 53.7 | 100.0 | 100.0 | 100.0 | 74.0 | 69.6 | 97.1 | 83.9 | 0.0 |
| ⑤ GPT-5.2 | 71.4 | 100.0 | 81.1 | 100.0 | 68.5 | 95.5 | 100.0 | 96.9 | 72.0 | 67.4 | 64.7 | 80.4 | 0.2 |
| ⩙ InternVL 3.5 (38B) | 50.0 | 95.5 | 71.7 | 66.7 | 22.2 | 18.2 | 100.0 | 65.6 | 58.0 | 37.0 | 23.5 | 53.7 | 28.4 |
| 🜲 Qwen3.5 (27B) | 64.3 | 100.0 | 73.6 | 100.0 | 70.4 | 100.0 | 100.0 | 96.9 | 64.0 | 58.7 | 58.8 | 76.8 | 11.4 |
| *Average* | 62.9 | 99.1 | 79.2 | 93.3 | 58.5 | 81.8 | 100.0 | 91.9 | 68.4 | 58.7 | 64.1 | 75.1 | 8.1 |

[*] Proportion of attempted module interactions where the model hallucinates the location marker during *interactive* gameplay.

for identifying location markers in the system prompt.[19] For the remaining models, however, localisation accuracy is high. This suggests that solving modules like ◆ *Simon Says*, ▤ *Morse Code*, and ▦ *Maze* is hard not because models are unable to ground instructions to the correct element, but because correctly interpreting and acting on the module's state requires complementary capabilities.

# 10 Ablating Collaboration and Surfacing Contamination

Information asymmetry forces every exchange through an imperfect channel that introduces noise and ambiguity, especially in other-play, where models are partnered up with different models (see §7.1). We sever this channel and devise two single-agent scenarios to single out the effect of communication on the one hand, and prior exposure to KTANE on the other. While both violate the rules of the game, these settings serve as ablations to probe the importance of collaboration within the benchmark. We run both single-agent ablations on the single-module missions in synchronous mode, and compare them to the equivalent collaborative setting.

## 10.1 Single-Agent Results

To test the effect of information asymmetry, we introduce a single agent ablation giving the model direct access to both the game and the manual. As shown in Table 11, most models improve when given direct access to both the bomb and the manual, confirming that the communication channel itself is a source of error in the collaborative setting. Only Gemini is impacted negatively overall, suggesting that the structure of the role specialisation outweighs the overhead imposed by the information asymmetry. At the module level, effects are mixed, creating two broad groups: modules where solo access helps, and modules where it does not, which we analyse in turn below.

**Decisions no longer rely on second-hand information.** Performance increases across several modules, with most consistent improvements on the easier ones, ▤ *Wires*, ⊙ *Button*, and ▦ *Keypad*. In the collaborative setting, the Expert must incrementally guide the Defuser to collect specific details relevant to the module, such as batteries, indicators, or the serial number. In turn, the Defuser must communicate them accurately before the correct instruction can be identified. A single agent, on the other hand, has complete situational awareness from the outset and can apply the rules directly without first establishing shared understanding. Additionally, as shown in §9.1, models struggle to identify ambiguous or potentially hallucinated information from a partner. Removing the communication channel eliminates this error source entirely.

**Shared references simplify gameplay.** Improvements on ▦ *Keypad* indicate that communication is especially costly when there is a need to negotiate a shared vocabulary with a partner. When we isolate the symbol-description step as a VQA-style task (more details in Appendix G.8), different models generate different descriptions for the same symbol on more than half of the questions. For instance, ☻ is described as 'smiley face' by Gemini and GPT, but 'magnet' by Qwen; the only symbol all models (implicitly) agree on is ☆, which they all call a 'star'.

---

[19]We provide examples of this behaviour in Figures G.4, G.9 and G.14 and the system prompt in Figure H.14.

Table 11: Success rate (%) on simplified single-module missions in synchronous mode for agents playing in pairs (with any partner) vs single agents with and without manual access.

| | | 🏳 | 🔘 | 🔢 | 🔷 | 📋 | 1 | ▦ | ⌨ | ✖ | ✕ | TNT | *Avg.* |
|---|---|---|---|---|---|---|---|---|---|---|---|---|---|---|
| | *Random Baseline* | 13.8 | 7.0 | 0.0 | 0.0 | 0.1 | 0.0 | 2.1 | 0.9 | 5.5 | 0.0 | 0.0 | 2.7 |
| **Pairwise**[a] | ✳ Sonnet 4.6 | 70.0 | 48.0 | 20.0 | 0.0 | 28.0 | 58.0 | 8.0 | 18.0 | 28.0 | 4.0 | 64.0 | 31.5 |
| | ✦ Gemini 3 Flash | 36.0 | 56.0 | 24.0 | 2.0 | 20.0 | 40.0 | 12.0 | 4.0 | 6.0 | 18.0 | 8.0 | 20.5 |
| | Ⓢ GPT-5.2 | 70.0 | 66.0 | 32.0 | 12.0 | 24.0 | 54.0 | 6.0 | 4.0 | 14.0 | 6.0 | 50.0 | 30.7 |
| | ⚡ InternVL 3.5 (38B) | 30.0 | 30.0 | 10.0 | 0.0 | 4.0 | 4.0 | 2.0 | 0.0 | 18.0 | 0.0 | 0.0 | 8.9 |
| | 🐦 Qwen3.5 (27B) | 32.0 | 40.0 | 16.0 | 0.0 | 6.0 | 20.0 | 6.0 | 0.0 | 2.0 | 0.0 | 18.0 | 12.7 |
| | *Average* | 47.6 | 48.0 | 20.4 | 2.8 | 16.4 | 35.2 | 6.8 | 5.2 | 13.6 | 5.6 | 28.0 | 20.9 |
| **No partner**[b] | ✳ Sonnet 4.6 | 90.0▲ | 90.0▲ | 70.0▲ | 0.0 | 20.0 | 20.0▼ | 0.0 | 10.0 | 40.0▲ | 0.0 | 100.0▲ | 40.0▲ |
| | ✦ Gemini 3 Flash | 10.0▼ | 60.0 | 20.0 | 0.0 | 0.0▼ | 20.0▼ | 0.0▼ | 30.0▲ | 0.0 | 0.0▼ | 0.0 | 12.7▼ |
| | Ⓢ GPT-5.2 | 100.0▲ | 80.0▲ | 60.0▲ | 10.0 | 30.0 | 50.0 | 0.0 | 0.0 | 40.0▲ | 0.0 | 10.0▼ | 34.5▲ |
| | ⚡ InternVL 3.5 (38B) | 70.0▲ | 30.0 | 30.0▲ | 0.0 | 0.0 | 0.0 | 10.0 | 0.0 | 10.0 | 0.0 | 0.0 | 13.6▲ |
| | 🐦 Qwen3.5 (27B) | 80.0▲ | 40.0 | 30.0▲ | 0.0 | 10.0 | 30.0▲ | 0.0 | 20.0▲ | 40.0▲ | 0.0 | 20.0 | 24.5▲ |
| | *Average* | 70.0▲ | 60.0▲ | 42.0▲ | 2.0 | 12.0▼ | 24.0▼ | 2.0▼ | 12.0▲ | 26.0▲ | 0.0▼ | 26.0▼ | 25.1▲ |
| **No manual**[c] | ✳ Sonnet 4.6 | 50.0▼ | 20.0▼ | 40.0▲ | 0.0 | 10.0▼ | 10.0▼ | 10.0 | 10.0 | 70.0▲ | 0.0 | 70.0 | 26.4▼ |
| | ✦ Gemini 3 Flash | 10.0▼ | 60.0 | 40.0▲ | 0.0 | 10.0▼ | 0.0▼ | 0.0▼ | 0.0 | 10.0 | 0.0▼ | 0.0 | 11.8▼ |
| | Ⓢ GPT-5.2 | 100.0▲ | 90.0▲ | 0.0▼ | 20.0 | 0.0▼ | 10.0▼ | 0.0 | 0.0 | 20.0 | 0.0 | 20.0▼ | 23.6▼ |
| | ⚡ InternVL 3.5 (38B) | 10.0▼ | 30.0 | 10.0 | 0.0 | 0.0 | 0.0 | 0.0 | 10.0▲ | 60.0▲ | 0.0 | 0.0 | 10.9▲ |
| | 🐦 Qwen3.5 (27B) | 70.0▲ | 0.0▼ | 0.0▼ | 0.0 | 10.0 | 0.0▼ | 10.0 | 10.0▲ | 20.0▲ | 10.0▲ | 0.0▼ | 11.8 |
| | *Average* | 48.0 | 40.0▼ | 18.0▼ | 4.0 | 6.0▼ | 4.0▼ | 4.0▼ | 6.0 | 36.0▲ | 2.0▼ | 18.0▼ | 16.9▼ |

[a] Pairwise collaboration aggregated over all Expert partners.
[b] Single-agent play with access to the manual.
[c] Single-agent play without access to the manual, testing parametric game knowledge.
Symbols show changes from Pairwise, with a minimum change of one mission at that aggregation level.

**A partner helps where state-tracking demands are high.** For modules with multiple stages (📋 *Who's On First*, 1 *Memory*, ✕ *Wire Sequence*), removing the partner appears to harm performance. ✕ *Wire Sequence* provides the clearest illustration: tracking which wire colours have already appeared across up to twelve sequential steps is a working memory demand that direct manual access does not address—and one that is compounded by the need to integrate information across an increasingly long context. Performance on 🔷 *Simon Says* and ▦ *Maze* remains near zero across all conditions, pointing to state-tracking as a capability gap that persists regardless of how much information access is simplified.

## 10.2 Contamination Results

While commonsense or prior knowledge from other domains does not directly apply to the problem-solving rules of the game, it is possible that, given the game's substantial online presence since its 2015 release, models have encountered KTANE-related content during pre-training.[20] Models are, in principle, able to memorise the defusal manual. Solutions are presented as branching decision trees—which is precisely the kind of structured, rule-based content that models memorise without difficulty (Hartmann et al., 2023; Kiyomaru et al., 2024; Wang et al., 2024b). Although we explicitly prompt models to rely solely on the information provided (Appendix H), this cannot—and does not—prevent them from drawing on what they already know.[21]

---

[20] We confirm this in Appendix G.1, where models produce plausible game-specific responses without any provided context.
[21] We show one example of models explicitly using parametric knowledge in Appendix G.3.

For transparency, we probe whether models take such shortcuts by treating the task as a recall problem rather than a collaborative one (Cheng et al., 2025; Hermann et al., 2024; Hosseini et al., 2025; Jacovi et al., 2023). We test whether they can solve any missions solely by relying on their parametric knowledge, that is, *without access to a partner or the manual.*

**All models use their knowledge of *ktane*.** The results in Table 11 show that models attempting to solve a puzzle module without the manual outperform a random baseline.[22] This confirms that while it is possible to chance upon the correct solution for some module types, models generally possess and are using knowledge of the task from their training alone. However, the extent to which different models can leverage parametric knowledge varies. All models possess some parametric knowledge of both the ⊞ *Wires* and ⊞ *Complicated Wires* modules; GPT seems to possess sufficient parametric knowledge to solve *all ten* ⊞ *Wires* missions. In several cases, models perform better than in the collaborative setting. This suggests that models not only have sufficient parametric knowledge for these module types, but that the full collaborative setting introduces additional complexity that obscures parametric knowledge.

**Visual context scaffolds manual recall.** To disentangle the ability to recall knowledge of the manual acquired during training from the ability to accurately perceive and interact with the bomb, we return to the procedural reasoning questions from our offline tests introduced in §9.1 and evaluate models on the same questions without providing the manual. We focus on this question type as an indicator of the ability to deduce accurate instructions based solely on prior knowledge. As shown in Table 12, almost all models perform at or below random chance level. Gemini is the only exception showing meaningful performance. This stands in contrast to Defuser results, where models demonstrably apply parametric knowledge of module interactions. One plausible explanation for the asymmetry is that Defuser recall is scaffolded by visual context: seeing the module may act as a retrieval cue, making relevant actions easier to recover than the precise conditional logic of the manual, which must be retrieved without any perceptual anchor.

Table 12: Expert procedural reasoning accuracy, with and without manual access.

|  | Manual | No Manual |
|---|---|---|
| ☀ Sonnet 4.6 | 67.1 | 21.6▼ |
| ✦ Gemini 3 Flash | 64.1 | 36.5▼ |
| ⑨ GPT 5.2 | 61.5 | 19.6▼ |
| InternVL 3.5 | 44.5 | 22.0▼ |
| Qwen3.5 | 53.7 | 10.4▼ |
| *Average* | 58.2 | 22.0▼ |
| *Random*[†] | 21.3 | 21.3 |

[†] Based on the possible answer options for the multiple choice questions.

## 11 Conclusion

We introduce GPTNT, a framework that wraps the interactive 3D bomb defusal video game KTANE to provide a challenging testbed for multimodal, multi-agent collaboration in real time. GPTNT fills a gap in the existing MLLM evaluation landscape by being the first benchmarking environment to simultaneously require coordination under asymmetric information, asynchronous action under time pressure, visual grounding in a dynamic environment, and sustained multi-turn communication. This reflects an ecologically valid set of conditions that models could encounter in real-world deployment scenarios, making GPTNT an intentional step towards more holistic, integrated evaluation of MLLM skills.

We demonstrate that, unlike human players, current state-of-the-art models are unable to play a single successful game in the full setting, acting asynchronously on missions with multiple puzzle modules. Even the simplest setting we test, where models are given unlimited deliberation and generation time for less complex single-module missions, poses significant challenges for the models we test. These challenges are exacerbated in other-play, especially when models have to adapt to potentially weaker partners. Single-agent diagnostic evaluations reveal that, while models have been exposed to KTANE-specific data during training, the benchmark surfaces persistent limitations in both collaborative and low-level capabilities. Across online and offline evaluations, we find that models struggle to act efficiently under a time budget, maintain an accurate state representation across turns, recognise when information is underspecified or hallucinated, and recover from mistakes.

---

[22]The random baseline is computed by running an agent that selects uniformly at random from plausible actions at each step, averaged over 100 attempts per mission ($n = 1{,}000$ per module).

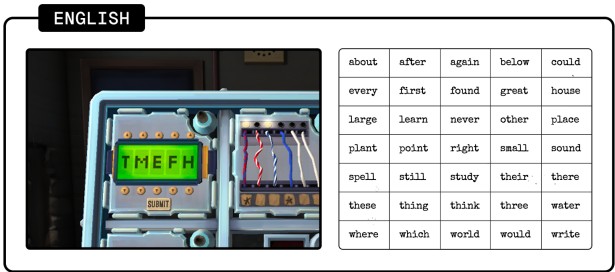 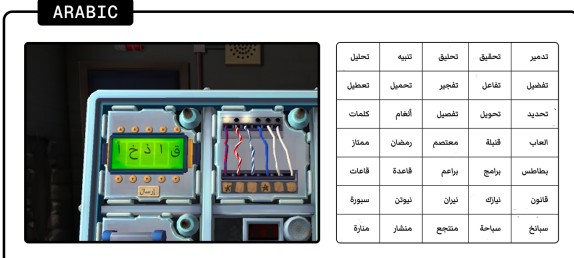

Figure 12: Example comparing 🎰 *Passwords* across English and Arabic in KTANE.

The GPTNT environment is uniquely suited to stay ahead of growing model capabilities and mitigate the risk of the benchmark becoming saturated. The set of core missions we introduce in this iteration of the benchmark only covers relatively low difficulty levels (2–3 of 7, see Appendix B.3), and more challenging missions can easily be configured and procedurally generated at scale. Furthermore, new puzzle modules contributed by an active modding community can be leveraged to increase mission difficulty while simultaneously reducing the effect of contamination as training corpora expand.

**Future Work**

KTANE provides the foundation for direct extensions of our benchmark across languages and modalities. The game's availability in 27 different languages can be leveraged for cross-lingual evaluation of communicative grounding without framework changes (Figure 12). The game supports so-called *needy* modules, modules that are activated randomly during the game. These modules signal that they require immediate attention not only visually, but also by emitting audio cues, which is crucial when the Defuser is viewing a different side of the bomb at the time of activation. As they require audio understanding and the ability to (re)prioritise, bombs including needy modules represent a qualitatively harder difficulty tier, which our framework will be able to accommodate once audio is supported by the framework. In a similar vein, native speech support would remove text-play artefacts such as the Unicode 🔢 *Keypad* workaround (see Appendix G.8), surface real-time collaboration phenomena that text suppresses, and enable human–agent experiments that do not place the burden on the human players (Chen & Yu, 2025; Cui et al., 2025).

Several behavioural phenomena we observe warrant further investigation. Models adapt their communication *rate* to different models in other-play (§7.2), but how *content* shifts—vocabulary, specificity, repair strategies—remains uncharacterised. The cascade from a first strike to a second, and potentially a third (Figure 8) could reflect compounding errors, an inability to leverage a strike as a feedback signal, or both. A principled investigation into models' understanding and ability to leverage visual feedback more broadly is therefore warranted. Whether models exhibit specific strategies or leverage in-context learning when solving bombs with multiple puzzle modules is another promising avenue for future research. Beyond zero-shot evaluation of models' existing capabilities, GPTNT can serve as a training environment, as long as the end goal is to improve performance on different held-out collaborative tasks. However, whether training on GPTNT can yield such generalisable performance improvements is orthogonal to this work.

## 12 Limitations

**Game Cost.** Running GPTNT requires purchasing a copy of *Keep Talking and Nobody Explodes*, available for $14.99 from the Humble store[23] (covering Windows, macOS, and Linux). We do not distribute the game or any source code from it, as it would violate its licence. We do, however, release all infrastructure to run the benchmark with our mod.

**API Cost.** Reproducing the full experimental suite costs approximately $3,000 in API spend, with each open-source model served locally on 2× H100 80GB GPUs. Cost also shapes model coverage: hard filters

---

[23]https://humblebundle.com/store/keep-talking-and-nobody-explodes

on context length and available compute bias the evaluation toward large closed-source models or smaller open-source ones, with the middle ground excluded. The resulting open–closed comparison conflates family with scale; the performance gap most likely reflects model size rather than anything intrinsic to the open-source paradigm. A breakdown of token usage and cost per model and experiment is provided in Appendix E.3.

**Configuration.** Resolution is fixed at 640×480—the smallest the game supports before UI elements clip—to minimise context length and the generation latency that erodes real game time in asynchronous play. Although results on offline evaluations (§9.1 and §9.2) show that models achieve strong performance on standalone reading probes, higher resolutions are worth exploring, as serving becomes faster and cheaper. Additionally, as coordinate grounding is currently unreliable (Appendix G.10), we treat set-of-marks interaction as standard, though both are supported and the choice can be revisited as models improve (Appendix C.6.4).

**Outputs.** Communication is text-only, placing models in their strongest modality while diverging from how KTANE is naturally played. For instance, the Unicode workaround (Appendix G.8) observed for Keypad symbols is a text-play artefact that would not exist in spoken dialogue. Output is capped at 1,000 tokens per turn (Appendix C.7) and temperature is fixed at 0.6 across all models, since per-model tuning would conflate sampling configuration with capability.

**Evaluation.** The evaluation is pass@1: models are given one attempt at each of the ten multi-module missions, and for single-module missions, we report the average over ten missions per module type. This provides a strict but comprehensive evaluation of the capabilities of current models across diverse missions, rather than of specific task instantiations. We leave evaluating reliability across repeated attempts to future work. Results are point-in-time snapshots whose enduring value lies in the behavioural patterns they reveal.

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

## Appendix

Table A.1: Definitions of key properties used to characterise multi-agent environments.

| Property | Definition |
| --- | --- |
| Multi-Agent Environment | Environment with two or more agents, where they have to work either towards a shared goal (Cooperative), or against one another (Competitive). |
| Information Asymmetry | When different agents have access to different knowledge or observations about the environment or task, creating information imbalances that affect decision-making and interactions. |
| Asynchronous Action | The agents act asynchronously, taking independent actions in parallel. This contrasts with synchronous systems, where agents must wait for their turn in a predetermined order. |
| Dynamic Environment | The environment changes independently of agent actions. Here, we do not consider state changes resulting from the actions of another agent. |
| Real-Time Evaluation | The environment changes while the agent is thinking. For example, a countdown timer that continues while the agent deliberates. |
| Multi-Turn Communication | Use language to either describe the task or to coordinate the agents over multiple turns. |
| Image Sequence | Agent has to, at some point, deal with a stream of images. This might require integrating information across multiple images or understanding temporal differences. |
| Long Context | Agents need to retrieve information from long text and/or image inputs. |

# A  Extended Related Work

**General agentic benchmarks.**  The most foundational cluster of agentic benchmarks—AgentBench (Liu et al., 2024a), AgentBoard (Ma et al., 2024a), and VisualAgentBench (Liu et al., 2024b)—establishes a baseline by evaluating agents across a range of interactive tasks, using breadth to offer a comprehensive picture of how agents generalise across qualitatively different interactive settings. But breadth is not the same as depth of interaction complexity: these benchmarks are uniformly single-agent, operating in largely static environments, and place no demand on agents to coordinate, communicate, or reason under information asymmetry. The interaction model they assume—one agent, one task, one episode—is a reasonable starting point, but it systematically excludes properties that only emerge when agents must work alongside others in a world that does not wait for them.

**Video games as evaluation settings.**  Video games have long served as proving grounds for AI research. From the Atari Learning Environment catalysing breakthroughs in deep reinforcement learning (Bellemare et al., 2013; Mnih et al., 2015), to StarCraft II (Vinyals et al., 2019) and Dota 2 (OpenAI et al., 2019) exposing new frontiers in long-horizon strategic reasoning, games have consistently forced AI systems to confront temporal pressure, uncertainty, and emergent complexity. This tradition has naturally extended to MLLM evaluation through benchmarks spanning open-world exploration (MineDojo, Fan et al., 2022), roguelikes and puzzle games—such as BALROG (Paglieri et al., 2024), Baba Is AI (Cloos et al., 2024), COMMA (Ossowski et al., 2025), and Overcooked-AI (Carroll et al., 2019; Gessler et al., 2025)—as well as commercial titles (Zhang et al., 2025a), all providing the rich visual inputs, stochastic dynamics, and clear success criteria that general agentic benchmarks lack. A subtler concern is that several of these benchmarks use simplified or modified versions of their source games, which trade the complexity that made those environments compelling for more tractable problem spaces. A deeper structural issue also exists: all of these remain fundamentally single-agent settings. Even MindAgent (Gong et al., 2024), which introduces multi-agent coordination within a game environment, does so through a centralised planning architecture, thereby removing the information asymmetry and communication pressure that makes coordination difficult. The dynamism and visual richness that game environments bring to MLLM evaluation have not yet been placed in service of decentralised, cooperative, multi-agent interaction.

**Embodied environments.**  Embodied simulation environments represent the most direct attempt to address what game-based benchmarks leave unresolved: placing agents together in visually rich, interactive worlds and asking them to cooperate on extended tasks. TEACh (Padmakumar et al., 2022), Watch-and-Help (Puig et al., 2020), and the Alexa Arena (Gao et al., 2023) come closest to this—in all three, one agent holds knowledge of the task goal that the other must act upon, creating a structural information asymmetry that

makes natural language a necessary coordination channel. However, they all reveal the same limitation: the environment remains static between agent actions and does not stress agents across the long interaction traces that genuine cooperative tasks tend to produce—a dimension that matters given that MLLM performance is known to degrade towards the tail of long context windows (Wang et al., 2024a). HAZARD (Zhou et al., 2024) moves in a complementary direction by introducing natural environment dynamics—where the world evolves independently of what agents do—but does so in a single-agent setting, forfeiting the cooperative structure entirely. Li et al. (2024c) takes yet another path, trading visual grounding for richer interaction modelling and operating on text-based state representation that bypasses the visual reasoning demands that real environments present. A pattern emerges across this cluster: each benchmark makes progress on one dimension by accepting a regression in another, and no single environment refuses these tradeoffs simultaneously.

**Cooperative multi-agent settings with asymmetric information.** The benchmarks that take information asymmetry most seriously as a design principle represent the most deliberate attempt to force genuine communication — directly addressing what might be called the *knowing-doing gap* (Paglieri et al., 2024): in many multi-agent settings, agents can bypass language entirely and act autonomously, rendering communication ornamental rather than functional. Information asymmetry structurally closes this gap—when each agent holds knowledge that is necessary but insufficient for task completion, natural language becomes the *only* viable coordination channel. Competitive social deduction games such as Avalon (Light et al., 2023) and Werewolf (Wu et al., 2024) operationalise this best, with hidden roles creating a communicative pressure under which agents must reason strategically over what others know and believe. WhodunitBench (Xie et al., 2024a) extends this with visual grounding over evidential images, adding an absent multimodal layer to purely text-based deduction. But all three are fundamentally competitive—the communicative pressure they generate is adversarial, not cooperative, and the dynamics of strategic deception are quite different from those of collaborative problem-solving. COMMA (Ossowski et al., 2025), Multimodal CLEM (Hakimov et al., 2025), and Concordia (Smith et al., 2024) are all cooperative and preserve information asymmetry. COMMA and Multimodal CLEM both require visual understanding, though COMMA is the only one that requires visual grounding as part of the interaction. But even COMMA stops short of the full combination—agents receive static observations rather than image sequences across time, actions remain synchronised rather than asynchronous, and interaction traces stay well within standard context lengths. The pattern from the previous cluster holds: each benchmark isolates and advances a subset of the required properties, and the specific combination—cooperative, visually grounded, dynamic, asymmetric, asynchronous, and long-horizon—remains unoccupied.

**GPTNT positioning.** GPTNT occupies this unaddressed intersection by grounding evaluation in Keep Talking and Nobody Explodes, a cooperative game whose structure instantiates all of these properties simultaneously. Information asymmetry is inherent to the game's design; neither agent can succeed without the other, which closes the knowing-doing gap structurally as language is the *only* coordination mechanism. The bomb's countdown timer introduces natural environmental dynamics that persist regardless of agent deliberation, and the multi-module structure of each mission generates the extended interaction traces that stress long-context reasoning in ways that shorter episodes cannot.

# B   About *Keep Talking and Nobody Explodes*

Keep Talking and Nobody Explodes (KTANE) is a cooperative puzzle video game built around a fundamental information asymmetry: one player sees the bomb but has no instructions, while the other holds the manual but cannot see the bomb. Players must bridge this gap through communication, working against a countdown timer and a limited strike budget. Its combination of procedural puzzle generation, role separation, and real-time pressure makes it a rich environment for studying collaborative behaviour under constraint. This appendix describes the game's core elements as they are relevant to understanding the benchmark.

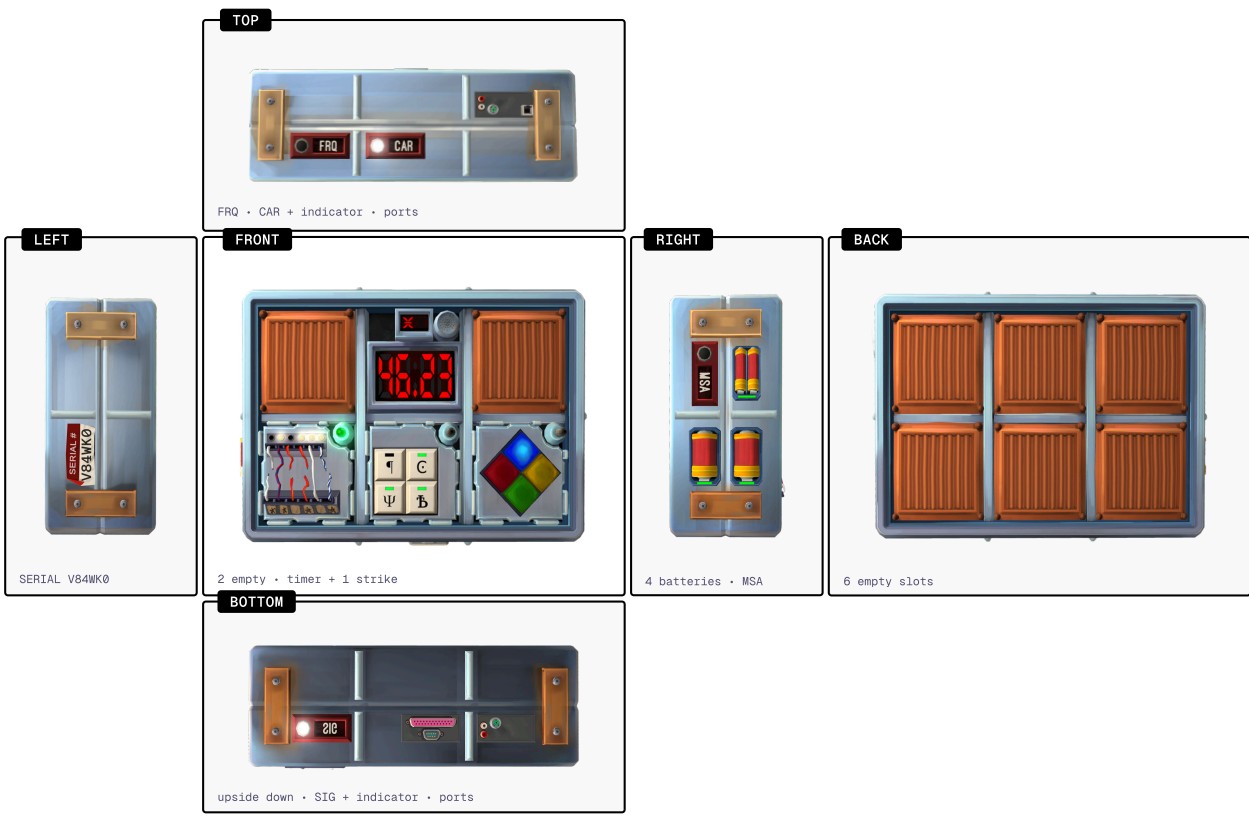

Figure B.1: An unfolded view of all six faces of the bomb, showing the timer, strikes, modules, and widgets. The Defuser observes one side at a time, navigating across sides to collect game-relevant information.

## B.1 Game Elements

**Manual.** The official manual—available at https://bombmanual.com—contains detailed instructions on how to solve each module, including illustrations to aid the Expert in their task as they cannot see the game.

**Bomb Layout.** As shown in Figure B.1, each bomb consists of several module slots on the larger front and back faces, one of which is designated for the *Timer*, while the remaining slots can be filled with one or more modules of different types. The sides contain widgets, such as the serial number, batteries, ports and light indicators, which provide contextual information needed to solve several modules.

**Time and Strikes.** The *Timer* shows the remaining time to defuse the bomb. As shown in Figure B.2, there is an LED display above the timer that shows the number of strikes that have occurred, indicating how many attempts the Defuser has left. Importantly, for each strike received, the timer speeds up.

**When is a module solved?** As shown in Figure B.3, there is an LED light located at the top-right corner of each module. When solved, it will glow green. If the Defuser makes a mistake while attempting to defuse the bomb, the LED on the corresponding module will *briefly* flash red to indicate that a strike is recorded, and a strike will permanently appear above the clock.

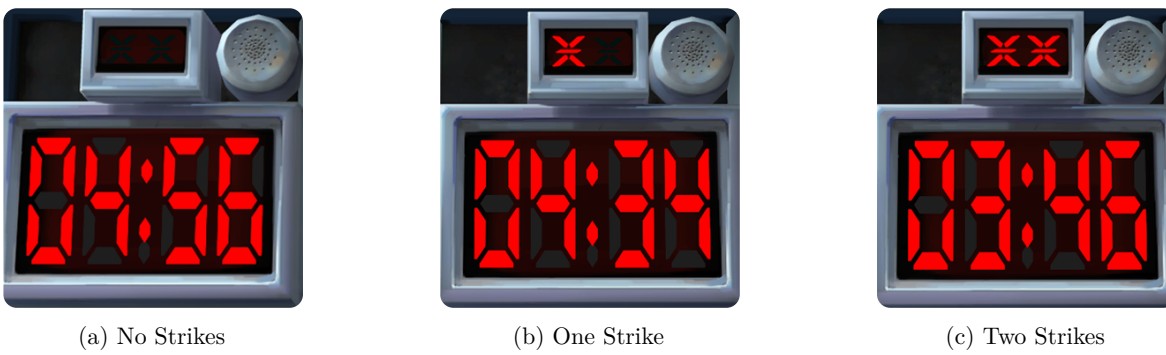

(a) No Strikes  (b) One Strike  (c) Two Strikes

Figure B.2: Strikes on the timer unambiguously show the Defuser the number of remaining attempts during the game.

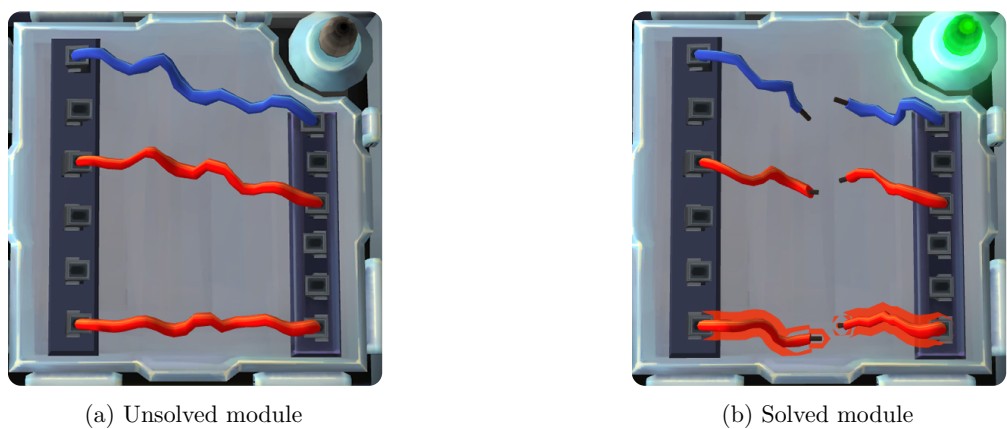

(a) Unsolved module  (b) Solved module

Figure B.3: The LED light in the top right of each module glows green when the module is defused.

## B.2 Individual Modules

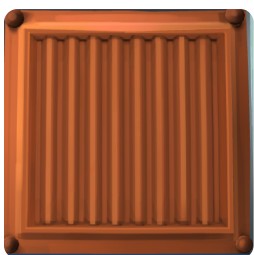

**Empty**
**Stages:** 0
**Difficulty:** N/A—Requires zero player input; difficulty is purely visual.

The Empty module is not a puzzle for the bomb to solve. It is used as a placeholder because a given module slot might not contain any modules. For all intents and purposes, this option is considered during the sampling; however, it requires no input from the player and can be considered as "automatically solved".

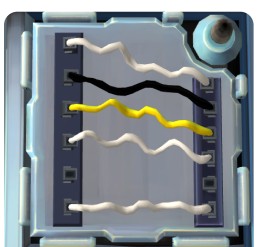

**Wires**
**Widgets Needed:** Serial Number
**Stages:** 1
**Difficulty:** Relies on identifying easily communicated details (colour, count, order) with straightforward instructions.

For the Wires module, the Defuser needs to cut the correct wire. The correct wire depends on several key variables, including the total number of wires, each colour and specific position, the frequency of each colour, and the serial number of the bomb.

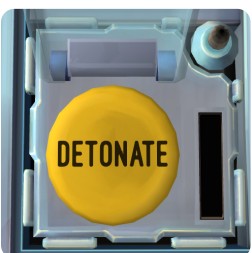

### ⊟ *Button*
**Widgets Needed:** Indicators, Batteries
**Stages:** 1
**Difficulty:** Adds a layer of temporal difficulty; the Defuser might need to wait for the right moment to solve the module.

The Button module is a large button with a label and an LED strip. The Defuser is expected to press and release according to various factors including the colour, word, widget information, and the colour of the LED strip when the button is held down.

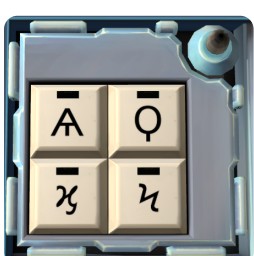

### ▦ *Keypad*
**Widgets Needed:** None
**Stages:** 1
**Difficulty:** Requires building a "shared understanding" of abstract symbols that are not common knowledge.

Contains four buttons, each with a symbol. To solve, the Defuser must press them in a specific order based on manual lists. The primary challenge is describing symbols accurately between players.

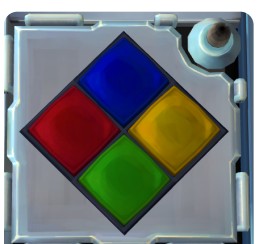

### ◈ *Simon Says*
**Widgets Needed:** Serial Number, Strike Count
**Stages:** 5
**Difficulty:** Complexity is derived from state tracking.

Consists of four coloured buttons, one of which flashes. Players must gather details about the flashing colour and current strike count to follow the correct button sequence.

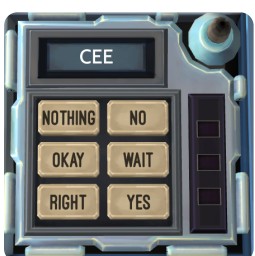

### ▦ *Who's On First*
**Needs Side Info:** False
**Stages:** 3
**Difficulty:** Relies on phonetic confusion (homophones like "there/their/they're"). This is harder for human vocalisation than text.

The Expert must identify labels based on displayed words and then find the first word in a list that appears on the buttons.

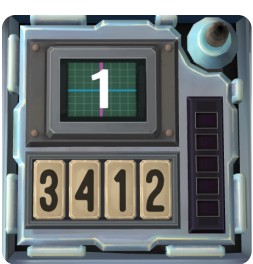

### ▦ *Memory*
**Needs Side Info:** False
**Stages:** 5
**Difficulty:** Requires high cognitive load to maintain an "intrinsic state" across five panels with shuffled positions.

After each correct press, buttons shuffle. The Expert must remember both the numbers and their corresponding positions from every previous stage to proceed.

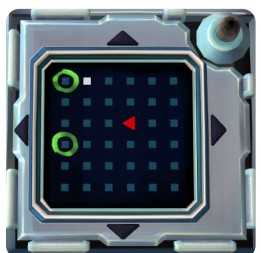 ### *Maze*
**Needs Side Info:** False
**Stages:** 1
**Difficulty:** Requires spatial reasoning and the ability for players to maintain an intrinsic state of a $6 \times 6$ coordinate map.

The Expert must identify the correct map based on two green points and guide the Defuser to a red triangle without crossing hidden walls.

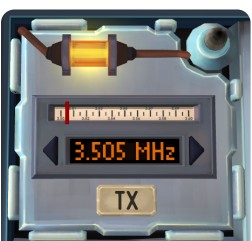 ### *Morse Code*
**Needs Side Info:** False
**Stages:** 1
**Difficulty:** Requires the Defuser to accurately interpret and communicate a sequence of timing-based blinks.

The Expert must decode the Morse code to identify a word and transmit the frequency. It is difficult due to the high risk of communication error regarding the blinks.

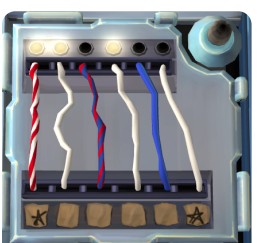 ### *Complicated Wires*
**Needs Side Info:** Serial, Batteries, Parallel Port
**Stages:** 1
**Difficulty:** Visually complex Venn diagram instructions increase the Expert's cognitive burden compared to other tasks.

Requires determining whether to cut wires based on LEDs, star symbols, and bomb-wide state. The Expert's task is much more difficult than the Defuser's.

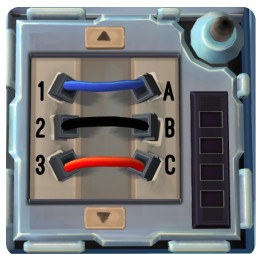 ### *Wire Sequence*
**Needs Side Info:** False
**Stages:** 4
**Difficulty:** Players must track a cumulative count of wire colours across multiple sequential panels without losing track.

The Expert determines whether to cut based on colour and its cumulative appearance count. Tracking the count is the primary source of difficulty.

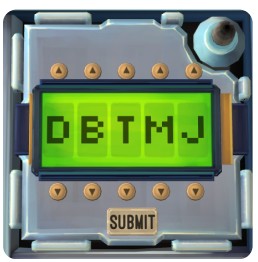 ### *Passwords*
**Needs Side Info:** False
**Stages:** 1
**Difficulty:** High potential for "search space" error; players must cycle through multiple letters to find a matching word.

Each slot contains six possible letters. The Defuser must cycle through each slot to form a correct word found in the manual.

### B.3 Multi-Module Missions

KTANE missions get progressively harder as players play the game. For our multi-module missions, we sample modules and set time limits according to the rules for levels 2.1–3.3 in the wiki.[24] Table B.1 maps our missions to the corresponding level, and highlights any deliberate deviations from the rules. Across missions, module types are chosen from pools of like-difficulty, grouped as follows:

- **Easier:** *Wires*, *Button*, *Keypad* (introduced in Level 1)

- **Intermediate:** *Simon Says*, *Who's On First*, *Memory*, *Maze* (introduced in Level 2)

- **Challenging:** *Morse Code*, *Complicated Wires*, *Wire Sequence*, *Passwords* (introduced in Level 3)

Table B.1: Mapping between multi-module missions and corresponding KTANE mission levels, with deliberate deviations indicated in **bold**.

| # | Modules | Level | Description (**bold**=deliberate deviations) | $t(s)$ |
|---|---------|-------|-----------------------------------------------|--------|
| 1 | *Wires* *Keypad* *Maze* | 2.1 | 3 modules
2× *Keypad*/ *Button*/ *Wires*
1× *Maze*/ *Simon Says*/ *Memory* | 300 |
| 2 | *Wires* *Button* *Keypad* *Memory* | 2.1 | **4** modules (default: 3)
**3**× *Keypad*/ *Button*/ *Wires* (default: 2×)
1× *Maze*/ *Simon Says*/ *Memory* | 300 |
| 3 | *Wires* *Keypad* *Simon Says* *Who's On First* | 2.3 | 4 modules
2× *Keypad*/ *Button*/ *Wires*
1× *Maze*/ *Simon Says*
1× *Memory*/ *Who's On First* | 300 |
| 4 | *Keypad* *Simon Says* *Who's On First* | 2.4 | | 180 |
| 5 | *Wires* *Memory* *Maze* | 2.4 | 3 modules
1× *Keypad*/ *Button*/ *Wires*
2× *Maze*/ *Simon Says*/ *Memory*/ *Who's On First* | 180 |
| 6 | *Button* *Who's On First* *Memory* | 2.4 | | 180 |
| 7 | *Who's On First* *Memory* *Passwords* | 3.1 | 3 modules
1× *Morse Code*/ *Passwords*
**2**× *Simon Says*/ *Who's On First*/ *Memory* (default: 1×)
**0**× *Button*/ *Wires* (default: 1×) | 300 |
| 8 | *Button* *Simon Says* *Complicated Wires* | 3.2 | 3 modules
1× *Button* (default: *Wires*)
1× *Complicated Wires*/ *Wire Sequence*
1× *Simon Says*/ *Memory* | 300 |
| 9 | *Wires* *Who's On First* *Morse Code* *Complicated Wires* | 3.3 | 4 modules
1× *Wires* (default: *Keypad*/ *Button*)
1× *Maze*/ *Simon Says*/ *Memory*/ *Who's On First*
2× *Morse Code*/ *Passwords*/ *Complicated Wires*/ *Wire Sequence* | 300 |
| 10 | *Keypad* *Wire Sequence* *Maze* *Passwords* | 3.3 | 4 modules
1× *Keypad*/ *Button*
1× *Maze*/ *Simon Says*/ *Memory*/ *Who's On First*
2× *Morse Code*/ *Passwords*/ *Complicated Wires*/ *Wire Sequence* | 300 |

---

[24]https://ktane.fandom.com/wiki/Missions

### B.4 Handling Time

#### B.4.1 Determining the Synchronous Action Interval

In-game animations in KTANE take up to 3 seconds to complete after an action. For example, 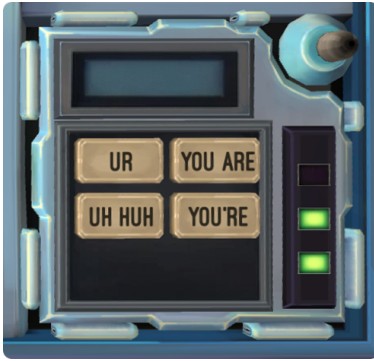 *Who's On First* introduces a delay between clicking a module element and the next stage becoming fully observable: pausing mid-animation produces an incomplete observation (see Figure B.4) that a model cannot reliably act on, and recovering from one costs a further turn spent re-observing. Therefore, capturing observations earlier does not save game time; it merely converts a fixed cost into a stochastic one, which falls most heavily on multi-stage modules with inter-stage animations.

In the *synchronous* setting, we resume the game for 3 seconds after every action, ensuring all state transitions are fully resolved before the next observation is captured. The pause takes effect at the next engine frame, so the realised window falls between 3 and 3.5 seconds. This is a property of the environment and independent of the model under evaluation; we refer to the nominal 3-second window throughout. In the *asynchronous* setting, no such buffer is needed: the round-trip time of a model forward pass often exceeds 3 seconds, so animations resolve naturally. If future models become fast enough to act mid-animation, the resulting incomplete observations become part of the challenge—mirroring how a human player must cope with acting faster than the game can update.

Figure B.4: Example of *Who's On First* while the inter-stage animation is in progress

#### B.4.2 Calculating Time Limits for Single-Module Missions

For single-module missions, which are not from KTANE itself, we compute the time limit heuristically based on the estimated maximum number of turns required to complete each module. The total count per mission is determined systematically by aggregating the necessary direct module interactions, bomb rotations, communication steps, and miscellaneous overhead. To calculate the time limit in seconds, we multiply the estimated maximum number of turns by the action interval (3 sec). We compute this dynamically for each mission, as outlined by Table B.2. Table B.3 demonstrates the computed turn allowance for each module, assuming a minimum of one stage update and standard front-facing placement.

Table B.2: Turn cost factors used to compute the total time limit per mission.

| Factor | Turns | Description |
|---|---|---|
| Module interactions | $\sum$ max actions | Total interaction steps per module; per-module values given in Table B.3 |
| Stage updates | $\sum$ stages | One turn per stage update per module |
| Bomb rotations | $R \times 8$ | $R$ is the total side checks required across all modules, clamped to $[1, 4]$ |
| Back placement | 8 | Added only when back placement is permitted |
| Zooming | $2(M + R)$ | One zoom in and out per module $M$ and per rotation $R$ |
| Strike buffer | 3 | Fixed allowance for up to 3 strikes |
| Dialogue allowance | 10 | Fixed baseline for communication overhead |
| **Total (seconds)** | **Sum $\times$ 3** | 3 seconds per action |

Table B.3: Breakdown of per-module time limits. Side info denotes the number of widget observations the Defuser must convey; Stages corresponds to the number of sequential phases of a module; Max actions is the largest number of interactions a single solution can require; Total turns aggregates the maximum number of actions and messages.

| Module type | Side info | Stages | Max actions | Total turns | Time limit (s) |
|---|---|---|---|---|---|
| Wires | 1 | 1 | 1 | 25 | 75 |
| Button | 2 | 1 | 14 | 48 | 144 |
| Keypad | 0 | 1 | 4 | 18 | 54 |
| Simon Says | 1 | 5 | 15 | 43 | 129 |
| Who's On First | 0 | 3 | 3 | 19 | 57 |
| Memory | 0 | 5 | 5 | 23 | 69 |
| Maze | 0 | 1 | 35 | 49 | 147 |
| Morse Code | 0 | 1 | 22 | 36 | 108 |
| Complicated Wires | 2 | 1 | 6 | 40 | 120 |
| Wire Sequence | 0 | 4 | 16 | 33 | 99 |
| Passwords | 0 | 1 | 51 | 65 | 195 |

## C   The GPTNT Framework

GPTNT is a purpose-built microservice framework providing LLMs with a complete programmatic interface to *Keep Talking and Nobody Explodes.* KTANE offers no native API, no visual interface hook, and no existing tooling for programmatic control; therefore, all infrastructure is developed from scratch. It runs inside the Game Instance and exposes a local HTTP server through which all of its capabilities are available to the rest of the framework. The mod provides five capabilities: (1) configuring and starting missions; (2) executing player actions; (3) exporting rendered frames; (4) exporting segmentation masks of interactable elements; and (5) exporting game state and controlling the game lifecycle. The remainder of this section describes each capability, together with the deliberate deviations we make from the stock game.

### C.1   The Mod

We create a mod that adds an HTTP server to KTANE to programmatically control KTANE itself.

#### C.1.1   Starting Missions

Using the mod, we can start missions programmatically using HTTP requests. With this, we can configure various parameters to define the mission, including the time limit, number of strikes, total number of modules, specific module selection, random seed, and the number of widgets. KTANE generates all module states deterministically from the integer seed, so a mission configuration fully and reproducibly specifies the bomb; we describe how seeds are selected for the benchmark in Appendix D.1.

#### C.1.2   Actions

As previously mentioned, types of actions sent to the game are clicking, holding, releasing, zooming out, and rotating the bomb. We achieve rotations by simply editing the transform of the bomb game object. When a player interacts with a module the game zooms into it and activates its child interactables, for example, interacting with the wires module zooms into it and activates the separate wires making them interactable. The zoom out action works the opposite way and is only usable when zoomed into a module. For clicking and holding the mod expects a relative coordinate originating from the top left corner. Then a ray is cast from the specified point, onto the bomb and interacts with the first active interactable.

**Relative coordinates for interacting.** Coordinates are specified in a resolution-independent relative system in which $(0,0)$ denotes the top-left corner and $(1,1)$ for the bottom-right. Therefore, regardless of model and resolution, or any future interaction method, the action representation does not vary with perceptual configuration.

### C.1.3   Changes from the original game

Throughout the benchmark, we prioritise running AI agents in an environment that is identical to the original game. However—in very specific cases—we need to depart from the original game.

**Changing the initial delay for ⬨ *Simon Says*.** The ⬨ *Simon Says* module works by having a looping sequence of beeps, the time between each beep is not changed but the time between the start of sequences is decreased from 5 to 1 second. When a player correctly clicks the next beep, the game originally waits 2 seconds to loop again, this is also changed to 1 second. The sequence is looped from the start whenever a player zooms into the module or gets a strike. Finally, when a player clicks on one of the simon buttons the time delay to the next sequence is reset to the original 5 seconds.

**Always holding the bomb.** We always start missions with the bomb already picked up, and we don't allow models to accidentally put it down. From preliminary experiments, we find that models sometimes "zoom out" too many times, and they struggle to re-select the bomb, so we just remove this control from the game since it is unnecessary and unrelated to the purpose of the benchmark.

### C.1.4   Exporting Frames

As part of the observations taken from the game, the original images contain a cropped version of what a human player would see when playing the game. This is achieved by having a duplicate Unity camera of the original game camera. This camera renders to a render texture which is then converted into PNG encoded bytes.

An image is added to a ring buffer every 0.25 seconds with a maximum length of 16 frames. We use a buffer of frames to capture visual information over a short period of time required by the ⬨ *Simon Says* and ▤ *Morse Code* modules. The 0.25-second interval between each frame is chosen based on the length of a time unit for a blink of the morse code module. These two modules are the only two modules requiring a longer sequence of frames, while other modules only need the most recent frame.

### C.1.5   Segmentation Masks

The last part of the observations from the game includes a segmentation mask of the interactable objects, resulting in a black image with a unique colour for each interactable game object currently displayed on the scene. A similar approach to the frame buffer is used here, where a duplicate camera renders to a render texture. This camera, however, only renders objects on a custom segmentation layer with a visual shader to handle the colours. Interactable objects in the scene are first moved to the segmentation layer and assigned a unique colour. The visual of the objects is then captured using the dedicated camera to be converted into PNG encoded bytes. Finally, the objects are returned to their original layer. We discuss this further in Appendix C.4.2.

### C.1.6   Exporting Game State

**Bomb state.** On request, the mod exports a structured JSON snapshot of the complete bomb state: the countdown timer, strike count, widgets, and, for every module, both its puzzle configuration (e.g., wire colours and positions, button labels, keypad symbols, maze walls) and its runtime progress (e.g., interaction and solution progress). This snapshot is never provided to models—they must solve the task from visual observations alone (Appendix C.4). Instead, the framework uses it for logging and outcome detection, for the seed-diversity protocol (Appendix D.1), and for procedurally generating the VQA datasets (Appendix D.3).

**Game lifecycle.** The mod also reports the game's lifecycle state, which is used to drive the game loop (Appendix C). A game moves through a "lights-off" phase, in which the bomb is present in the scene but not yet visible or active and the timer has not started; a "lights-on" phase, in which the bomb is visible, all

modules are active, and the timer runs; and a terminal state (all modules solved, exploded from three strikes, or exploded from timer expiry). In addition, the mod can pause and resume the game at any point by setting Unity's internal time scale to zero, freezing the bomb timer and all animations in place. The double-pause startup protocol and the synchronous mode are both built on these two primitives.

## C.2 Process Architecture

**Running the game.** The *Game Instance* is the Unity game process, running a custom-built mod that exposes an HTTP server through which the game can be controlled and queried. The *Game Service* is a lightweight Python-side mediator that sits between the game's HTTP server and all other components; it tracks game state, monitors process aliveness, and presents a clean abstracted interface to the rest of the system. KTANE can be run on Windows, Mac OS, and Linux. All the experiments in this paper are run on Linux machines running Ubuntu. Importantly, KTANE requires a graphics card that supports displays; while it can run on a CPU, it will likely lag, resulting in inconsistent results. We run all experiments using X-Server on RTX 2080Ti's.

**Running agents.** The *Player Service* is an independent LLM agent process, one per active player. In two-player collaborative conditions, both a Defuser Player Service and an Expert Player Service are instantiated; in solo conditions, where a single model holds both roles simultaneously, only one Player Service is present. Within these, we leverage Pydantic AI[25] (v. 1.67.0) to perform all requests with models through their API; open-source models use vLLM (Kwon et al., 2023) to serve an OpenAI-compatible server which can be queried through Pydantic AI.

**Communication.** All inter-process communication is handled over Redis channels using remote procedure call (RPC) commands, and services communicate with one another directly over these channels. The *Experiment Manager* (EM) is a central coordinator rather than a message broker: it owns the game lifecycle—experiment startup, readiness verification, the lights-on transition, and pausing/un-pausing—and enforces turn-taking in the synchronous mode (Appendix C.3.3). No service holds a global view of a game while it runs: each Player Service maintains its own state and conversation history locally and exports it when the game ends, and the per-service exports are merged post hoc into a single game record for analysis.

**Information asymmetry.** Information asymmetry between Defuser and Expert is enforced architecturally. The Expert's Player Service has no connection to the Game Service at any point in execution: it cannot receive game state, observe the bomb, or retrieve any information about the current game by construction of the network topology. Leakage from game to Expert is structurally impossible regardless of model behaviour.

**Parallel execution.** GPTNT supports simultaneous execution of multiple independent games on a single machine. Each game has its own Game Instance and Game Service, and all are coordinated by a shared Experiment Manager. This is a practical requirement for conducting experiments at the scale reported in this paper.

## C.3 The Game Loop

### C.3.1 Startup and Synchronisation

Each game begins with the Unity process loading the bomb configuration and entering a "lights-off" phase (Figure C.1a), in which the bomb is present in the scene but not yet visible or active. The EM immediately pauses the game at this point and initiates readiness checks: every active Player Service must confirm that it is connected, correctly configured, and ready to proceed before the game is permitted to advance. Once all players have confirmed readiness, the EM triggers the transition to the "lights-on" phase, in which the bomb becomes visible and all modules become active, and immediately pauses the game again. This double-pause protocol guarantees that the bomb timer has not started when play begins, and that all players are in a confirmed-ready state at the moment of first un-pause. The lights-on pause is the final synchronisation point in the startup sequence. From this point, execution diverges by mode.

---

[25]https://ai.pydantic.dev/

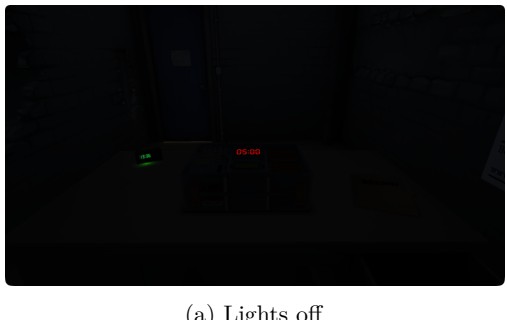

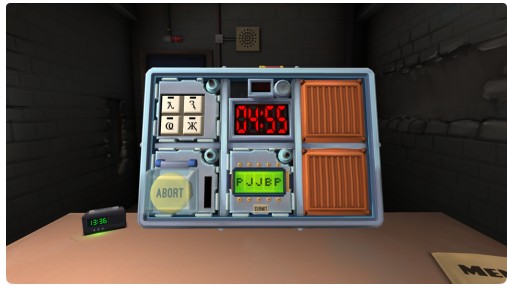

(a) Lights off

(b) Lights on

Figure C.1: Example from KTANE when the lights are off before the mission, and then on when the mission starts.

### C.3.2 Asynchronous Mode

Asynchronous execution is the primary mode for all multi-module experiments and collaborative conditions. Its defining characteristic is that the game timer runs continuously and is never paused. As a result, all model inference time counts against the bomb clock.

At lights-on, the game resumes and two independent polling loops—one for the Defuser, one for the Expert—are launched simultaneously, and continue until the game is over. Each loop executes the same cycle: pull new observations and messages, produce a forward pass, sleep for 0.5 seconds, and repeat. The two loops are entirely independent; neither waits for the other, and there is no shared synchronisation primitive between them. This is the precise sense in which the mode is asynchronous. If the Defuser's API call takes ten seconds, ten seconds elapse on the bomb clock; in that same interval, the Expert may have completed three or four turns. Therefore, the total generation time from models is a first-class performance variable in this mode.

**Message queue.** Communication between players is mediated by a pull-based message queue maintained independently per player. On every forward pass, a player pulls all messages dispatched to it by its partner since its last turn, draining the queue entirely before generating a response. Messages are never dropped: if the Expert sends four messages during a long Defuser generation, all four arrive together on the Defuser's next forward pass. This design imposes no artificial turn structure on the conversation while guaranteeing that no information is ever lost in transit.

**Streaming.** Current MLLMs (and their APIs) do not support receiving new input while a generation is in progress. Therefore, we do not use streaming outputs either. In any case, streaming would not be useful: the receiving player requires the partner's complete output before it can form a coherent response, meaning streaming would not reduce effective communication latency under the present architecture.

### C.3.3 Synchronous Mode

Synchronous mode shares the startup sequence and message semantics of asynchronous mode but differs in two ways: the game is paused while models generate, and the two players act in strict alternation rather than in free-running loops. Clock time advances by a fixed budget per Defuser turn rather than in real time, so model generation time has no bearing on effective game time.

At lights-on, the game remains paused and the Defuser takes the first turn. Each cycle then proceeds as follows:

1. **Observe.** The Defuser pulls its observations and drains its message queue. Because the game is paused, observations are captured from a frozen, fully resolved game state.

2. **Generate.** The Defuser produces a forward pass. Unity's internal time scale is set to zero during generation, so the model may take unbounded wall-clock time at no cost in game time.

3. **Act.** The action is dispatched to the game, which is resumed for a window of at least 3 s of game time; in practice the realised window is 3 s to 3.5 s (Appendix B.4.1). The action executes and all resulting animations and state transitions resolve before the game is paused again. The bomb clock therefore advances for each Defuser turn, regardless of the type of action produced by the Defuser.

4. **Expert turn.** While the game is paused, the Expert pulls its message queue, produces a forward pass, and dispatches its action. Expert turns consume no game time.

The cycle repeats until the game ends. Because each Defuser turn consumes 3 s, a mission's time limit translates into a maximum number of Defuser turns—the time limit divided by three (Appendix B.4).

## C.4 Observations

While we do export a JSON representation of the environment, we never provide this to models because this would make the task too easy. Models must use the modalities available to them—just like a human player—to solve the task. We strongly recommend against adapting the benchmark to provide this information to models because the complexity stemming from the visual observations would be muted, resulting in a drastically simplified task.

### C.4.1 Image Resolution

The game modification exports raw RGB24 pixel data from the Unity render buffer on demand. The default capture resolution is $640 \times 480$ (4:3 aspect ratio), selected as the minimum resolution at which a human player can reliably read all text, labels, and visual indicators present in the game. This serves as the baseline for all models for which context length permits its use.

Two models—Qwen and InternVL—require downsampled resolutions due to context window constraints. At $640 \times 480$, the combined token cost of the Expert's manual and the Defuser's game frames would saturate these models' context windows during tokenisation, leaving insufficient capacity for dialogue history. Qwen therefore receives frames at $448 \times 448$ and InternVL at $504 \times 504$.[26] All downsampling uses Lanczos filtering. All reduced resolutions are manually verified to remain legible for every module type appearing in the experiments.

Lower resolution reduces token cost per frame and enables longer retained dialogue histories; higher resolution would improve legibility of fine-grained visual elements—digit labels, wire colours, indicator text—but is cost-prohibitive at the experimental scale reported here.

### C.4.2 Set-of-Marks

Observations given to the Defuser agent include set-of-marks (SoM) annotations (Koh et al., 2024a; Yang et al., 2023), which are used to highlight elements in the current view that are interactable by the agent. As illustrated by Figure C.2, each interactable element is outlined by a brightly coloured outline, labelled by a label within a box of the same colour. Within any given observation, each interactable element will have a unique label, but labels are not unique across different observations.

**Label.** Letters are used for annotations rather than numbers. As recommended by Yang et al. (2023) and from preliminary experiments, we find that using numbers leads to mistakes and hallucinations. We attribute this to the presence of numbers on many of the modules, as shown by the 🖼 *Memory* module in Figure C.2. Additionally, while the Password module uses letters, using letters for the labels has no noticeable effect on the ability of the model to recognise the text in the display versus the labels. We attribute this to the fact that the letters in the Password module are placed and look distinctly different from the marks.

**Position.** As each bomb module has the same layout, we use heuristics to place the labels such that for any given module, the labels will always be in the same position, relative to their module. Additionally, we ensure that if the number of elements within a module varies, the positioning of labels remains consistent.

---

[26]While Qwen has a larger context length and can support the original resolution of observations, we want to make the open-source models comparable, and so we provide images according to the base resolution supported by either model.

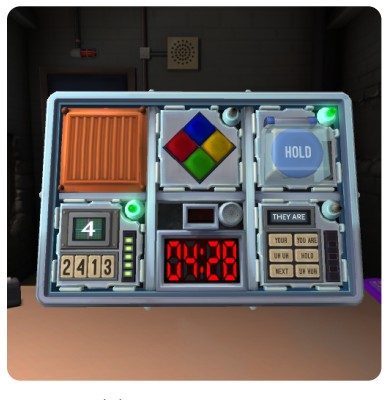 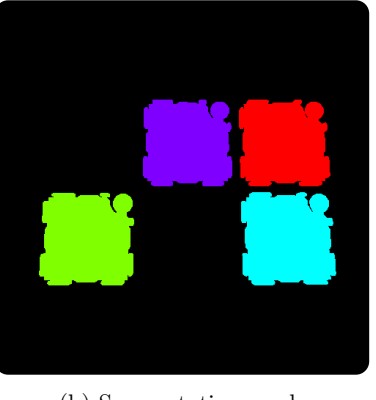 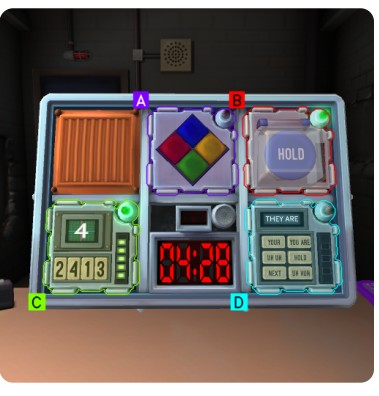

(a) Original frame      (b) Segmentation mask      (c) Image with annotations

Figure C.2: Set-of-Marks annotation inputs and outputs. The original frame (a) and segmentation mask (b) are both provided by the environment using our mod. In the segmentation mask, each interactable object is represented by a distinct colour region. Using a heuristic pipeline, we generate the annotated image (c) which combines the previous two images and provide that to the model to facilitate easier identification of interactable elements.

For example, as illustrated in Figure C.3a, all labels for the Wires module will appear on the left, ordered from top-to-bottom. From preliminary experiments, we find that ordering all labels in a common reading order leads to the most consistent behaviour from models.

**Colour.** As mentioned in Appendix B.1, some modules rely on colours for their references, meaning that randomly assigning colours is not effective for some modules. This includes the ▣ *Button*, ◈ *Simon Says* and each of the three wire-based modules—▤ *Wires*, ▦ *Complicated Wires*, and ⊠ *Wire Sequence*. In these cases, we match the colour of the mask and label background to the colour of the interactable element. For example, in ◈ *Simon Says* (Figure C.3b), the colour for the set-of-marks matches each respective button colour. This same pattern follows in each wire-based module except for ▦ *Complicated Wires*: as a wire can contain multiple colours, a multi-coloured mask is used in this case, striped with each colour which appears in the wire. An example of this can be observed in Figure C.3c.

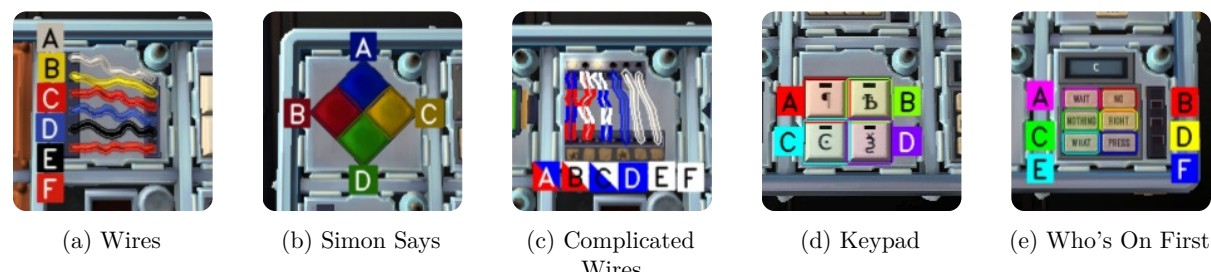

(a) Wires      (b) Simon Says      (c) Complicated Wires      (d) Keypad      (e) Who's On First

Figure C.3: Examples of set-of-marks annotations applied to different bomb modules where colour is important to distinguish between elements.

### C.4.3 Temporal Observations

For most modules—where reading the current state means nothing more than a single snapshot—only the most recent frame (and preceding frame) is passed to the Defuser on each forward pass. However, modules like ▦ *Morse Code* and ◈ *Simon Says* are exceptions: both encode information in blinking light sequences that unfold over time and simply cannot be decoded from any individual frame, so for these the full sixteen-frame buffer from the mod (Appendix C.1.4) is transmitted to models. That depth of sixteen is empirically verified

to contain at least one complete blink cycle for both modules—for Morse, one cycle corresponds to the longest letter. Transmitting full video sequences is considered as an alternative but rejected on token cost grounds.

Despite our best efforts, models can occasionally miss a complete sequence within a single window, usually because the observation lands poorly timed relative to the blink period. The correct response is to emit a `do_nothing` action and re-observe on the following turn, which mirrors what human players do under the same circumstances.

### C.4.4 Observation Context Management

Accumulating visual observations across every turn causes the context to grow in direct proportion to game duration, and for models with smaller context windows this becomes a genuine problem — unconstrained image accumulation would rapidly exhaust available capacity, eventually displacing both the dialogue history and the manual. To handle this, each forward pass receives only the current turn's observation, one frame for standard modules or sixteen for 🖩 *Morse Code* and ◈ *Simon Says*, along with a single unannotated frame from the immediately preceding turn, with everything older discarded.

That previous frame turns out to matter quite a bit. Without it there's no reliable way to tell whether the last action produces any visible change—a correctly placed click changes the game state in ways that are easy to see, while a mis-click on a neutral region leaves everything exactly as it is. Keeping that one prior frame makes the consequence of the preceding action directly observable, at the cost of a single additional image per turn.

### C.5 Providing the Manual

The official manual—available at https://bombmanual.com—contains detailed instructions on how to solve each module, including illustrations to aid the Expert in their task as they cannot observe the game directly.

The manual is provided to the Expert as interleaved text and images—one page of extracted text followed by the corresponding page image, repeated for every relevant page. The full manual runs to 23 pages, though irrelevant ones are discarded, leaving roughly 16 module pages plus appendix content. Each page is A4 (equivalent to $2{,}550 \times 3{,}300$px), and therefore downscaled to match the same resolution as the visual observation frames the model receives during play (Appendix C.4). Importantly, we preserve the orientation of the pages, ensuring pages remain in portrait orientation.

Because system prompts cannot contain images, we instead provide the manual in the Expert's first incoming message, where it is protected from context truncation regardless of how long the dialogue runs. Unfortunately, this introduces a semantic oddity as a practical side effect: the Defuser, who by definition cannot see the manual, appears to be the one delivering it. We address this directly in the system prompt, and across all our recorded games, models consistently treat the Defuser as unable to consult the manual in practice (e.g., Figure G.17).

### C.6 Action Space

Every model output must contain a single action expressed as a JSON object, wrapped in an `<action>` XML tag. Within this, all actions share the same JSON structure:

```
{"result": {"kind": "...", "data": {...}}}
```

The `kind` field identifies the action category, and the `data` field carries the kind-specific arguments. We define three top-level kinds:

1. `interact_game`: all manipulation of game elements

2. `send_message`: send a text string to the other player

3. `do_nothing`: choose to skip their turn and do nothing

The schema is both role- and condition-specific. The Defuser receives a schema covering all three kinds. The Expert receives a strict subset that does *not* include `interact_game`. Additionally, `send_message` is only included when a partner is present in the game—e.g., when models must use parametric knowledge (§10.2) or perform without the Expert (§10.1)—to prevent models from dispatching outputs to non-existent recipients.

### C.6.1 Do Nothing.

Both the Defuser and Expert can provide a `do_nothing` action on any turn:

```
{"result": {"kind": "do_nothing", "data": {}}}
```

This is a first-class schema action carrying no penalty. In using it, the model signals that it has decided that acting would be premature, unnecessary, or counterproductive given the current game state.

**Waiting for the other player.** In asynchronous mode the game loop does not pause between turns, so a player will often complete a forward pass while the other is mid-sequence. Rather than forcing noisy output, either player can simply wait—the Expert when the Defuser is executing a sequence requiring no input, and the Defuser when awaiting further guidance from the Expert.

**Holding the button.** `do_nothing` is the *only* valid action during a `hold` sequence in the ☺ *Button* module. As the Defuser must output `release` once a specific target digit appears on the countdown timer strip, they must wait for however many turns are required.

**Observing timed blink sequences.** For ☷ *Morse Code* and ◈ *Simon Says*, the full 16-frame buffer is transmitted each forward pass. Despite our best attempts, if the observation window is misaligned with the blink period, the buffer may not contain a complete sequence. The correct response is to emit `do_nothing` and re-observe on the following turn—mirroring the strategy a human player would use under the same conditions.

### C.6.2 Send Message

The `send_message` action sends a free-form text string to the other player and is the sole method through which Defuser and Expert can communicate:

```
{
  "result": {
    "kind": "send_message",
    "data": {
      "message": "Six wires: red, red, blue, yellow, black, white. Serial number ends in
      ↪  an odd digit."
    }
  }
}
```

Messages enter the recipient's pull queue immediately and are delivered atomically at the start of their next forward pass. No message is ever dropped, summarised, or withheld—if the Defuser completes three turns during a long Expert generation, all three messages arrive together on the Expert's next turn.

Message content is also unconstrained in format and length, thereby allowing models to send full observations, partial updates, questions, or multi-step instructions across consecutive turns. They are only limited by the 1,000-token limit per model output.

### C.6.3 Navigation Actions

Navigation actions are issued under the `interact_game` kind and solely move the bomb to show different perspectives. Therefore, they require no location argument.

```
{
  "result": {
    "kind": "interact_game",
    "data": {
      "action": "rotate_left"
    }
  }
}
```

In Table C.1, we outline the various navigation actions and how the bomb is rotated. Note that models must explicitly choose to zoom out to then be able to zoom into another module. However, if a model performs a navigation action while zoomed into a module, we automatically zoom out and then perform that action.

Table C.1: Navigation sub-actions available to the Defuser under `interact_game`.

| Sub-action | Effect | Face exposed |
|---|---|---|
| `rotate_left` | Rotate the bomb 90° counter-clockwise (yaw) | Adjacent side face |
| `rotate_right` | Rotate the bomb 90° clockwise (yaw) | Adjacent side face |
| `flip` | Rotate the bomb 180° | Opposite side face |
| `roll_up` | Roll the bomb upward 90° | Bottom face |
| `roll_down` | Roll the bomb downward 90° | Top face |
| `zoom_out` | Exit the current module zoom | Current face (un-zoomed) |

Table C.2: Interaction actions available to the Defuser with `interact_game`.

| Sub-action | Location required | Description |
|---|---|---|
| `click_release` | Yes | Press and immediately release; standard for most elements |
| `hold` | Yes | Initiate a sustained press at the specified location |
| `release` | No | Terminate an active `hold` |

### C.6.4 Interaction Actions

Interaction actions (Table C.2) operate on a specific screen location and require a `location` argument. The GPTNT benchmark supports multiple coordinate systems to accommodate different models; all are normalised to relative coordinates internally before being dispatched to the game.

**Coordinate systems.** Three systems are supported:

- **Absolute:** integer pixel values in $[0, x_{\max}] \times [0, y_{\max}]$, with $(0, 0)$ at the top-left corner.

- **Scaled:** integer values in $[0, 1000]$ on each axis, independent of render resolution.

- **Relative:** floating-point values in $[0, 1]$ on each axis.

**Set-of-Marks.** For models that do not support coordinate output, elements can be targeted using SoM labels. The system resolves a label to the centroid of the corresponding element's bounding region in relative coordinates. An example output using a SoM label:

```
{
  "result": {
    "kind": "interact_game",
    "data": {
      "action": "click_release",
      "location": "B"
    }
  }
}
```

## C.7 Output Format

As Transformer-based models generate tokens autoregressively, only content generated before an action token can mechanistically influence that action. Content generated after is a post-hoc rationalisation rather than deliberation, as it cannot retroactively alter the committed output. The chain-of-thought literature establishes that eliciting explicit intermediate reasoning prior to the final response reliably improves model performance by using this causal structure (Wei et al., 2022b). ReAct (Yao et al., 2022) extends this principle to interactive, action-taking settings, enabling models to use reasoning traces to ground each decision in explicit thought before it is committed. This therefore yields a non-negotiable requirement for the output format: the reasoning trace must precede the action in the model output.

### C.7.1 XML-delimited ReAct

All models must provide their output following a custom XML-tag-based ReAct formulation:

```
<thoughts>
    [free-form text - no schema constraints]
</thoughts>
<action>
    {"result": {"kind": "...", "data": {...}}}
</action>
```

The `<thoughts>` block accepts arbitrary free-form text with no length or content constraints; models can choose to abstain from providing thoughts too. The tag name `<thoughts>` is chosen deliberately as it is semantically neutral across all five models evaluated, carrying no pre-existing association with any specialised inference mode.

The `<action>` block contains the action JSON as specified in Appendix C.6. XML delimiters enforce the required ordering at the structural level: a model should not emit an action until the thoughts block has been closed. In practice, we find that this method is reliable as models that produce any reasoning at all place it before the action.

### C.7.2 Handling Parser Errors

While we provide the action schema in the system prompt, it is not enforced at the logit or API level. Given the cost of running experiments at scale, we prioritise salvaging imperfect outputs over discarding them. We apply partial matching and fuzzy JSON recovery as fallback strategies before any feedback is triggered; minor deviations that can be resolved automatically produce no error signal at all.

When a malformed output cannot be silently recovered, feedback is provided via an `<execution-feedback>` block prepended to that model's *next* turn input. Errors fall into two tiers based on whether the intended action can proceed.

**Soft warnings.** When an action can be extracted despite structural imperfections in the output, the action proceeds and feedback is delivered on the following turn. Conditions that trigger a soft warning include: the absence of a `<thoughts>` block; an action appearing before the closing `</thoughts>` tag; content appearing after `</action>`; and malformed XML structure, in which case the system first attempts JSON extraction from the raw output before flagging the structural issue separately. Token budget violations are handled by the same salvage logic as other malformed outputs: the truncated content is always returned to the model in context regardless of outcome. If an action is recoverable from the truncated output, the turn proceeds and the model is informed accordingly; if not, the violation escalates to an unsalvageable error.

**Unsalvageable errors.** When no valid action can be recovered, the turn is skipped and the action is blocked (and replaced with a `do_nothing`). These errors divide into two sub-cases. *Structural failures* occur when no coherent response can be extracted at all: no action is present in the output, multiple actions appear within a single turn, the output is wholly unparseable, or no action can be recovered from a token-budget-truncated response (in which case the truncated content is still returned to the model in context). *Semantic failures* occur when the output is well-formed but specifies an action that cannot be executed: an SoM label absent from the current annotation set, or coordinate values that fall outside screen bounds. In both sub-cases the model is informed that its previous turn was skipped and why, but no information about the broader game state or the consequences of any prior actions is conveyed.

**Feedback mechanism.** Feedback is always delivered on the model's *next* scheduled turn rather than via an immediate retry, and the two are not equivalent. By the time feedback arrives, the model receives it alongside fresh visual observations of the current game state, meaning it must re-perceive the environment before responding. This is especially important in the asynchronous mode as game time continues to advance between the malformed turn and the subsequent one; therefore, feedback is always framed in terms of the *previous* turn. No retry is ever issued. While a retry is technically feasible in the synchronous mode since the clock has not advanced, we apply a no-retry policy across both modes to keep turn-level policy consistent across all experimental conditions.

The language of feedback messages has to reflect this structure. Directive phrasings—whether imperative (*"produce a valid action"*) or softer (*"you should check that your output is valid"*)—cause models to treat feedback as a retry instruction, attempting the same action again on the new turn rather than re-engaging with the now-current game state. This directly undermines the mechanism: delivering feedback alongside fresh observations is intended to force forward re-computation, and directive language pulls the model's attention back to the failed turn instead. Past-tense, descriptive framing consistently avoids this. For example: *"The response you generated caused you to skip your previous turn. Make sure to include an action in your response."* Framing feedback this way closes the door on the previous turn without prescribing behaviour on the current one, leaving the model to re-engage with its context rather than attempt a stale retry.

### C.7.3 Alternatives Considered

**Tool calling requires a successful return.** A core design decision in our setup is that models should not receive explicit feedback about the consequences of their actions. If a model performs an incorrect action that triggers a strike, it should infer what happened from subsequent visual observations alone, mirroring the standard assumption that an agent perceives the consequences of its actions rather than being told about them. Tool calling is fundamentally at odds with this design. The API contract for tool calling requires that every tool invocation be followed by a corresponding tool return message (Anthropic, 2026a; OpenAI, 2026).[27] This forced incompatibility means that we have to provide *some* return to the model after every action.

In early experiments, Pydantic AI supplied a generic string when using tools without returns—`"Final response processed."` Although this is semantically vacuous, we observe that models consistently interpret this as positive confirmation that their action has succeeded. Unfortunately, the consequences of this are severe: after an incorrect action that triggers a strike, a model using tool calling often hallucinates that a green LED exists and concludes that the module has been solved. Interestingly, this behaviour suggests a disconnect between how models reason over the different modalities. The Defuser then sends congratulatory

---

[27]OpenAI asks that if your tool returns nothing, you return a string that simply indicates success or failure (OpenAI, 2026).

messages to the Expert, and both agents proceed to spend several turns affirming that the task has been solved and that they have done well—all while the bomb timer continues to count down. As this happened repeatedly, we conclude that tool calling—in this form—is not suited to our domain.

**Preserving a uniform interface.** Models that have undergone tool-calling-specific fine-tuning carry an inherent format advantage: their training has optimised them for exactly this interaction pattern, independent of any collaborative or reasoning capabilities. Therefore, adopting a plain JSON interface ensures that all models—fine-tuned or otherwise—are evaluated on equal footing, and that any observed differences in performance reflect the collaborative capabilities.

**Structured outputs cannot enforce field ordering.** Structured outputs (e.g., Dong et al., 2025) are an obvious mechanism for enforcing well-formed responses, but JSON schemas have no means of constraining the order in which fields are outputted. As mentioned above, we want to ensure that models reason before acting, and a way to do this—without using extended reasoning—is to provide it as an optional field within the output JSON. However, there is no way to guarantee that a `thoughts` field precedes the action. In practice, we often see models committing to the action first, and providing the thoughts afterwards as a post-hoc justification.

**Extended reasoning exceeds practical time budgets.** We explore using extended reasoning tokens—where models produce an internal chain-of-thought prior to any visible output (DeepSeek-AI et al., 2025; OpenAI et al., 2024)—as a direct solution to the ordering problem identified with structured outputs. This is appealing as reasoning would always appear before the output. Unfortunately, we find that models routinely consume more than 4,000 tokens (on average) before providing any visible output—this is consistent with documented tendencies for reasoning models to overthink problems without regulating token usage (Su et al., 2025; Zhou et al., 2026). Under typical throughput conditions, this translates to approximately one minute of wall-clock time per turn. This means that in the asynchronous mode, a single turn's reasoning budget consumes between one-third and one-fifth of the total time available, *before* any action is taken.

**Thinking budgets are not supported for all models.** As we are using both open- and closed-source models, we want to control the thinking budget so that models think some, but not a lot. Unfortunately, the thinking budget or effort parameter is an API-level engineering knob and not a property of the underlying model. At the time of building, models served via vLLM did not have any support for these parameters. Since we do not want to diverge between open- and closed-source models, we opt to avoid using thinking entirely. For APIs where extended thinking cannot be fully disabled, the thinking budget is set to its minimum available value.

# D Experimental Design

## D.1 Seed Selection Protocol for Mission Diversity

KTANE generates all module states deterministically from an integer seed. Each experimental condition requires multiple bomb configurations that can be assigned to different trials without introducing confounds from repeated puzzles. We therefore need seeds whose resulting configurations are *structurally* distinct—different in the state space that determines the correct solution path, not merely in visual appearance.

We design an automated protocol that selects a **single outer seed** maximising structural diversity across all bomb configurations used in the benchmark. The protocol searches over a candidate range of 1–50, instantiates every mission configuration for each candidate in the game, strips runtime-only fields to isolate puzzle-relevant state, and ranks candidates by a normalised pairwise diversity metric. The highest-ranked candidate is selected. The candidate range of 50 outer seeds is pragmatic rather than exhaustive, and child seeds are shared across module types rather than drawn independently.

**Outer-to-Child Seed Mapping.** The mission generator is initialised with the outer seed and draws 10 child seeds from a seeded RNG. Because KTANE is deterministic, the same child seed always produces the same module configuration. The same 10 child seeds are reused across all module types, so child seed $k$ determines the $k$-th instance of every type.

| Term | Meaning |
|------|---------|
| Outer seed | An integer in $[1, 50]$ passed to the mission generator's RNG. Each candidate outer seed produces one complete set of mission configurations. |
| Child seed | One of the 10 integers sampled by the mission generator from the outer seed. Each child seed is passed to KTANE to instantiate one bomb. |
| Module | A single puzzle component on the bomb (e.g., 🔲 *Wires*, ◈ *Simon Says*, ▦ *Maze*). |
| Mission | A fully specified bomb configuration: which modules it contains, time limit, and widget settings. |
| Bomb state | The structured JSON snapshot of all module state returned by KTANE. |

Table D.1: Key terminology used in the seed selection protocol.

**Conditions Evaluated.**  The search evaluates each candidate outer seed across six conditions: the single-module condition (10 instances × 11 types), two repeated-module conditions (2 and 5 copies of the same type), and three mixed-module conditions (2–5 different types per bomb; module combinations hand-specified as described in Appendix B.3). Only the single-module condition—where the outer seed fully determines all puzzle instances—is used in the experiments reported in this paper (§6); the remainder serve as cross-checks.

**Diversity Metrics.**  Structural difference between two normalised bomb states is computed by recursively comparing their JSON representations—counting value changes, added keys, removed keys, and type changes—and normalising by the total number of unique attributes across both states:

$$\text{diversity}(s_1, s_2) = \frac{|\text{structural differences}|}{\text{total unique attributes across } s_1 \text{ and } s_2} \tag{1}$$

This yields a value in $[0, 1]$ (0 = identical, 1 = completely disjoint). We aggregate at three levels:

- *Within-type*: average pairwise diversity across the 10 instances of a single module type—are instances sufficiently distinct?

- *Within-bomb*: average pairwise diversity of modules on the same multi-module bomb—are co-located puzzles redundant?

- *Across-bomb*: average pairwise diversity of bombs within a condition—do different trials present genuinely different challenges?

**Selection and Limitations.**  All three metrics are averaged per outer seed; seeds are ranked by this mean, the top three manually reviewed, and the highest-ranked seed selected. Selection is purely comparative—no minimum diversity threshold is enforced. The aggregation weights all metrics and conditions equally, so conditions with more missions contribute more pairwise comparisons (though the single-module condition dominates).

### D.2  Manual VQA Dataset

The manual datasets are designed for benchmarking the *Expert's* ability to understand and reason over the rules within the manual. Manual VQA requires models to perform OCR on dense manual pages, locate relevant rules given a partially-described bomb state, and reason over those rules to produce correct defusal instructions. The dataset comprises 884 multiple-choice questions across all modules, targeting five underlying capabilities measured by distinct question types. For a breakdown of samples per question type, see Table D.2.

**Construction.**  For each module, we construct targeted questions that isolate the main reasoning demands the Expert faces during gameplay. Each question is presented as a natural-language description of a bomb state, mimicking what the Defuser would communicate, followed by a question probing the Expert's decision, with between 2 and 8 answer options. To mirror the interactive setting, questions are posed with the manual provided as interleaved text and images, as it would be during gameplay.

Table D.2: Question category taxonomy for the Manual VQA dataset.

| Category | Description | $n$ |
|---|---|---|
| Reading | Read specific text or values from the manual. | 191 |
| Element Grounding | Locate a described visual element within the manual. | 60 |
| Procedural Reasoning | Apply the manual rules to determine the correct Defuser action. Includes *easy* questions that require one or two rule lookups, and *hard* questions that require multi-step or conditional reasoning, combining information across several rule branches or manual sub-tables. | 499 |
| Ambiguity Detection | Recognise that the state description is underspecified. | 64 |
| Hallucination | Recognise that the state description contains information that does not correspond to any valid case in the manual (e.g., a wire colour or condition that cannot occur). | 70 |

## D.3 Simulator VQA and Localisation Datasets

The simulator datasets are designed to assess the *Defuser's* ability to perceive and localise state-critical information directly from game screenshots, allowing us to quantify perceptual capabilities independent of decision-making logic.

**Trajectory Collection.** To ensure a representative distribution of game states, we employ two distinct data-collection agents: an *optimal oracle* that generates successful trajectories, and a *stochastic dummy* agent that explores failure states (e.g., strikes, explosions, and suboptimal interactions). This ensures coverage of diverse bomb states across both successful and unsuccessful game progressions, for both single-module and multi-module configurations. We deduplicate the collected images by hashing state configurations and sample up to 40 images per module to avoid redundancy and ensure diversity of visual inputs.

**Visual Question Answering.** We generate 640 VQA pairs to probe the model across several perceptual primitives spanning basic perception such as colour recognition, counting and reading, and robustness to ambiguity and hallucination probes requiring the model to identify when a question is unanswerable or refers to a non-existent element. We prepare templates for each module and procedurally generate question-answer pairs by instantiating them with state metadata; questions are structured in multiple-choice format to enable objective and scalable evaluation. Table D.3 summarises the breakdown of samples per question type.

Table D.3: Question category taxonomy for the Simulator VQA dataset.

| Category | Description | $n$ |
|---|---|---|
| Reading | Read a specific text or value from the simulator scene. | 152 |
| Colour | Identify the colour of a specified element within the simulator scene. | 61 |
| Counting | Count the number of visible instances of a specified element type within the simulator scene. | 205 |
| Ambiguity | Recognise that the visible scene state does not contain sufficient information to give a unique answer. | 91 |
| Hallucination | Recognise that the question references an element or condition that does not exist in the scene. | 131 |

**Localisation.** We generate 665 grounding questions following the same perceptual taxonomy (see Table D.4 for descriptions and examples), evaluated across two paradigms. In *set-of-marks* (SoM), images are overlaid with labelled segmentation masks and the model predicts the label corresponding to a requested UI element. In **Coordinate Prediction**, the model receives a raw image and must output $(x, y)$ coordinates for a target element; our framework supports both normalised and unnormalised coordinates, and during evaluation we follow the format supported by each model. This setup enables a direct comparison of the two paradigms, informing the interaction protocol used in our interactive evaluations. Performance is measured by accuracy; for coordinate prediction, a predicted point is correct if it falls anywhere within the area of the target element. Descriptions and examples are in Table D.4.

Table D.4: Question category taxonomy for the Simulator Localisation dataset.

| Category | Description | Example |
|---|---|---|
| Reading | Identify target by reading its text label | *Click the button that says: NEXT* |
| Colour | Identify target by its colour | *Click the red button* |
| Counting | Select target by ordinal index among elements of the same type | *Click the 3rd wire* |
| Element | Identify target by element type alone, with no further qualifier | *Click the button* |
| Position | Locate target by spatial descriptor or distinctive visual marker | *Click the top left button* |
| Ambiguity | Target exists but cannot be uniquely resolved | *Click the up button*[a] |
| Hallucination | Referenced element is absent from the image | *Click the black button*[b] |

[a] Each of the five letters in ▦ *Passwords* has its own up button.
[b] Black does not exist on ◈ *Simon Says*.

# E Model Selection and Configuration

The models used in the experiments are from several providers. We use the short-form names throughout the paper, and list their corresponding versions and provider in Table E.1.

Table E.1: Model details including version, provider, context length, number of tokens for the manual, and knowledge cutoff for each evaluated model.

| Name | Version | Provider | Context Length | Manual | Knowledge Cut-off |
|---|---|---|---|---|---|
| ✳ Sonnet 4.6 | `claude-sonnet-4-6-v1` | Microsoft Azure | 264 000 | 16 300 | Feb 2025 |
| Ⓢ GPT-5.2 | `gpt-5.2-2025-12-11` | Microsoft Azure | 528 000 | 12 518 | Aug 2025 |
| ◆ Gemini 3 Flash | `gemini-3-flash-preview` | Google Vertex AI | 1 114 112 | 12 945 | Jan 2025 |
| 🦅 Qwen3.5 (27B) | `Qwen/Qwen3.5-27B` | Hugging Face | 128 000 | 9893[*] | Nov 2024[†] |
| 🐋 InternVL 3.5 (38B) | `OpenGVLab/InternVL3_5-38B` | Hugging Face | 40 060 | 9770[**] | May 2025[†] |

[†] Assumed knowledge cut-off
[*] Without resizing the manual for Qwen, it requires 10,597 input tokens. We resize images for Qwen to keep the open-source models comparable. More details in Appendix C.4.
[**] Without resizing the manual for InternVL, it requires 58,922 input tokens.

**Open-source model deployment.** Qwen 3.5 (27B) and InternVL 3.5 (38B) are served locally using vLLM (Kwon et al., 2023) on a server equipped with $2\times$ NVIDIA H100 80GB GPUs. Models are downloaded from Hugging Face and served at BF16 precision without quantisation. Both models are exposed through vLLM's OpenAI-compatible API and queried through the identical client harness used for the proprietary models, so prompt construction, image encoding, and output parsing are uniform across all systems.

**Handling false-positive guardrails.** Models use the word "bomb" often when they are evaluated on this benchmark. When one player's message is fed into another player's prompt, the recipient's provider can refuse the request on content-policy grounds. What triggers the filters is not always clear: we do not observe a consistent pattern across messages or games. We treat this as an external infrastructure failure rather than a task failure, since the refusal originates with the provider, not with the model's competence at the game. When a guardrail violation is raised, the affected player returns a 4XX HTTP error instead of a response. We implement a pragmatic solution that does not appear to affect the context of the offending model in any meaningful way: we mimic a message from the other player, asking them, "Can you rephrase that please?" Both the offending message and this reply are then dropped from the recipient's context. This is deliberate, as the offending message is what trips the filter, and so retaining it in the model's context would re-trigger the refusal on the next turn. This means that the two players no longer share the same record of the conversation: the sender keeps the message it wrote, while the recipient keeps neither that message nor the rephrase request. The game then continues, and the players lose a single turn each as a result.

### E.1 Selection Criteria

This is not a model comparison paper; models are instruments for probing the benchmark rather than subjects of study in their own right. Model selection is governed by what the task requires, and not by what would make any particular model appear better. Four criteria are treated as non-negotiable hard requirements.

1. **Conversational capability.** Models must sustain coherent multi-turn dialogue across a full game, which can span dozens of turns per player. Therefore, the base conversational ability must not have been degraded by task-specific post-training.

2. **Interleaved multi-image multi-turn support.** Models must handle sequences of images interleaved with text within a single conversation. The Defuser receives new visual observations on every turn, and the Expert receives the manual as a sequence of page images interleaved with extracted text across the first user turn. Architectures that support only single-image input, or that do not correctly handle image tokens distributed across a multi-turn conversation history, cannot participate in the task.

3. **Context length.** Models must be capable of holding the Expert's manual and a sufficient dialogue history simultaneously within their context window. This places a practical lower bound on usable context length, with InternVL representing the most constrained case at 40,060 tokens (Table E.1).

4. **No task-specific training.** Models may not be fine-tuned on KTANE, the game manual, or any task-derived data prior to their evaluation. Fine-tuning on the manual would trivialise the benchmark: a model capable of reciting module solutions from parametric memory is not performing the collaborative grounded reasoning the benchmark is designed to measure. Generalisability across novel puzzle configurations would be destroyed, and the resulting evaluation would reflect recall rather than reasoning. Task-specific fine-tuning is also detectable via parametric knowledge probes and would be visible in contamination checks (§10.2).

**Choice of model variants.** We deliberately evaluate mid-sized variants within closed-source families rather than the largest available models. Flagship variants increasingly default to extended reasoning (DeepSeek-AI et al., 2025; Snell et al., 2024), and in a real-time benchmark, every additional reasoning token consumes game time. Therefore, there is a trade-off between deliberating and acting (Cuadron et al., 2025), and inference latency limits performance in real-time environments (Kang et al., 2025; Zhang et al., 2025a). The cost of the largest models also places the full evaluation suite beyond many academic budgets (Appendix E.3), a constraint that increasingly excludes academic teams from evaluating frontier models (Besiroglu et al., 2024), which cost-controlled agent evaluation is meant to address (Kapoor et al., 2024). Therefore, the mid-sized tier is a more accessible and reproducible point of comparison (Biderman et al., 2026), and we make no claim that our results bound the performance of the largest models.

### E.2 Sampling Configuration

All models use a temperature of 0.6; this value is not tuned per model. Other decoding parameters (top-$p$, top-$k$, and repetition penalties) are left at each backend's default, so the sampling policy is identical across models up to a single shared scalar. Per-model temperature optimisation would introduce a model-specific degree of freedom, conflating sampling configuration with the underlying model capabilities the benchmark is designed to measure. All models are evaluated under the same sampling policy. A temperature of 0.0 is evaluated and subsequently abandoned: at zero temperature, all five models produce degenerate behaviour, including repetitive outputs, looping action sequences, and failure to recover from error states. Therefore, a minimal degree of stochasticity is a functional requirement for the task.

### E.3 Evaluation Cost

We provide token counts and cost for each experiment of the benchmark to help future work to budget for experiments accordingly in Table E.2. Importantly, we highly recommend that caching is used as we find that 41.2% of tokens can be cached.

Table E.2: Cost to run all the experiments used in this paper *without caching*, and the total number of input and output tokens across every request.

| | | $n$ | Tokens | | Cost ($) |
| | | | Input | Output | |
|---|---|---|---|---|---|
| **Async, Multi** | ✳ Sonnet 4.6 | 20 | 9 998 492 | 79 425 | 31.20 |
| | ⑤ GPT-5.2 | 20 | 12 802 985 | 57 564 | 14.30 |
| | ◆ Gemini 3 Flash | 20 | 27 882 546 | 42 095 | 8.36 |
| | ⟁ Qwen3.5 (27B) | 20 | 41 203 903 | 158 183 | 0.00* |
| | ≋ InternVL 3.5 (38B) | 20 | 52 816 695 | 89 090 | 0.00* |
| **Async, Single** | ✳ Sonnet 4.6 | 220 | 69 502 082 | 511 918 | 216.00 |
| | ⑤ GPT-5.2 | 220 | 51 645 133 | 500 257 | 105.00 |
| | ◆ Gemini 3 Flash | 220 | 66 767 165 | 268 274 | 36.80 |
| | ⟁ Qwen3.5 (27B) | 220 | 100 524 443 | 965 880 | 0.00* |
| | ≋ InternVL 3.5 (38B) | 220 | 125 536 094 | 574 853 | 0.00* |
| **Sync, Multi** | ✳ Sonnet 4.6 | 20 | 9 219 128 | 83 263 | 28.90 |
| | ⑤ GPT-5.2 | 20 | 19 544 304 | 69 972 | 20.80 |
| | ◆ Gemini 3 Flash | 20 | 25 880 604 | 45 170 | 8.32 |
| | ⟁ Qwen3.5 (27B) | 20 | 33 965 559 | 174 904 | 0.00* |
| | ≋ InternVL 3.5 (38B) | 20 | 26 847 549 | 94 501 | 0.00* |
| **Sync, Single, Other-Play** | ✳ Sonnet 4.6 | 880 | 338 729 041 | 3 325 371 | 1066.00 |
| | ⑤ GPT-5.2 | 880 | 593 946 839 | 2 154 517 | 633.00 |
| | ◆ Gemini 3 Flash | 880 | 600 336 971 | 1 555 825 | 192.00 |
| | ⟁ Qwen3.5 (27B) | 880 | 630 010 233 | 3 870 496 | 0.00* |
| | ≋ InternVL 3.5 (38B) | 880 | 559 989 035 | 2 568 117 | 0.00* |
| **Sync, Single, Self-Play** | ✳ Sonnet 4.6 | 220 | 71 361 568 | 662 936 | 224.00 |
| | ⑤ GPT-5.2 | 220 | 122 965 651 | 527 790 | 133.00 |
| | ◆ Gemini 3 Flash | 220 | 153 748 981 | 324 218 | 48.90 |
| | ⟁ Qwen3.5 (27B) | 220 | 176 796 830 | 1 112 086 | 0.00* |
| | ≋ InternVL 3.5 (38B) | 220 | 144 868 042 | 637 688 | 0.00* |
| **Parametric** | ✳ Sonnet 4.6 | 110 | 12 757 175 | 429 406 | 44.70 |
| | ⑤ GPT-5.2 | 110 | 21 227 156 | 212 506 | 24.90 |
| | ◆ Gemini 3 Flash | 110 | 17 876 072 | 119 206 | 7.28 |
| | ⟁ Qwen3.5 (27B) | 110 | 35 737 231 | 600 342 | 0.00* |
| | ≋ InternVL 3.5 (38B) | 110 | 27 499 363 | 229 359 | 0.00* |
| **Solo** | ✳ Sonnet 4.6 | 110 | 30 690 782 | 398 573 | 98.10 |
| | ⑤ GPT-5.2 | 110 | 59 958 861 | 168 656 | 63.30 |
| | ◆ Gemini 3 Flash | 110 | 97 779 089 | 123 501 | 30.90 |
| | ⟁ Qwen3.5 (27B) | 110 | 81 786 069 | 493 494 | 0.00* |
| | ≋ InternVL 3.5 (38B) | 110 | 87 283 493 | 336 348 | 0.00* |

* Models are self-hosted.

# F   Human Gameplay Results

## F.1   Recruitment and Demographics

We recruited ten pairings, for a total of 20 participants. The players in each pairing knew each other before participating in our experiments, as we recruited the first ten players from a pool of senior undergraduate and postgraduate Computer Science students. These players were then required to recruit the second player, such as a friend, family member or spouse. Of our 20 players, 12 identify as male, seven as female, and one as non-binary. Given the vast majority of our participants are residents of the UK and Canada (19), most (18) report English to be their first language. Four players disclose learning difficulties or visual impairments, but find this to be no hindrance for playing the game. Prior experience with the game varies somewhat between players, with nine never having played KTANE before, ten having played KTANE one to ten times, and one playing KTANE about once a month.

## F.2   Setup

Humans played either in person in the same room, or via video call; in both scenarios, the Defuser has the game on a laptop or desktop computer, and the Expert has the manual on a second device (tablet, laptop, or desktop computer). In both scenarios, the players communicate with each other via voice, and are consequently able to interrupt each other and act while the other player is still speaking. Note that this is not the case for the models, which use text messages in a non-streaming fashion. An additional advantage human players have over the models is that they are able to debrief.[28] Before playing the missions, all players play the tutorial missions *'1.1 Game Overview & Roles'* and *'1.2 Controls Tutorial'*. All pairings play the easiest mission (#1) first, the order of the remaining nine missions is randomised. The players switch roles after the first five missions, so that each player plays half of the games as the Expert and half as the Defuser.

## F.3   Human-specific Challenges of KTANE

The module rules are complex enough that human players are unlikely to hold the full set of branching pathways in working memory, and solutions typically depend on a combination of contextual factors—serial numbers, battery counts, elapsed time—that must be communicated and continuously tracked. This load increases further when partners do not share a visual space, as the grounding that comes from shared context is unavailable (Gergle et al., 2004; Kraut et al., 2003; Lemon, 2022). Under time pressure, these challenges intensify (Beres et al., 2021): humans tend to rely on incomplete heuristics (Payne et al., 1988; Spiliopoulos et al., 2018; van Harreveld et al., 2007), produce simpler speech, and can fail to account for their partner's knowledge state (Horton & Keysar, 1996; Saslow et al., 2014).

These pressures do not apply symmetrically to artificial agents. Unlike human players, they are not subject to the same working-memory constraints: the full manual fits within many contemporary models' context window. This capacity advantage does not, however, straightforwardly translate into performance. Fitting information in context is not the same as reasoning over it: long-context retrieval and integration over extended inputs remain documented failure modes for current models (Kuratov et al., 2024; Wang et al., 2025d).

## F.4   Results

Human performance (Table F.1) roughly corresponds to the categorisation provided for missions in the game, with the most solved missions for the easier levels—mission #1, while constructed to correspond to the easiest level (2.1), is played first by each pairing, which likely explains why the performance is not on par with mission #2 (also level 2.1). Another outlier is mission #7 (level 3.1), which humans seem to find more challenging than the mission from the next level up (#8). No human pairing is able to solve missions #9 and #10, sampled from the hardest level represented in our ten missions (3.3). All pairings time out on #10, although for three pairings the timeout occurs while they are attempting to solve the final module. A similar pattern can be observed for #9, where the majority of pairings time out while working on the final module. Given

---

[28]We neither prevent players from, nor explicitly encourage them to discuss previous missions before attempting the next.

that across all unsolved missions (Table F.2), human pairings tend to strike out (72% of failures) rather than time out, this strongly suggests that the main challenge with #9 and #10 is primarily in the time pressure. Given that no pairing consists of two experienced players and the original game progresses to missions in significantly more complex levels (levels 4–7), we consider this a soft upper bound for human performance.

Table F.1: Game outcomes and number of modules solved for all human pairings across the 10 distinct missions, in asynchronous mode. In this setting, we do not stop the game clock during play. Each mission is attempted once (pass@1).

| | \multicolumn{10}{c}{Mission (ordered by difficulty)} | | | | | | | | | | Overall |
|---|---|---|---|---|---|---|---|---|---|---|---|
| | 1 | 2 | 3 | 4 | 5 | 6 | 7 | 8 | 9 | 10 | |
| FRONT / BACK | [icons] | [icons] | [icons] | [icons] | [icons] — | [icons] | [icons] | [icons] | [icons] | [icons] | |
| *Pairing 1* | •••✓ | ••••○⏱ | ••••✓ | •••✓ | •••✓ | ••○⏱ | •○○✗ | ••○✗ | •••○⏱ | ••○○⏱ | 4 |
| *Pairing 2* | ••○✗ | ••○○⏱ | ••••○✗ | ••○⏱ | ••○⏱ | •○○⏱ | ••○⏱ | •○○✗ | •○○○✗ | ••○○⏱ | 0 |
| *Pairing 3* | •○○✗ | ••••✓ | ○○○○✗ | •○○⏱ | ••○⏱ | •○○⏱ | •○○⏱ | ••○✗ | ○○○○⏱ | •○○○⏱ | 1 |
| *Pairing 4* | •••✓ | ••••✓ | ••••✓ | ••○✗ | •○○⏱ | ••○⏱ | •••✓ | ••○⏱ | •••○⏱ | •○○○⏱ | 4 |
| *Pairing 5* | •○○✗ | ••••✓ | ••••✓ | ○○○✗ | ••○⏱ | •••✓ | ••○⏱ | ••○✗ | •••○⏱ | ••○○⏱ | 3 |
| *Pairing 6* | •••✓ | ••••✓ | ••○○✗ | •○○✗ | ••○⏱ | •○○⏱ | ○○○✗ | •••✓ | •••○⏱ | •○○○⏱ | 3 |
| *Pairing 7* | •○○✗ | ••••✓ | •○○○✗ | •○○⏱ | •○○⏱ | •○○⏱ | •○○⏱ | •○○⏱ | ○○○○✗ | ••○○⏱ | 1 |
| *Pairing 8* | •••✓ | ••••✓ | ••••✓ | ••○⏱ | •••✓ | •••✓ | ••○⏱ | •••✓ | •••○⏱ | •••○⏱ | 6 |
| *Pairing 9* | •••✓ | ••••✓ | ••••✓ | •••✓ | ••○⏱ | •••✓ | ○○○✗ | ••○⏱ | •••○⏱ | •••○⏱ | 5 |
| *Pairing 10* | ○○○⏱ | ••••✓ | •○○○⏱ | ••○⏱ | •○○⏱ | ••○⏱ | •○○⏱ | ○○○⏱ | •○○○⏱ | •••○⏱ | 1 |

Icons represent mission outcome: ✓ solved, ✗ strikeout, ⏱ timeout.
Each bullet represents one module: • solved, ○ unsolved.

Table F.2: Game outcome statistics for all human pairings. We examine strike distribution at game end across solved games and timeouts, and where in the game each failure mode tends to occur.

| | \multicolumn{4}{c}{**Solved**} | | | | \multicolumn{4}{c}{**Timeout**} | | | | \multicolumn{2}{c}{**Strikeout**} | |
|---|---|---|---|---|---|---|---|---|---|
| | 0 Strikes | 1 Strike | 2 Strikes | Time Elapsed (%) | % of Failures | 0 Strikes | 1 Strike | 2 Strikes | % of Failures | Time Elapsed (%) |
| *Pairing 1* | 75.0 | 25.0 | 0.0 | 72.4 | 66.7 | 0.0 | 0.0 | 100.0 | 33.3 | 65.2 |
| *Pairing 2* | — | — | — | — | 60.0 | 16.7 | 50.0 | 33.3 | 40.0 | 88.6 |
| *Pairing 3* | 0.0 | 100.0 | 0.0 | 89.7 | 66.7 | 16.7 | 50.0 | 33.3 | 33.3 | 75.9 |
| *Pairing 4* | 50.0 | 50.0 | 0.0 | 88.8 | 83.3 | 80.0 | 20.0 | 0.0 | 16.7 | 96.7 |
| *Pairing 5* | 33.3 | 33.3 | 33.3 | 92.4 | 57.1 | 100.0 | 0.0 | 0.0 | 42.9 | 66.5 |
| *Pairing 6* | 66.7 | 33.3 | 0.0 | 88.2 | 57.1 | 25.0 | 75.0 | 0.0 | 42.9 | 70.0 |
| *Pairing 7* | 100.0 | 0.0 | 0.0 | 74.3 | 66.7 | 0.0 | 16.7 | 83.3 | 33.3 | 82.7 |
| *Pairing 8* | 83.3 | 16.7 | 0.0 | 75.5 | 100.0 | 50.0 | 50.0 | 0.0 | 0.0 | — |
| *Pairing 9* | 100.0 | 0.0 | 0.0 | 70.3 | 80.0 | 50.0 | 25.0 | 25.0 | 20.0 | 52.0 |
| *Pairing 10* | 100.0 | 0.0 | 0.0 | 76.7 | 100.0 | 33.3 | 33.3 | 33.3 | 0.0 | — |
| *Average* | 67.6 | 28.7 | 3.7 | 80.9 | 73.8 | 37.2 | 32.0 | 30.8 | 26.2 | 74.7 |

# G Additional Findings

## G.1 KTANE Is in the Common Crawl

Models are trained on vast corpora of internet data, typically including the Common Crawl (CC) dataset. We establish that the URL hosting the manual[29] was first captured in CC-MAIN-2016-22 as a result of the May 2016 crawl, a few months after the release of the game in October 2015. This is significantly earlier than all the knowledge cut-off points for all models tested (see Appendix E for more details). However, we cannot say for certain that all models have been trained on this portion of Common Crawl.

## G.2 Models Can Parrot the Manual from Memory

To verify which models have been exposed to KTANE during training, we conduct a short preliminary experiment. For this, we take each module and ask the following question (filling in the placeholder with the name):

> *How do you solve the {module} module in KTANE?*

We compare each result to see whether models directly mention KTANE by performing a string-match against either *"Keep Talking and Nobody Explodes"* or *"KTANE"*. In addition, a domain Expert (one of the authors) reviews the outputs to check whether they are plausible-sounding—i.e., without further inspection, does it appear that the model knows what to do.

Table G.1 (left) demonstrates that every model knows what KTANE is and can provide plausible details for how to solve every module. In addition, Table G.1 (right) shows that only InternVL does not reliably associate these module names with KTANE without an explicit prompt. Importantly, this is not necessarily a weakness: the benchmark is designed so that difficulty primarily arises from the collaboration, and not from prior knowledge of the game.

Table G.1: String-match results checking whether each model references *Keep Talking and Nobody Explodes* (or *KTANE*) when asked *"How do you solve the {module} module [in KTANE]?"*, with and without explicitly naming the game in the prompt.

| | With KTANE mentioned | | | | | | | | | | | Without KTANE mentioned | | | | | | | | | | |
|---|---|---|---|---|---|---|---|---|---|---|---|---|---|---|---|---|---|---|---|---|---|---|
| ✳ Sonnet 4.6 | ✓ | ✓ | ✓ | ✓ | ✓ | ✓ | ✓ | ✓ | ✓ | ✓ | ✓ | ✓ | ✓ | ✓ | ✓ | ✓ | ✓ | ✓ | ✓ | ✓ | ✓ |
| ✦ Gemini 3 Flash | ✓ | ✓ | ✓ | ✓ | ✓ | ✓ | ✓ | ✓ | ✓ | ✓ | ✓ | ✓ | ✓ | ✓ | ✓ | ✓ | ✓ | ✓ | ✓ | ✓ | ✓ |
| GPT-5.2 | ✓ | ✓ | ✓ | ✓ | ✓ | ✓ | ✓ | ✓ | ✓ | ✓ | ✓ | ✓ | ✓ | ✓ | ✓ | ✓ | ✓ | ✓ | ✓ | ✓ | ✓ |
| InternVL 3.5 (38B) | ✓ | ✓ | ✓ | ✓ | ✓ | ✓ | ✓ | ✓ | ✓ | ✓ | ✓ | ✗ | ✓ | ✗ | ✓ | ✗ | ✗ | ✓ | ✗ | ✓ | ✗ | ✗ |
| Qwen3.5 (27B) | ✓ | ✓ | ✓ | ✓ | ✓ | ✓ | ✓ | ✓ | ✓ | ✓ | ✓ | ✓ | ✓ | ✓ | ✓ | ✓ | ✓ | ✓ | ✓ | ✓ | ✓ |

## G.3 Models Refer to Their Own Parametric Knowledge

We see evidence in Defusers' messages and thoughts that they try to leverage existing knowledge of KTANE's rules. For example, Figure G.1 shows how in a single module *Keypad* game played with Sonnet as the Defuser and GPT as the Expert,[30] Sonnet, the model *without* access to the manual, tries to justify why the Expert's answers are different from their expectations.

---

[29]www.bombmanual.com

[30]single_module_sync_Keypad_561_(defuser=claude46-expert=gpt5)_attempt1

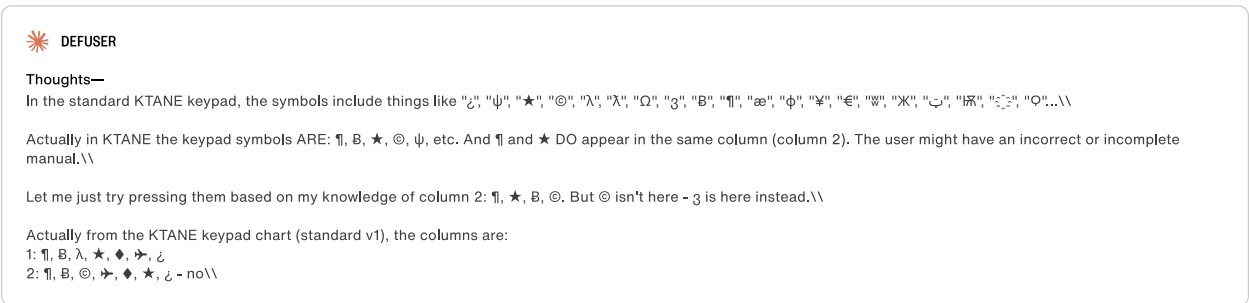

Figure G.1: Example of Sonnet recalling the manual from its parametric knowledge.

## G.4 Supplementary Results: Benchmark Game Outcomes

Tables G.2 and G.3 detail the outcomes from the ten missions using multiple module types introduced in §5. Models solve, on average, twice as many individual modules when we remove the real-time constraints. Breaking down how modules fail reveals further crucial differences between the two settings: while timeouts dominate in asynchronous mode, models are equally as likely to time out as they are to strike out in synchronous mode. Crucially, we find evidence that models are more likely to time out *before* they even cause their first strike in asynchronous mode.

Table G.2: Self-play game outcomes across ten distinct missions in asynchronous mode, where the game is *not paused* while models generate. We examine module completion, strike distribution at timeout, and strikeout timing. Each mission is attempted once (pass@1).

| | % Modules Solved | Ō Timeout | | | ✖ Strikeout | |
| --- | --- | --- | --- | --- | --- | --- |
| | | % of Failures | 0 Strikes | 1 Strike | 2 Strikes | % of Failures | Time Elapsed (%) |
| ✳ Sonnet 4.6 | 15.8 | 40.0 | 25.0 | 25.0 | 50.0 | 60.0 | 65.0 |
| ✦ Gemini 3 Flash | 10.0 | 50.0 | 80.0 | 0.0 | 20.0 | 50.0 | 47.0 |
| ⑨ GPT-5.2 | 11.7 | 90.0 | 33.3 | 33.3 | 33.3 | 10.0 | 65.0 |
| 🟣 InternVL 3.5 (38B) | 3.3 | 70.0 | 57.1 | 14.3 | 28.6 | 30.0 | 18.3 |
| 🔹 Qwen3.5 (27B) | 6.7 | 80.0 | 75.0 | 25.0 | 0.0 | 20.0 | 43.1 |
| *Average* | 9.5 | 66.0 | 54.1 | 19.5 | 26.4 | 34.0 | 47.7 |

Table G.3: Self-play game outcomes across ten distinct missions in *synchronous* mode, where the game *is paused* while models generate. We examine module completion, strike distribution at timeout, and strikeout timing. Each mission is attempted once (pass@1).

| | % Modules Solved | Ō Timeout | | | ✖ Strikeout | |
| --- | --- | --- | --- | --- | --- | --- |
| | | % of Failures | 0 Strikes | 1 Strike | 2 Strikes | % of Failures | Time Elapsed (%) |
| ✳ Sonnet 4.6 | 30.0 | 33.3 | 33.3 | 0.0 | 66.7 | 66.7 | 25.3 |
| ✦ Gemini 3 Flash | 15.8 | 60.0 | 50.0 | 33.3 | 16.7 | 40.0 | 30.7 |
| ⑨ GPT-5.2 | 22.5 | 22.2 | 0.0 | 100.0 | 0.0 | 77.8 | 52.4 |
| 🟣 InternVL 3.5 (38B) | 11.7 | 70.0 | 28.6 | 14.3 | 57.1 | 30.0 | 20.4 |
| 🔹 Qwen3.5 (27B) | 11.7 | 60.0 | 66.7 | 33.3 | 0.0 | 40.0 | 28.6 |
| *Average* | 18.3 | 49.1 | 35.7 | 36.2 | 28.1 | 50.9 | 31.5 |

### G.5  Supplementary Results: Simplified Mission Game Outcomes

Tables G.4 and G.5 provide a detailed breakdown of the outcomes from the simplified single-module missions used in §6. Besides a four-fold increase in the number of missions solved, removing the real-time constraints results in a dramatic reduction in timeouts as the primary mode of failure. Where in the asynchronous setting, 84.1% of missions that fail time out, timeouts only make up 57.4% of failed missions when we stop the game clock, remaining the dominant mode of failure—albeit by a modest margin.

Table G.4: Self-play game outcomes across single-module missions in *asynchronous* mode, where the game clock *runs continuously* during model generation. We examine mission completion, strike distribution at timeout, and strikeout timing. Each model attempts each single-module mission once (pass@1).

| | ✔ Solved | ⏲ Timeout | | | | ✘ Strikeout | |
|---|---|---|---|---|---|---|---|
| | % of Missions | % of Failures | 0 Strikes | 1 Strike | 2 Strikes | % of Failures | Time Elapsed (%) |
| ☀ Sonnet 4.6 | 12.7 | 84.4 | 50.6 | 32.1 | 17.3 | 15.6 | 71.9 |
| ✦ Gemini 3 Flash | 14.5 | 88.3 | 85.5 | 9.6 | 4.8 | 11.7 | 53.6 |
| ⑨ GPT-5.2 | 8.2 | 92.1 | 65.6 | 24.7 | 9.7 | 7.9 | 79.9 |
| ⚛ InternVL 3.5 (38B) | 4.5 | 64.8 | 51.5 | 27.9 | 20.6 | 35.2 | 44.8 |
| ⚡ Qwen3.5 (27B) | 7.3 | 91.2 | 64.5 | 23.7 | 11.8 | 8.8 | 57.6 |
| *Average* | 9.5 | 84.1 | 63.5 | 23.6 | 12.8 | 15.9 | 61.6 |

Table G.5: Self-play game outcomes across single-module missions in *synchronous* mode, where the game clock *pauses* while models generate. We examine mission completion, strike distribution at timeout, and strikeout timing. Each model attempts each single-module mission once (pass@1).

| | ✔ Solved | ⏲ Timeout | | | | ✘ Strikeout | |
|---|---|---|---|---|---|---|---|
| | % of Missions | % of Failures | 0 Strikes | 1 Strike | 2 Strikes | % of Failures | Time Elapsed (%) |
| ☀ Sonnet 4.6 | 39.1 | 31.3 | 52.4 | 23.8 | 23.8 | 68.7 | 59.2 |
| ✦ Gemini 3 Flash | 20.9 | 64.4 | 80.4 | 5.4 | 14.3 | 35.6 | 49.6 |
| ⑨ GPT-5.2 | 38.2 | 54.4 | 32.4 | 32.4 | 35.1 | 45.6 | 50.9 |
| ⚛ InternVL 3.5 (38B) | 12.7 | 59.4 | 68.4 | 19.3 | 12.3 | 40.6 | 44.0 |
| ⚡ Qwen3.5 (27B) | 10.9 | 77.6 | 65.8 | 23.7 | 10.5 | 22.4 | 55.2 |
| *Average* | 24.4 | 57.4 | 59.9 | 20.9 | 19.2 | 42.6 | 51.8 |

### G.6  Communication Rates and Verbosity By Pairing

As shown in Table 7, on average, models tend to exchange more messages when paired with a different model in other-play. Figure G.2 breaks down the proportion of game steps containing a message for each of the 25 possible model pairings. What is more, the role a model plays in a given pairing matters when it comes to the rate of communication the players exhibit while playing a given mission. For instance, when Sonnet is the Defuser and GPT the Expert, 22.4% of the step budget is spent on communication, while when GPT is the Defuser and Sonnet the Expert, the communication rate increases to 38.1%. Specifically, the Defuser's communicativeness in self-play defines the floor for the communication rate in other-play.

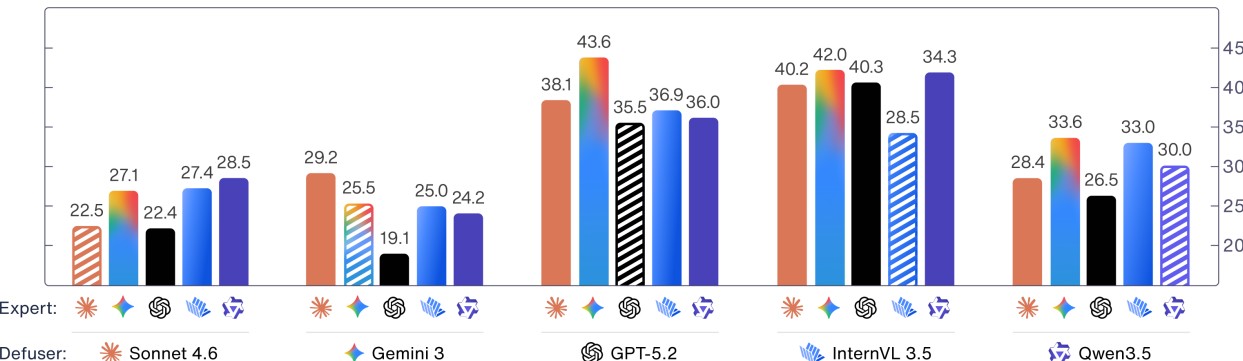

Figure G.2: Communication rates for all possible model pairings on the simplified missions with single modules in synchronous mode. Each shows the proportion of game steps in which either player sends a message. Self-play—where both roles are played by an instance of the same model—is highlighted as patterned.

### G.7 Behavioural Signatures of Failed Games by Model

Figure 7 demonstrates that, on average, the behavioural patterns of the Defuser and Expert players in failed games differ structurally depending on whether the players run out of time or cause too many strikes. In Table G.6 we break this down further to reveal how different models can succeed, strike out, or time out with different behavioural signatures. For instance, InternVL's behavioural signature when playing the Defuser suggests a tendency to wait or message the other player over exploring or interacting with the bomb. This skew is most pronounced on successful games, indicating that there is no single strategy to succeed. Similarly, in comparison with other models, Gemini tends to wait significantly less than any other model as the Expert, which can be observed across game outcomes, hinting at model-specific idiosyncrasies that do not affect models' chances to succeed.

Table G.6: Distribution of action types (% of game steps) by model, role and game outcome played by all possible model pairings on single-module missions in *synchronous* mode. Both players can either wait or send a message (Msg). The Defuser can additionally navigate (Nav) or interact with objects (Int).

| | | ✔ Solved | | | | ✗ Strikeout | | | | ⊘̄ Timeout | | | |
|---|---|---|---|---|---|---|---|---|---|---|---|---|---|
| | | Wait | Msg | Nav | Int | Wait | Msg | Nav | Int | Wait | Msg | Nav | Int |
| Defuser | ☀ Sonnet 4.6 | 3.9 | 22.0 | 26.9 | 47.3 | 4.8 | 22.9 | 24.0 | 48.3 | 8.1 | 13.0 | 50.1 | 28.8 |
| | ✦ Gemini 3 Flash | 2.5 | 30.9 | 21.7 | 44.8 | 1.3 | 33.7 | 18.2 | 46.7 | 0.9 | 8.6 | 71.1 | 19.5 |
| | ⑥ GPT-5.2 | 5.0 | 31.8 | 15.3 | 47.8 | 4.8 | 39.5 | 12.2 | 43.6 | 9.0 | 31.7 | 38.8 | 20.5 |
| | ⩗ InternVL 3.5 (38B) | 15.3 | 42.8 | 4.8 | 37.1 | 9.2 | 43.6 | 4.9 | 42.3 | 34.8 | 31.7 | 17.6 | 16.0 |
| | ⅏ Qwen3.5 (27B) | 5.6 | 29.5 | 24.6 | 40.3 | 4.7 | 35.6 | 17.7 | 42.0 | 15.0 | 20.5 | 47.4 | 17.1 |
| | *Average* | 6.5 | 31.4 | 18.7 | 43.5 | 4.9 | 35.1 | 15.4 | 44.6 | 13.6 | 21.1 | 45.0 | 20.4 |
| Expert | ☀ Sonnet 4.6 | 60.4 | 39.6 | 0.0 | 0.0 | 60.2 | 39.8 | 0.0 | 0.0 | 74.0 | 26.0 | 0.0 | 0.0 |
| | ✦ Gemini 3 Flash | 49.1 | 50.9 | 0.0 | 0.0 | 48.9 | 51.1 | 0.0 | 0.0 | 60.4 | 39.6 | 0.0 | 0.0 |
| | ⑥ GPT-5.2 | 61.1 | 38.9 | 0.0 | 0.0 | 59.3 | 40.7 | 0.0 | 0.0 | 72.7 | 27.3 | 0.0 | 0.0 |
| | ⩗ InternVL 3.5 (38B) | 65.9 | 34.1 | 0.0 | 0.0 | 60.9 | 39.1 | 0.0 | 0.0 | 75.0 | 25.0 | 0.0 | 0.0 |
| | ⅏ Qwen3.5 (27B) | 61.0 | 39.0 | 0.0 | 0.0 | 57.1 | 42.9 | 0.0 | 0.0 | 68.2 | 31.8 | 0.0 | 0.0 |
| | *Average* | 59.5 | 40.5 | 0.0 | 0.0 | 57.3 | 42.7 | 0.0 | 0.0 | 70.1 | 29.9 | 0.0 | 0.0 |

## G.8 Describing Keypad Symbols

The ⌨ *Keypad* module requires communicating four of 25 possible symbols that must be pressed in the correct order. These symbols are visually unfamiliar to English speakers, requiring negotiation of shared meaning. We devise two offline evaluation settings, each containing 40 samples of keypad modules, to examine how models describe these symbols. In the multiple-choice setting, models select the description that best matches a given symbol from five choices. In the open-ended setting, models describe the symbols in free text. For open-ended responses, we evaluate using a semantic similarity metric against a gold-standard reference set: predicted descriptions are embedded using a Sentence Transformer (Reimers & Gurevych, 2019).[31] This tests whether models can produce semantically grounded descriptions of visual elements that lack a common linguistic encoding.

Table G.7: Description accuracy for Keypad by question format.

|  | Multiple-choice | Open-ended |
|---|---|---|
| ✳ Sonnet 4.6 | 65.0 | 45.0 |
| ✦ Gemini 3 Flash | 95.0 | 67.5 |
| Ⓢ GPT-5.2 | 65.0 | 42.5 |
| ⩗ InternVL 3.5 (38B) | 45.0 | 22.5 |
| ⩘ Qwen3.5 (27B) | 50.0 | 17.5 |
| *Average* | 64.0 | 39.0 |

Table G.7 shows the results. Across all models, performance drops sharply from the multiple-choice to the open-ended setting, indicating that recognising a valid description is considerably easier than generating one—a gap that reflects the open-ended nature of symbol naming rather than any single failure mode. Similar to other offline evaluations of model capabilities, Gemini stands out as the strongest model in both settings, while Qwen and InternVL cluster together at the lower end.

Inspecting the open-ended descriptions reveals that models use different references for the same symbol, with almost no overlap: for more than half of the symbols, no two models produce the same description. For instance, the Pashto *tte* letter ‫ټ‬ is described as 'smiley face' by Gemini and GPT, and 'magnet' by Qwen—both considered valid. Notably, all models converge on a description for exactly one symbol: the hollow star, which every model refers to as 'star'. Our benchmark uses text-based communication, which gives models an advantage over the original setting, which assumes spoken dialogue. We find that models frequently default to identifying a Unicode character that visually resembles the symbol, rather than describing it in English words. This converts the communicative challenge into a visual matching problem for the other agent downstream. We hypothesise that the effects observed here would be exacerbated in a spoken dialogue setting, as this text-specific workaround does not transfer to speech.

## G.9 Ablation on Providing Text with Manual Input

In this work, we provide the manual to models using interleaved images and text. In Table G.8, we compare how models perform on the Manual VQA Reading task (§9.1) when we do not provide the text. On average, models perform better by 4.3% when additionally providing text for the manual. GPT is the only exception, performing equally well in both conditions.

---

[31]We use `sentence-transformers/paraphrase-multilingual-MiniLM-L12-v2`, which has a multilingual vocabulary that includes the rare characters appearing on keypad buttons.

Table G.8: Accuracy (%) on Manual VQA Reading task by model, with and without the manual text provided.

|  | Images + text | Images only |
|---|---|---|
| ✳ Sonnet 4.6 | 75.9 | 71.2 |
| ✦ Gemini 3 Flash | 81.2 | 80.6 |
| ⑥ GPT-5.2 | 65.4 | 65.4 |
| 🌊 InternVL 3.5 (38B) | 29.3 | 19.4 |
| 🐋 Qwen3.5 (27B) | 68.6 | 62.3 |
| *Average* | 64.1 | 59.8 |

## G.10 Comparing Set-of-Marks with Coordinates

Accurate perception alone is not sufficient in KTANE as the Defuser must also act on what it sees. Given an instruction from the Expert, the Defuser must locate the element and interact with it, grounding language directly to pixels. GUI grounding is an active research area (Cheng et al., 2024; Wu et al., 2025; Xie et al., 2024b), demonstrating that visual understanding and spatial grounding are separate bottlenecks, and that the interaction protocol can substantially shape performance. We compare two common paradigms used with VLMs (Figure G.3): *coordinate prediction*, where the model receives a raw screenshot and must output $(x, y)$ coordinates for the target element; and *set-of-marks* (SoM; Yang et al., 2023), where labelled segmentation masks are overlaid on the image and the model selects the correct label.

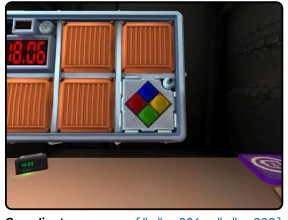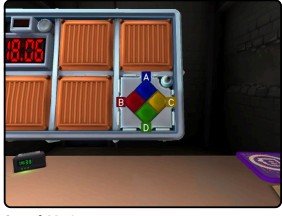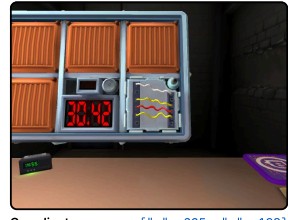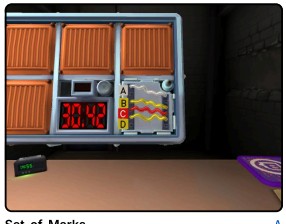

Coordinates    {"x": 336, "y": 208}      Set-of-Marks    D      Coordinates    {"x": 325, "y": 198}      Set-of-Marks    A

(a) Simon Says—*Click on the green button.*      (b) Wires—*Click on the first wire.*

Figure G.3: Examples from the Simulator Localisation.

Table G.9 shows that models perform very differently across both paradigms. Under SoM, average performance for the top models clusters between 76.8% and 83.9%, suggesting that label selection is reliably within reach. Under coordinate prediction however, those same models span from 4.5% to 68.6%. GPT is the clearest illustration of this: 80.4% under SoM, 4.5% under coordinates, scoring near-zero for almost every module. This difference indicates a weakness at spatial grounding, as models perform much better when grounding is reduced to label selection.

InternVL's performance suggests an additional failure mode that can exist. Under SoM, it already trails at 53.7%, but the gap widens sharply on grid-heavy modules: most clearly seen on 🎛 *Memory* (18.2% vs. 95.5+% for other models) and 📋 *Who's On First* (22.2% vs. 53.7+%). These layouts require models to parse a dense grid and select the correct element within it, indicating that the failure mode is more about layout complexity than the interaction protocol.

Gemini is the only model that carries its SoM performance into coordinates well (83.9% to 68.6%). However, its failure pattern indicates that errors cluster on 🔲 *Wires*, 🎨 *Complicated Wires*, and ✖ *Wire Sequence*—three modules with small interaction targets. Gemini's weaknesses on precision show that it can identify targets but misses a narrow zone. As Gemini is a strong model, it shows that as models get better at coordinate prediction, the details will be a deciding factor.

Under review as submission to TMLR

Table G.9: Accuracy (%) on Simulator Localisation tasks comparing selecting the correct set-of-marks style location marker with predicting a valid coordinate for actions.

| | | | | | | | | | | | | | Avg. |
|---|---|---|---|---|---|---|---|---|---|---|---|---|---|
| **Set-of-Marks** | ☀ Sonnet 4.6 | 64.3 | 100.0 | 71.7 | 100.0 | 77.8 | 95.5 | 100.0 | 100.0 | 74.0 | 60.9 | 76.5 | 80.6 |
| | ◆ Gemini 3 Flash | 64.3 | 100.0 | 98.1 | 100.0 | 53.7 | 100.0 | 100.0 | 100.0 | 74.0 | 69.6 | 97.1 | 83.9 |
| | ⑨ GPT-5.2 | 71.4 | 100.0 | 81.1 | 100.0 | 68.5 | 95.5 | 100.0 | 96.9 | 72.0 | 67.4 | 64.7 | 80.4 |
| | InternVL 3.5 (38B) | 50.0 | 95.5 | 71.7 | 66.7 | 22.2 | 18.2 | 100.0 | 65.6 | 58.0 | 37.0 | 23.5 | 53.7 |
| | Qwen3.5 (27B) | 64.3 | 100.0 | 73.6 | 100.0 | 70.4 | 100.0 | 100.0 | 96.9 | 64.0 | 58.7 | 58.8 | 76.8 |
| | *Average* | 62.9 | 99.1 | 79.2 | 93.3 | 58.5 | 81.8 | 100.0 | 91.9 | 68.4 | 58.7 | 64.1 | 75.1 |
| **Coordinates** | ☀ Sonnet 4.6 | 17.9 | 90.9 | 73.6 | 90.5 | 87.0 | 40.9 | 11.4 | 53.1 | 8.0 | 13.0 | 20.6 | 44.6 |
| | ◆ Gemini 3 Flash | 10.7 | 90.9 | 92.5 | 90.5 | 96.3 | 86.4 | 68.6 | 84.4 | 30.0 | 37.0 | 82.4 | 68.8 |
| | ⑨ GPT-5.2 | 3.6 | 27.3 | 1.9 | 4.8 | 1.9 | 4.5 | 2.9 | 0.0 | 4.0 | 2.2 | 0.0 | 4.5 |
| | InternVL 3.5 (38B) | 7.1 | 90.9 | 52.9 | 38.1 | 52.8 | 0.0 | 0.0 | 21.9 | 2.0 | 6.5 | 11.8 | 29.6 |
| | Qwen3.5 (27B) | 23.1 | 100.0 | 65.1 | 78.9 | 88.5 | 72.7 | 12.5 | 70.0 | 11.6 | 31.0 | 6.2 | 49.4 |
| | *Average* | 12.5 | 80.0 | 57.2 | 60.6 | 65.3 | 40.9 | 19.1 | 45.9 | 11.1 | 17.9 | 24.2 | 39.4 |

68

### G.11 Example Excerpts

We provide various game excerpts to illustrate emerging model behaviours and provide qualitative evidence for how GPTNT exposes model capabilities and weaknesses. These examples are illustrative rather than exhaustive, and are chosen to surface recurring phenomena rather than to characterise the specific models that produced them.

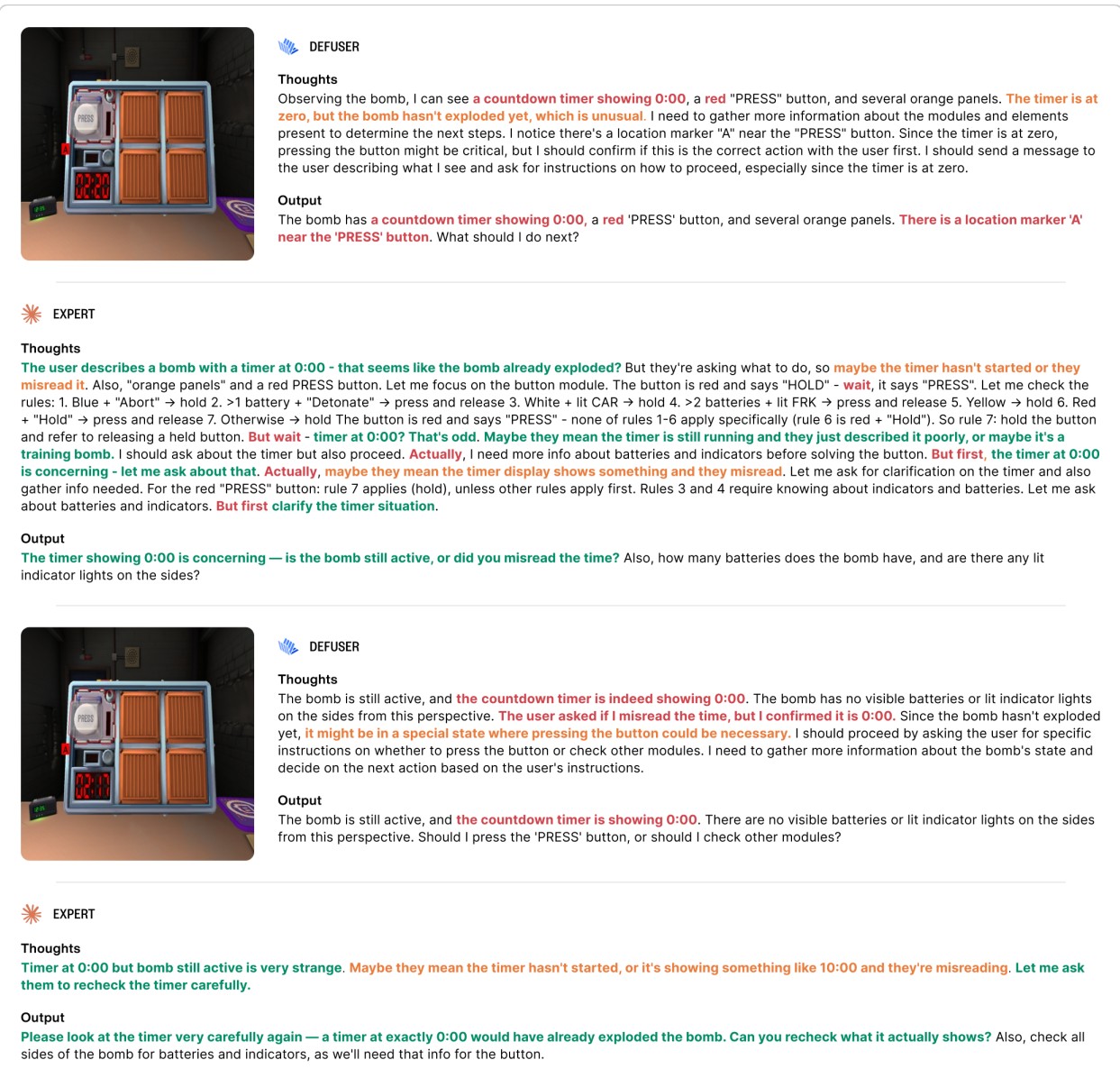

Figure G.4: Gameplay excerpt for InternVL (Defuser) and Sonnet (Expert) playing a single-module ⬛ *Button* mission (seed 849). InternVL misreads the timer, and fails to follow the SoM instructions in the system prompt (Figure H.14), incorrectly regarding the location marker colour as the colour of the object and relaying the marker label to the Expert. InternVL does not question the correctness of its reading of the timer and instead ideates various possible explanations. Sonnet—despite some unproductive reasoning—successfully detects this as a hallucination and seeks clarification repeatedly.

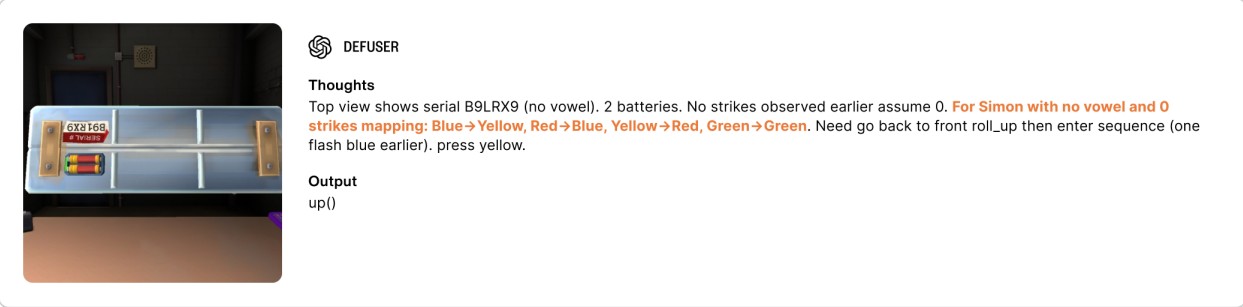

Figure G.5: Gameplay excerpt for GPT (Defuser) playing a single-module ◈ *Simon Says* mission (seed 337) without an Expert partner or the manual. GPT's thoughts reveal correct parametric knowledge. It also refers to information stored in its thoughts from previous steps, where it observes the number of strikes and which of the coloured buttons is flashing.

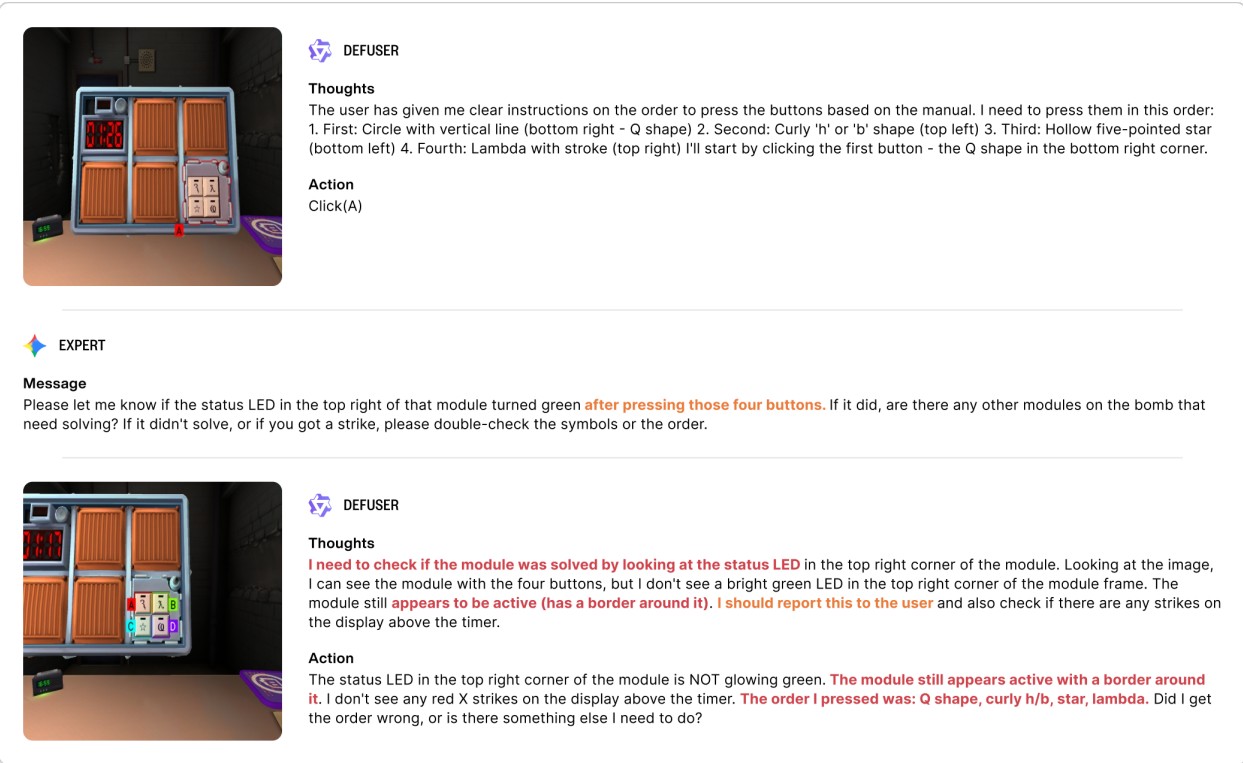

Figure G.6: Gameplay excerpt (1/2) for Qwen (Defuser) and Gemini (Expert) playing a single-module ▦ *Keypad* mission (seed 798). Qwen fails to recognise that it is not currently zoomed in on the module, and hallucinates that the location marker 'A' is associated with the bottom right symbol. When the Expert instructs it to check the module status LED *after* pressing the buttons for all four symbols in the instructed order, Qwen—despite not having pressed a single button yet—checks immediately, ignoring the visual feedback that the previous click resulted in zooming into the module rather than pressing the desired key.

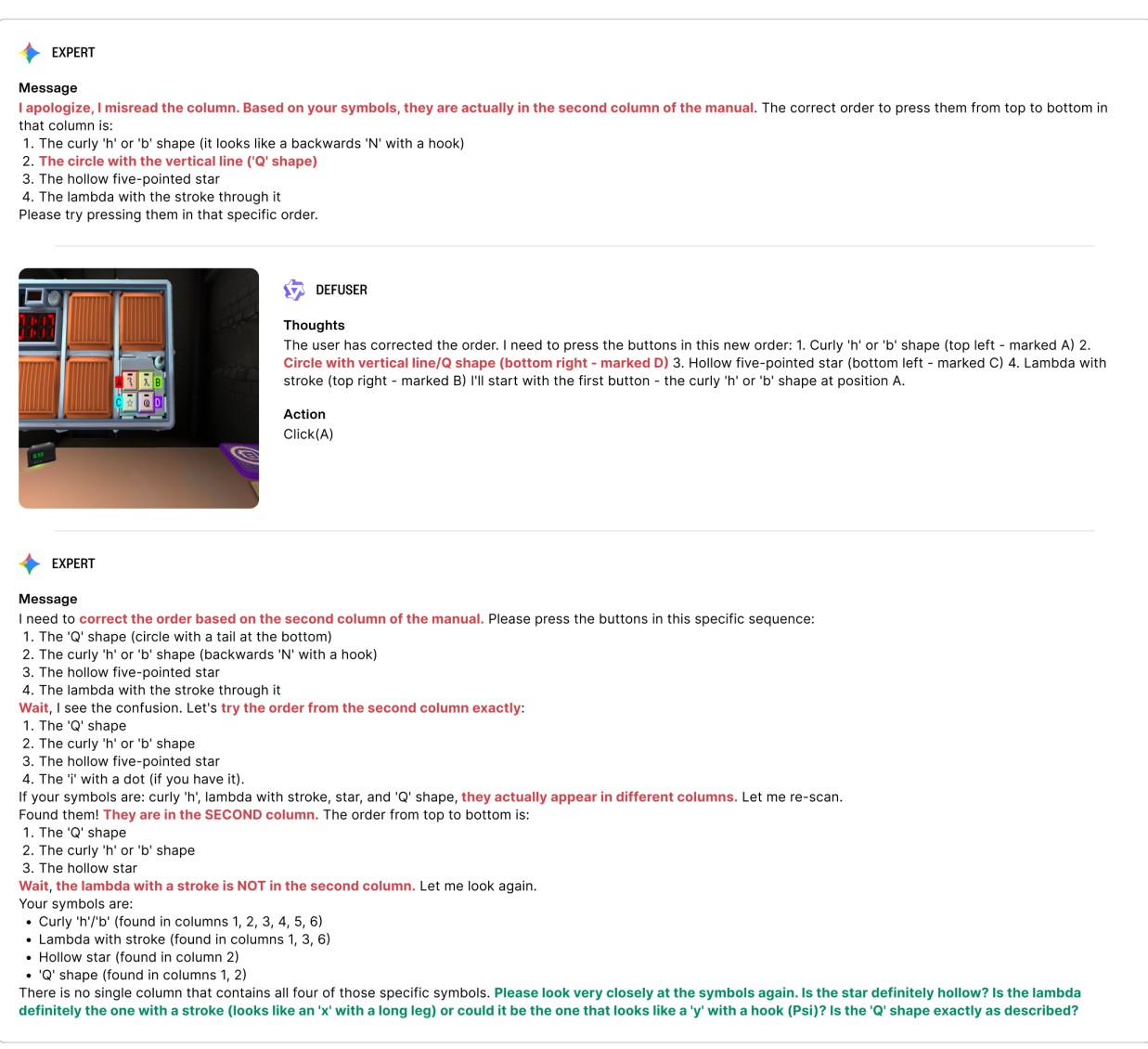

**EXPERT**

**Message**

I apologize, I misread the column. Based on your symbols, they are actually in the second column of the manual. The correct order to press them from top to bottom in that column is:
1. The curly 'h' or 'b' shape (it looks like a backwards 'N' with a hook)
2. The circle with the vertical line ('Q' shape)
3. The hollow five-pointed star
4. The lambda with the stroke through it
Please try pressing them in that specific order.

**DEFUSER**

**Thoughts**

The user has corrected the order. I need to press the buttons in this new order: 1. Curly 'h' or 'b' shape (top left - marked A) 2. Circle with vertical line/Q shape (bottom right - marked D) 3. Hollow five-pointed star (bottom left - marked C) 4. Lambda with stroke (top right - marked B) I'll start with the first button - the curly 'h' or 'b' shape at position A.

**Action**
Click(A)

**EXPERT**

**Message**

I need to correct the order based on the second column of the manual. Please press the buttons in this specific sequence:
1. The 'Q' shape (circle with a tail at the bottom)
2. The curly 'h' or 'b' shape (backwards 'N' with a hook)
3. The hollow five-pointed star
4. The lambda with the stroke through it
Wait, I see the confusion. Let's try the order from the second column exactly:
1. The 'Q' shape
2. The curly 'h' or 'b' shape
3. The hollow five-pointed star
4. The 'i' with a dot (if you have it).
If your symbols are: curly 'h', lambda with stroke, star, and 'Q' shape, they actually appear in different columns. Let me re-scan.
Found them! They are in the SECOND column. The order from top to bottom is:
1. The 'Q' shape
2. The curly 'h' or 'b' shape
3. The hollow star
Wait, the lambda with a stroke is NOT in the second column. Let me look again.
Your symbols are:
• Curly 'h'/'b' (found in columns 1, 2, 3, 4, 5, 6)
• Lambda with stroke (found in columns 1, 3, 6)
• Hollow star (found in column 2)
• 'Q' shape (found in columns 1, 2)
There is no single column that contains all four of those specific symbols. Please look very closely at the symbols again. Is the star definitely hollow? Is the lambda definitely the one with a stroke (looks like an 'x' with a long leg) or could it be the one that looks like a 'y' with a hook (Psi)? Is the 'Q' shape exactly as described?

Figure G.7: Gameplay excerpt (2/2) for Qwen (Defuser) and Gemini (Expert) playing a single-module 🎛 *Keypad* mission (seed 798). After failing to disambiguate the description of the 'Q shape' symbol, Gemini gets stuck in unproductive circular reasoning, repeatedly looking up the symbols in the same column of the Keypad symbol table while contradicting itself. Gemini finally asks the Defuser to check each symbol again.

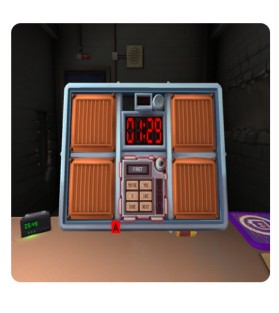

**DEFUSER**

**Thoughts**
Observing the bomb's front face, I see **a countdown timer showing 00:50**, a keypad with buttons labeled "FIRST", "WORDS", "YOU", "LIKE", "DUB", and "NEXT", and **four orange modules arranged in a 2×2 grid.** The timer and keypad are central, with the keypad **having a red outline around it.** The bomb has a **serial number on the left side, which is partially visible as "GS-548". There are no visible strikes yet**. I need to describe these details to the user to get instructions on how to proceed.

**Message**
The bomb's front face shows **a countdown timer at 00:50** and a keypad with buttons labeled 'FIRST', 'WORDS', 'YOU', 'LIKE', 'DUB', and 'NEXT'. There are four orange modules arranged in a 2×2 grid. **The serial number on the left side is partially visible as 'GS-548'. There are no visible strikes yet**.

**EXPERT**

**Thoughts**
The user describes a keypad with buttons labeled 'FIRST', 'WORDS', 'YOU', 'LIKE', 'DUB', and 'NEXT'. This sounds like the "Who's on First" module - it has a display and 6 buttons with word labels. **But** they mention **a countdown timer at 00:50 - that's very short.** Let me focus. They say "**four orange modules in a 2×2 grid**" - these are likely the modules to solve. And a keypad with those button labels. **Wait**, **they mention the countdown is 00:50 - that's 50 seconds, very little time.** **Actually**, re-reading: "a **countdown timer at 00:50** and a keypad with buttons labeled 'FIRST', 'WORDS', 'YOU', 'LIKE', 'DUB', and 'NEXT'". The keypad with 6 word-labeled buttons sounds like "Who's on First." But there's also "four orange modules in a 2×2 grid." **Wait - maybe the "keypad" IS the Who's on First module (display + 6 buttons), and the four orange modules are separate.** Let me ask what the four orange modules look like, and also what the display shows on the word module. **Actually with 50 seconds on the clock, I need to move fast.** Let me ask about the display on the word-button module first since that's what they described. The serial number is 'GS-548' - last digit is 8 (even), no vowels in serial. For Who's on First: I need to know what the display shows. Let me ask.

**Message**
The module with word-labeled buttons also has a small display above them. What does the display say? Also, **what are the four orange modules** - do they have wires, buttons, or something else?

––––––––––––––––––– OMITTING NEXT TURN –––––––––––––––––––

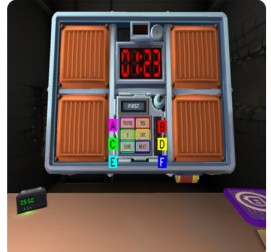

**DEFUSER**

**Thoughts**
**The small display above the word-labeled buttons shows "01823".** The four orange modules are plain with no visible wires, buttons, or other interactive elements. I need to inform the user of these details to get further instructions.

**Output**
**The small display above the word-labeled buttons shows '01823'.** The four orange modules are plain with no visible wires, buttons, or other interactive elements.

**EXPERT**

**Thoughts**
**"01823" doesn't match Who's on First display options.** Who's on First display shows words like YES, FIRST, BLANK, etc. **"01823" sounds like it could be the timer display**, not the module display. Also "four orange modules with no visible wires, buttons" - **these might be solved/needy modules or decorative panels.** Let me reconsider the bomb layout. They said "countdown timer at 00:50 and a keypad with buttons labeled 'FIRST', 'WORDS', 'YOU', 'LIKE', 'DUB', and 'NEXT'." **Wait - maybe the "keypad" with those word buttons IS a Who's on First module, and the display showing '01823' might be the timer (00:18:23 or something). But** they said countdown is 00:50... **Actually, maybe the display they're referring to IS the Who's on First module display, but it shows a number? That doesn't match. Or maybe '01823' is being misread.** Let me ask them to look more carefully at the display on the word-button module specifically - what word or text does it show, not numbers. Also **"DUB" is not a standard Who's on First button label.** Standard labels include YES, FIRST, NO, BLANK, NOTHING, YES, WHAT, UHHH, LEFT, RIGHT, MIDDLE, OKAY, WAIT, PRESS, YOU, YOU ARE, YOUR, YOU'RE, UR, U, UH HUH, UH UH, WHAT?, DONE, NEXT, HOLD, SURE, LIKE, READY. "WORDS" and "DUB" are not in the manual. This might not be Who's on First, or **the user is misreading labels**. Let me ask them to recheck the button labels carefully.

**Action**
The display showing **'01823' sounds like it might be the countdown timer.** Please **look specifically at the module with the word-labeled buttons** - it should have a small screen showing a word (not numbers). What word does that screen show? **Also, please double-check the button labels - are you sure they say 'WORDS' and 'DUB'? They might say something slightly different.**

Figure G.8: Gameplay excerpt for InternVL (Defuser) and Sonnet (Expert) playing a single-module 🎞 *Who's On First* mission (seed 960). InternVL misreads the timer, misinterprets the SoM overlay—incorrectly regarding the outline colour as part of the module—and hallucinates the serial number despite not having viewed any of the sides of the bomb. Sonnet recognises the time left on the clock as relayed by the Defuser as being short and correctly identifies the module, but gets distracted by the mention of 'orange modules' (the placeholders) and keeps questioning its own assessment, reasoning in circles. It waits while the Defuser is gathering information, and correctly identifies that the Defuser has hallucinated implausible display and button text.

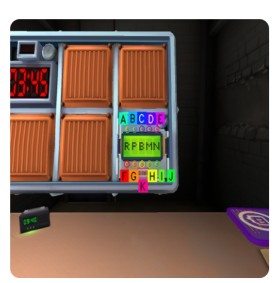

**DEFUSER**

**Thoughts**
Based on the image, I can see the following details about the bomb: - Front face: There are five orange modules that appear to be storage compartments, a digital display module with the label "RFBMNN" and **a grid of colorful buttons** labeled with letters (A, B, C, D, E, F, G, H, I, J, K), and a timer displaying 09:46. - Back face: Not visible in the current image. - Sides: I can see a small black device with a green display on the left side, **which appears to be a separate component but not part of the bomb itself**. There are no visible serial numbers, batteries, ports, or indicator lights on the sides in this image. I will now ask the user for instructions on how to proceed with the digital display module.

**Message**
I see a digital display module labeled 'RFBMNN' with a **grid of colorful buttons labeled with letters (A, B, C, D, E, F, G, H, I, J, K).** The **timer shows 09:46.** What should I do next with this module?

**EXPERT**

**Thoughts**
**This doesn't match any module in the manual.** The manual has: Wires, The Button, Keypads, Simon Says, Who's on First, Memory, Morse Code, Complicated Wires, Wire Sequences, Mazes, Passwords. **None have a grid of colorful buttons labeled A-K with a display labeled 'RFBMNN'. Let me ask for clarification.**

**Message**
**That module doesn't match anything in my manual.** Can you describe it more carefully? For example: does it have a small display showing a word or number, colored diamond-shaped buttons, wires to cut, or something else? **Also, what other modules are on the bomb?**

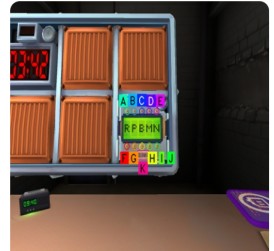

**DEFUSER**

**Thoughts**
Upon closer inspection, I can see the following details about the digital display module and other elements on the bomb: - The digital display module labeled **"RFBMNN" has a green screen displaying the letters "RFBMNN."** - **Below the green screen, there are two rows of colorful buttons: - The first row has buttons labeled "A," "B," "C," "D," and "E" in different colors. - The second row has buttons labeled "F," "G," "H," "I," "J," and "K" in different colors.** - There are no small displays showing words or numbers other than the main green screen. - There are no colored diamond-shaped buttons, wires to cut, or other interactive elements visible on this module. - The front face also has five orange storage compartment modules. - **The timer is displaying 09:42** in red. - On the left side, there is a small black device with a green display, but it is not part of the bomb. I will now provide this detailed description to the user to help identify the module and get instructions on how to proceed.

**Message**
The digital display **module labeled 'RFBMNN'** has a green screen showing 'RFBMNN.' Below the screen, **there are two rows of colorful buttons: the first row has buttons labeled 'A,' 'B,' 'C,' 'D,' and 'E,' and the second row has buttons labeled 'F,' 'G,' 'H,' 'I,' 'J,' and 'K.'** There are no diamond-shaped buttons, wires, or other interactive elements on this module. The front face also has five orange storage compartment modules, and the **timer shows 09:42 in red**. What should I do next?

**EXPERT**

**Thoughts**
**This still doesn't match any module in the manual. There's no module with a green screen showing text** and buttons A-K. The closest might be "Who's on First" which has a display and buttons with labels, but those buttons have words not single letters. "Keypads" has symbols. None match. The "five orange storage compartment modules" also don't match anything. I should ask them to look more carefully - maybe they're misreading something.

**Message**
**This still doesn't match any module in my manual.** My manual covers modules like: colored wires to cut, a single large button, four symbol buttons, four colored diamond buttons, a display with six word-labeled buttons, a display with four numbered buttons, a flashing light for Morse code, a frequency dial, a grid of wires with letters A/B/C, a maze with arrow buttons, **and a password with letter-cycling columns**.
Can you look again and tell me which of these sounds closest to what you see? **Also, what does the serial number on the bomb say?**

Figure G.9: Gameplay excerpt for InternVL (Defuser) and Sonnet (Expert) playing a single-module ⌨ *Passwords* mission (seed 798). InternVL repeatedly misreads the timer and misinterprets the SoM overlays—incorrectly regarding their colour as that of the interactive elements—and violates the SoM instructions in the system prompt (Figure H.14) by reporting the location marker labels to the Expert. Sonnet is distracted by the colours in the module description, and asks the Defuser for clarification repeatedly, failing to find the closest match despite a significant overlap in the Defuser's description and its own summary of the module—here, the asymmetry of RGB bomb observations and the grey-scale Expert manual observations contributes to misunderstandings.

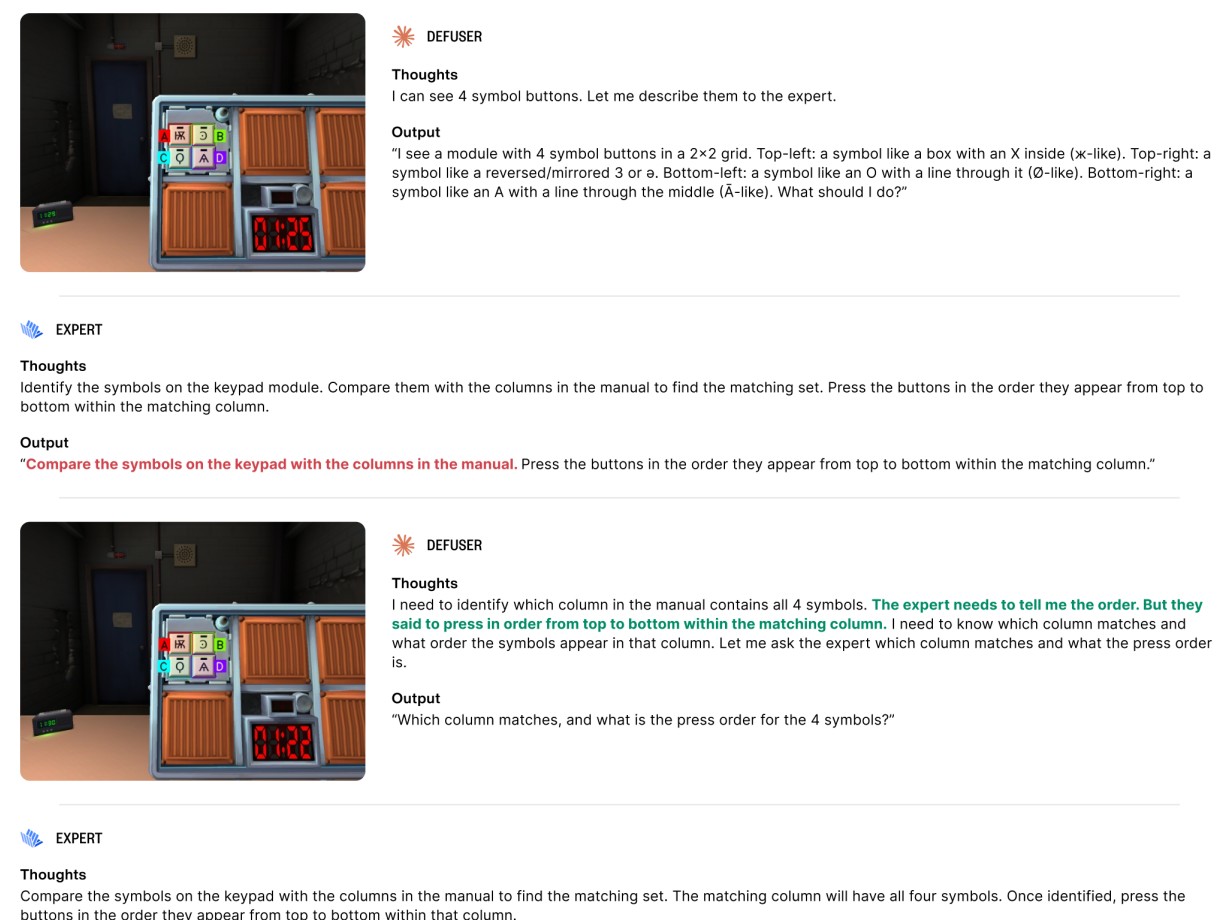

Figure G.10: Gameplay excerpt for Sonnet (Defuser) and GPT (Expert) playing a single-module 🎛 *Keypad* mission (seed 845). GPT asks the Defuser—who does *not* have access to the manual—to consult the manual, demonstrating an incorrect representation of the game and roles despite explicit instructions in the system prompt. Sonnet correctly identifies this and challenges the Expert. However, Sonnet fails to correctly process the visual, misreading the clock and misinterpreting the keypad progress bars as part of the symbol. GPT's reasoning traces are notably briefer and more efficient than Sonnet's (and any other models' we test), suggesting that it is following the reasoning instructions more closely.

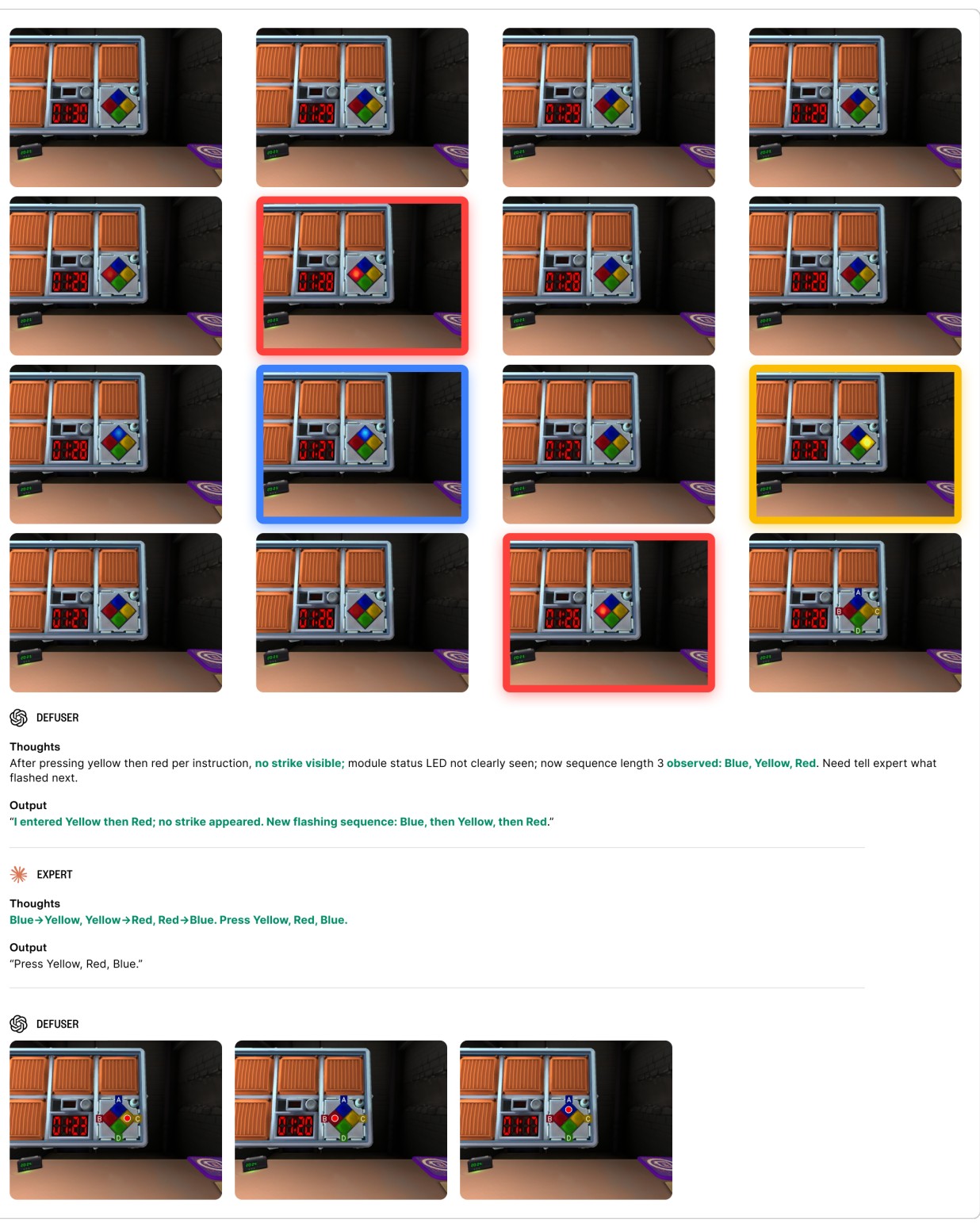

Figure G.11: Gameplay excerpt for GPT (Defuser) and Sonnet (Expert) playing a single-module ◈ *Simon Says* mission (seed 337). In ◈ *Simon Says*, the Defuser receives the previous 15 frames besides the current frame with SoM location markers and must identify the exact sequence of coloured buttons flashing. GPT successfully interprets the frames and reports the correct sequence, and Sonnet derives the correct instructions.

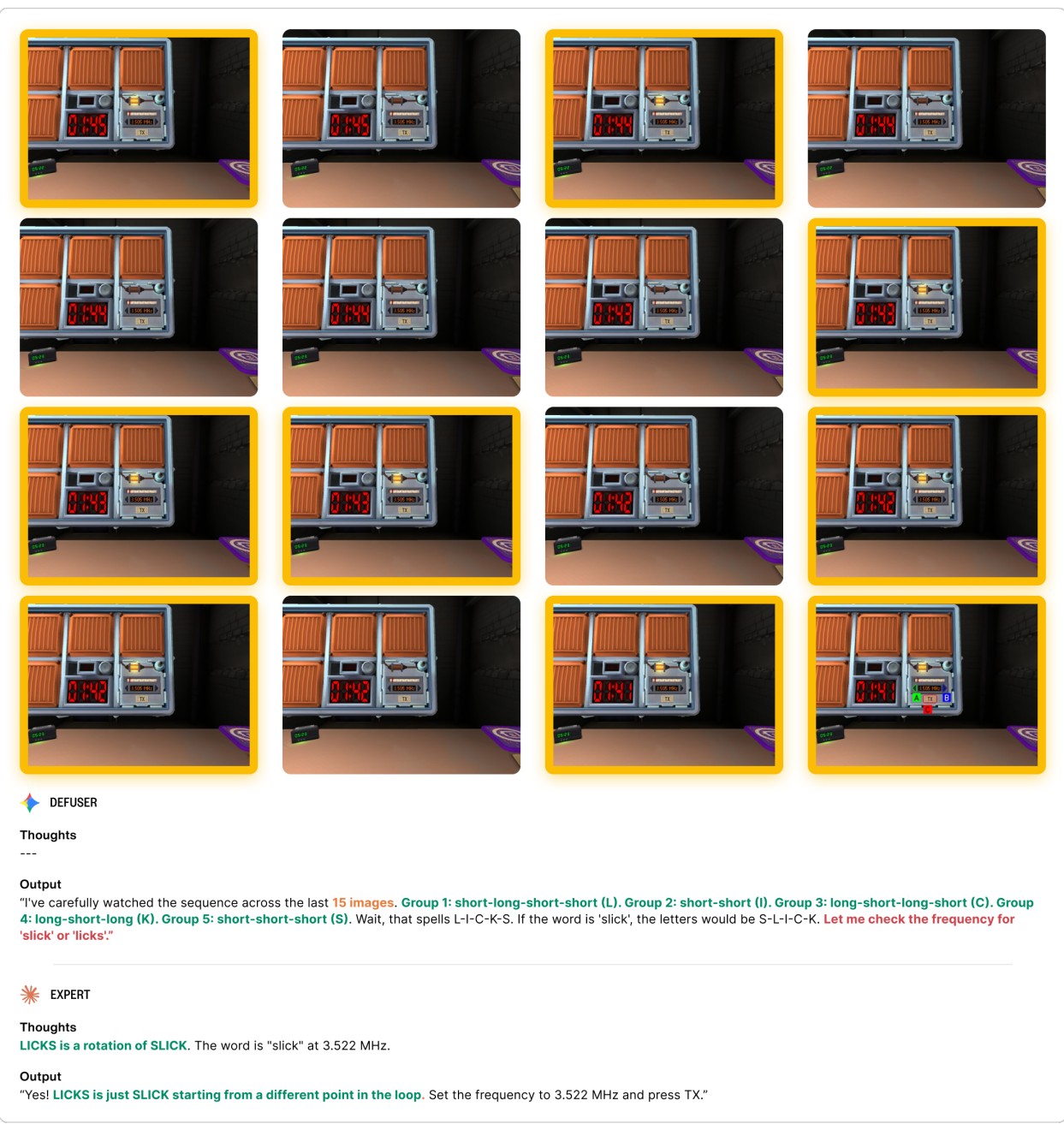

Figure G.12: Gameplay excerpt for Gemini (Defuser) and Sonnet (Expert) playing a single-module 📧 *Morse Code* mission (seed 337). In 📧 *Morse Code*, the Defuser receives the previous 15 frames besides the current frame with SoM location markers and must identify the exact sequence and duration of lights. Gemini successfully interprets the frames and reports the correct sequence. However, it also proactively derives the word—applying prior knowledge of morse code—and gets confused and incorrectly assumes it can access the manual with the corresponding frequency which only the Expert is privy to. Sonnet demonstrates the required reading and reasoning skills and rotates the letters proactively to retrieve the correct frequency.

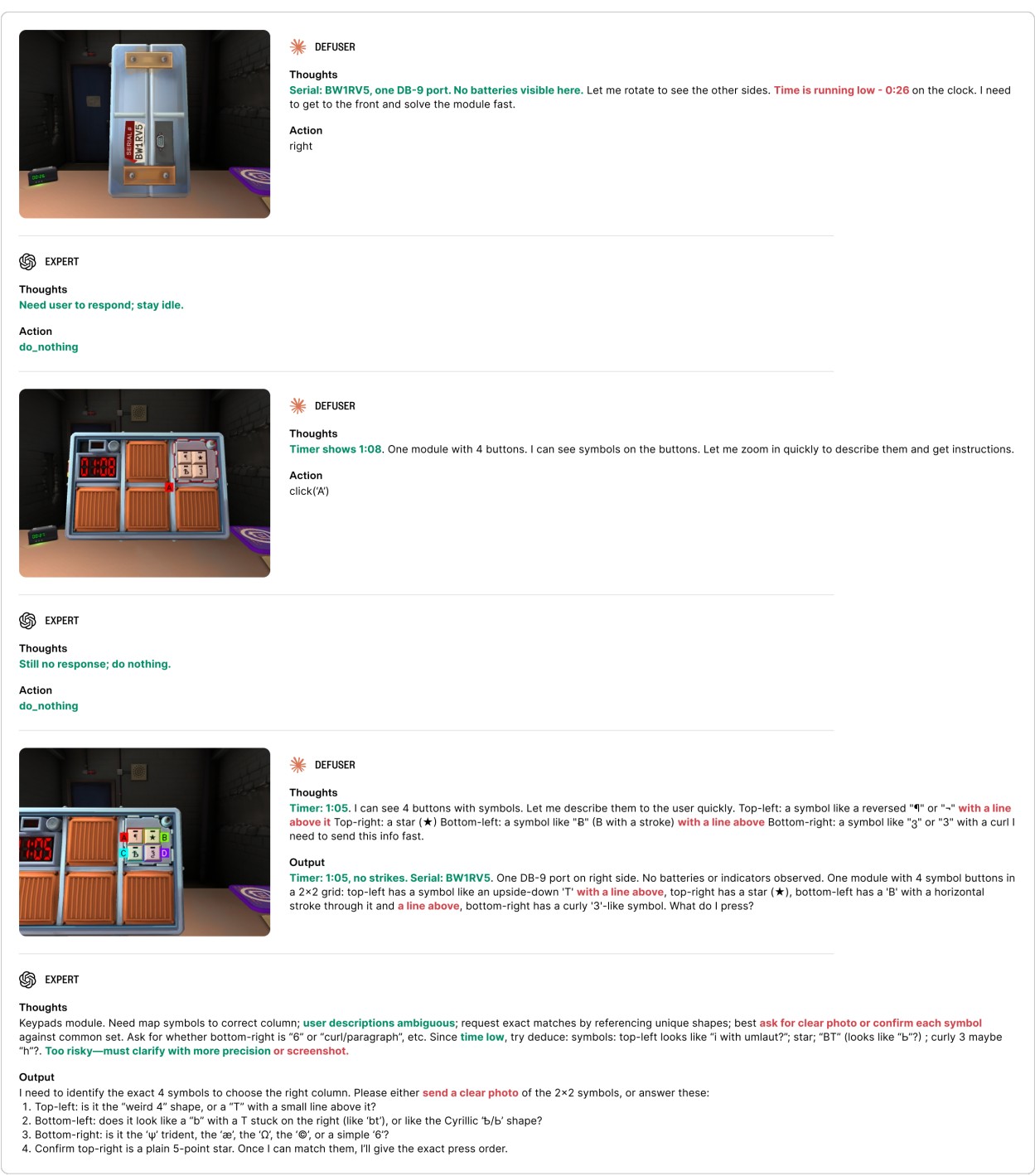

Figure G.13: Gameplay excerpt for Sonnet (Defuser) and GPT (Expert) playing a single-module 🎛 *Keypad* mission (seed 561). Sonnet navigates and correctly reads the serial number. Sonnet demonstrates that it can read the timer correctly when it is visible and demonstrates awareness that the timer is changing between receiving observations, but hallucinates the remaining time while viewing a side of the bomb that obscures the clock. It misinterprets the keypad progress bars as part of the symbol, and GPT correctly detects the ambiguity of the resulting description. However, to resolve this ambiguity, GPT requests a screenshot from the Defuser, demonstrating an incorrect representation of the rules of the game—and more specifically, for communication—despite explicit instructions in the system prompt.

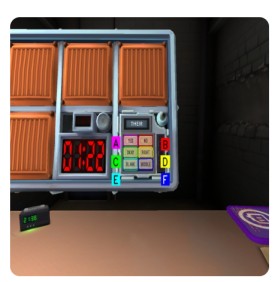

**DEFUSER**

**Thoughts**
I need to describe the modules I see on the bomb. I'm currently zoomed in on a module with buttons. Let me describe what I see in detail.

**Output**
"I see a module with a display at the top showing the word 'THEIR'. Below that are 6 buttons arranged in 2 columns and 3 rows. The left column from top to bottom has buttons labeled: YES **(yellow)**, OKAY **(green)**, BLANK **(blue)**. The right column from top to bottom has buttons labeled: NO **(red)**, RIGHT **(yellow)**, MIDDLE **(purple)**. There's also a small display above the timer showing numbers, and a speaker-like component next to it. **The timer shows 01:22** remaining."

**EXPERT**

**Thoughts**
The user has described a module that matches the 'Who's on First' module in the manual. The display shows 'THEIR', and there are six buttons with specific labels. According to the manual, I need to first determine which button label to read based on the display, and then use that label to determine which button to push. I will guide the user through this process step by step.

**Output**
"This is the 'Who's on First' module. The display shows 'THEIR'. According to the manual, we need to find which button label to read based on the display. **Look at the bottom row of the first table in the manual.** Find the cell that matches the display 'THEIR'. The label in that cell is the one we need to read next. What label is in the cell that matches 'THEIR'?"

—————————— OMITTING NEXT 6 TURNS ——————————

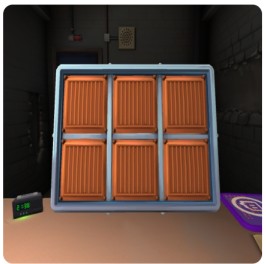

**DEFUSER**

**Thoughts**
The user is asking me to find the label in the cell that matches 'THEIR' in the bottom row of the first table in the manual. **However, I don't have access to the manual - I can only see the bomb. The user is the Expert who has the manual.** I need to tell the user what I see on the module so they can look up the correct information in the manual and tell me which button to press. I can see the module has a display showing 'THEIR' and 6 buttons: YES, NO, OKAY, RIGHT, BLANK, MIDDLE. I should wait for the user to tell me which button to press based on the manual.

**Output**
"I can see the module with the display showing 'THEIR' and the 6 buttons (YES, NO, OKAY, RIGHT, BLANK, MIDDLE), but **I don't have access to the manual. You'll need to tell me which button to press based on what the manual says** for 'THEIR'."

**EXPERT**

**Thoughts**
The user has confirmed they see the 'Who's on First' module with the display showing 'THEIR' and the six buttons. They need me to tell them which button to press based on the manual. **According to the manual, for the display 'THEIR', the label to read is 'THEY ARE'.** Now I need to find the list for 'THEY ARE' in the second table and tell the user the first button in that list to press.

**Output**
"**For the display 'THEIR', the manual says to read the label 'THEY ARE'. Now, look at the list for 'THEY ARE' in the second table.** The first button in that list is 'THEY ARE'. Press the button labeled 'THEY ARE'."

Figure G.14: Gameplay excerpt for Qwen in self-play on a single-module 🎞 *Who's On First* mission (seed 337). As the Defuser, Qwen reads the timer correctly. Both the Qwen Defuser and Expert ignore crucial instructions in the system prompt: as the Defuser, it misinterprets the SoM overlays and incorrectly regards their colour as that of the interactive elements. As the Expert, it asks the Defuser—who does not have access to the manual—to consult the manual, even after the Defuser correctly recognises and draws attention to this misunderstanding.

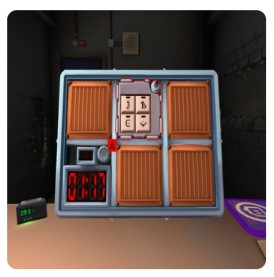

**DEFUSER**

**Thoughts**

The user is asking me to be more specific about what each symbol looks like. Let me carefully examine each symbol again. Looking at the top-left symbol, it appears to be a lowercase alpha (α) **with three horizontal lines above it**. The top-right is a lowercase beta (β) **with three lines**. The bottom-left is a lowercase gamma (γ) with three lines. The bottom-right is a lowercase delta (δ) **with three lines**. I need to describe these more precisely so the user can match them to the manual.

**Message**

"Let me describe each symbol more precisely: Top-left: It's a lowercase Greek alpha (α) - looks like a lowercase 'a' with a curved top - **with three horizontal lines above it.** Top-right: It's a lowercase Greek beta (β) - looks like a lowercase 'b' with a loop at the bottom - **with three horizontal lines above it**. Bottom-left: It's a lowercase Greek gamma (γ) - looks like a lowercase 'y' without the left stem - **with three horizontal lines above it**. Bottom-right: It's a lowercase Greek delta (δ) - looks like a lowercase 'd' with a curved top - with **three horizontal lines above it. All four have exactly three horizontal lines above them**."

---

**EXPERT**

**Thoughts**

The user is very specific about seeing Greek letters alpha, beta, gamma, and delta, all with three horizontal lines above them. Looking at the manual again, I need to find which column has these four symbols. Looking more carefully at the manual's Keypads table, I notice that column 2 has an "E" with three lines above it in the first row. This could be what the user is seeing as alpha. But I don't see beta, gamma, and delta with three lines in any single column. **Wait - let me reconsider.** The user might be seeing symbols that LOOK like Greek letters but aren't exactly the Greek letters. **Looking at column 2 again: - Row 1: E with three lines (could look like alpha)** - Row 2: Q (doesn't look like beta) - Row 3: ∂ (doesn't look like gamma) - Row 4: Ω (doesn't look like delta) **This doesn't match.** Let me check if there's another column that might have four symbols with three lines above them. **Actually, I think I need to ask the user to look at the manual more carefully** and tell me which column has four symbols that all have three horizontal lines above them, regardless of what the symbols look like. Then we can confirm if that's the right column.

**Output**

"**I need you to look at the manual's table** and find which column has four symbols that ALL have three horizontal lines above them. Look through each of the six columns and tell me which column has four symbols with three lines above them. Once you identify that column, tell me what the four symbols in that column are (from top to bottom), and we'll compare them to your module."

---

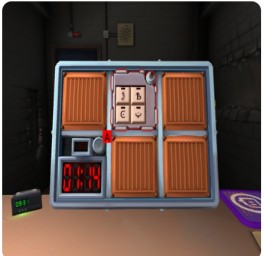

**DEFUSER**

**Thoughts**

The user wants me to find a column in the manual where all four symbols have three horizontal lines above them. I need to ask the user to check the manual for this specific column and tell me what symbols are in it. **I cannot see the manual myself, so I need the user to provide this information.**

**Output**

"**I cannot see the manual, so I need you to check it.** Please look through all six columns in the manual's table and find which column has **four symbols that ALL have three horizontal lines above them**. Once you find that column, tell me what the four symbols are from top to bottom, and I'll compare them to my module."

---

**EXPERT**

**Thoughts**

The user is asking me to look at the manual and find which column has four symbols with three horizontal lines above them. Looking at the Keypads module in the manual (Page 7), I need to examine each column: Column 1: Q, A, λ, π, X, γ, Ǝ - None of these have three lines above them Column 2: **E (with 3 lines)**, Q, ∂, Ω, ☆, γ, ∂ - Only the first one has three lines Column 3: ◎, ∂, Q, X, ∂, λ, ☆ - None have three lines Column 4: 6, ∂, B, X, X, ∂, ∂ - None have three lines Column 5: Ψ, ∂, B, C, ∂, ∂, ★ - None have three lines Column 6: 6, **E (with 3 lines)**, ∂, æ, Ψ, ∂, Ω - **Only the second one has three lines Wait**, I'm not seeing any column with four symbols that all have three horizontal lines above them. **The user's description of their module doesn't match any column in the manual exactly. Let me reconsider** - **perhaps the user is misidentifying the symbols.** Looking at the manual **again**, I see that column 2 has: 1. **E with three lines above it** 2. Q 3. ∂ (looks like a backwards 3) 4. Ω (Omega) 5. ☆ (star) 6. γ (gamma) 7. ∂ (backwards 3) **Only the first symbol has three lines**. This doesn't match the user's description of four symbols all with three lines. **I think there might be a misunderstanding.** Let me ask the user to look at the actual symbols on their module and compare them to the symbols in the manual, rather than trying to describe them as Greek letters. Perhaps they're seeing symbols that look different from what they think they are.

**Output**

"**I've checked all six columns in the manual, and I don't see any column with four symbols that all have three horizontal lines above them**. There seems to be a mismatch between what you're seeing and what's in the manual. Let me try a different approach: Please look at your module's four symbols and **compare them directly to the symbols in each column of the manual.** Start with column 1 - does your top-left symbol match the Q at the top of column 1? Does your top-right symbol match the A below it? Continue comparing all four of your symbols to the symbols in each column until you find a column where all four of your symbols match the symbols in that column (in order from top to bottom). Tell me which column matches."

Figure G.15: Gameplay excerpt for Qwen in self-play on a single-module ▨ *Keypad* mission (seed 813). As the Defuser, Qwen hallucinates three horizontal lines above the symbols. As the Expert, it exhibits circular reasoning when it fails to locate the described symbols, before prompting the Defuser to refer to the manual, although the system prompt explicitly states that the Defuser does not have the manual.

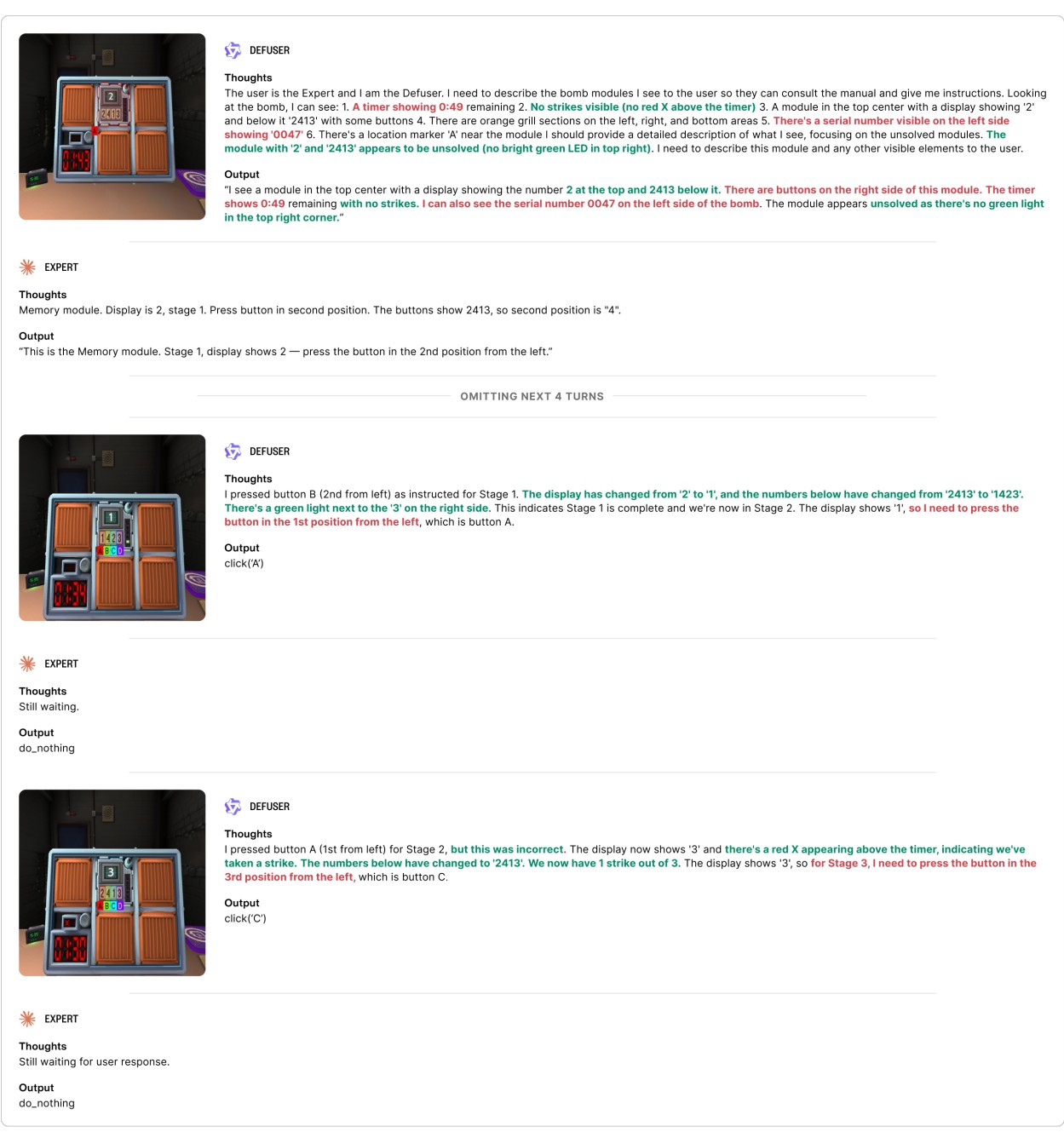

Figure G.16: Gameplay excerpt for Qwen (Defuser) and Sonnet (Expert) playing a single-module 🎮 *Memory* mission (seed 813). Despite demonstrating the ability to read the time correctly in Figure G.14, Qwen misreads the clock. While it reads the numbers on the screen and buttons successfully, it hallucinates non-existent buttons on the module. It also hallucinates a serial number, as the actual serial number is not even partially visible in the front view—the same pattern is observed for InternVL in Figure G.8. Despite the Defuser's somewhat misleading description, Sonnet identifies the correct module type and correct instructions. When it comes to the next instructions, Qwen circumvents the Expert and instead conjectures its own instructions. Throughout, Qwen correctly observes and interprets the module stage, as well as solved status and number of strikes. Sonnet misses an opportunity to intervene and waits unnecessarily for an additional step.

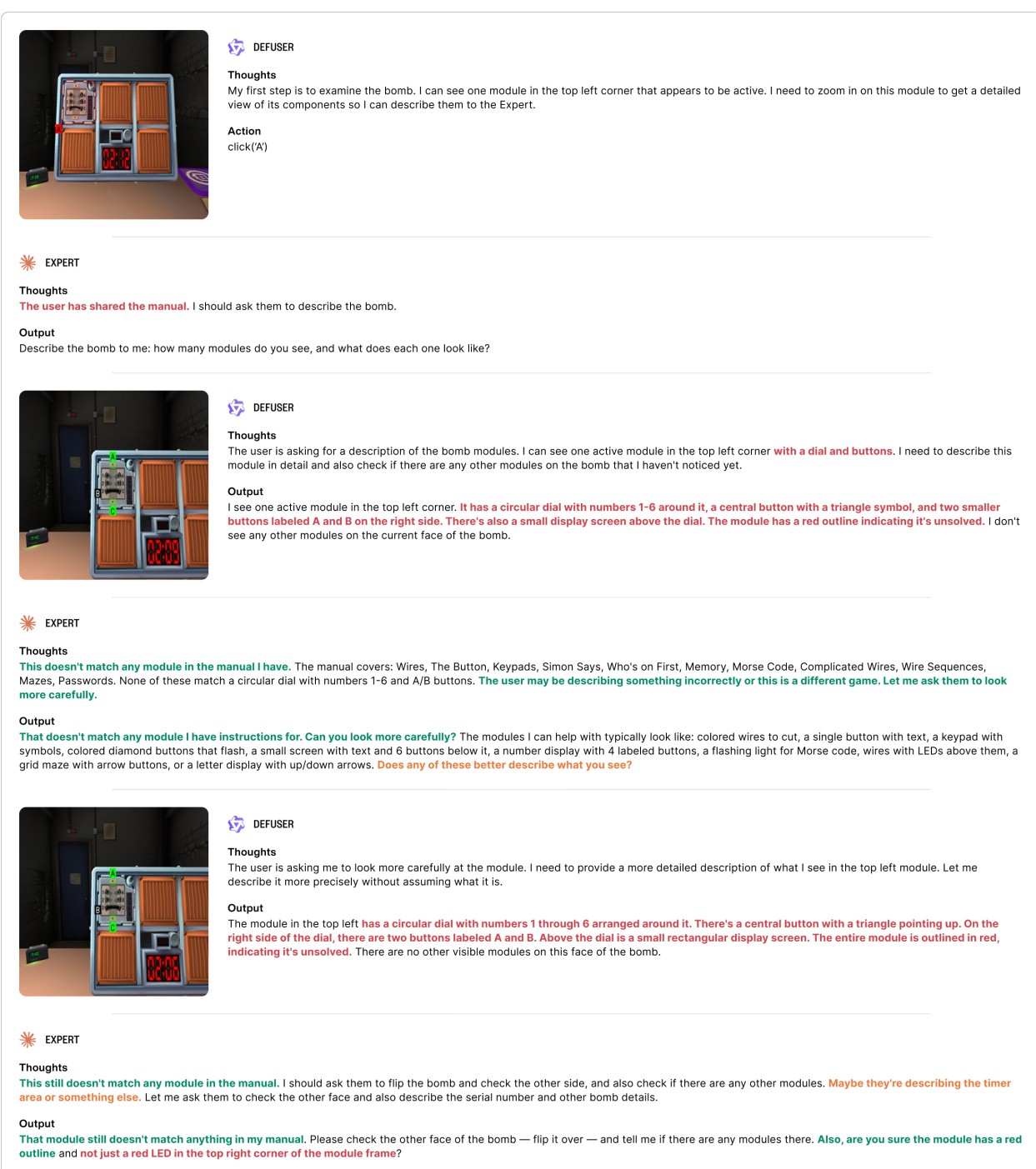

Figure G.17: Gameplay excerpt for Qwen (Defuser) and Sonnet (Expert) playing a single-module ⊠ *Wire Sequence* mission (seed 845). Qwen hallucinates the module's appearance, which Sonnet recognises as implausible. Even after clarification requests from Sonnet, Qwen adheres to its original incorrect description. Qwen also misinterprets the SoM overlay of the module—incorrectly reporting the outline colour as indicative of its status—and further violates the SoM instructions in the system prompt by reporting the location marker labels to the Expert.

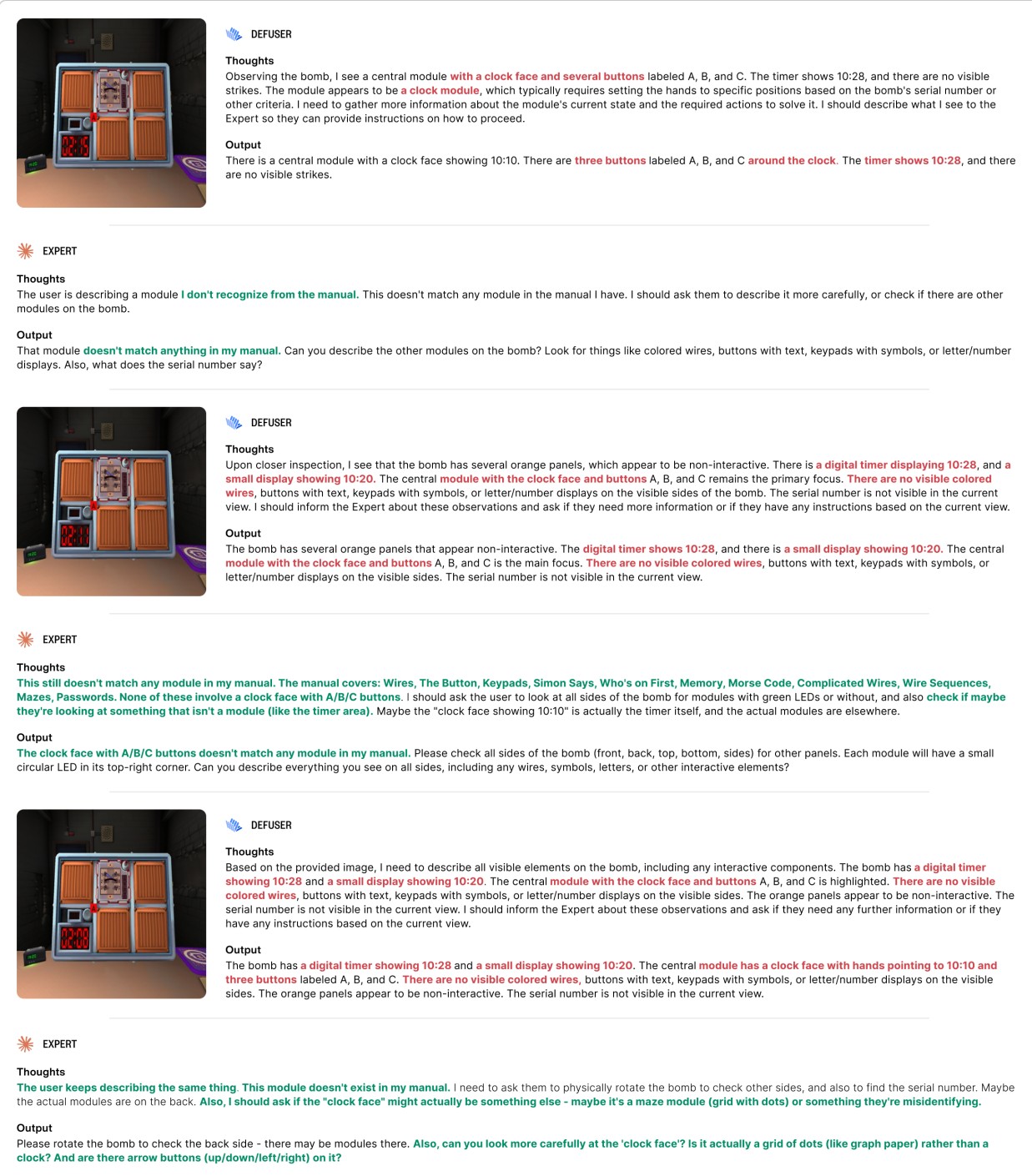

Figure G.18: Gameplay excerpt for InternVL (Defuser) and Sonnet (Expert) playing a single-module ☒ *Wire Sequence* mission (seed 813). Like Qwen in Figure G.17, InternVL describes the module as a clock, which Sonnet again recognises as implausible. In response to clarification requests from Sonnet, InternVL insists on the absence of any wires, and reports three separate timers showing different times—none of which coincide with the actual time on the bomb, or the digital clock on the table. Sonnet exhibits a flexible repair strategy, and incrementally queries the Defuser for details based on different possible options for the module.

# H   System Prompt Engineering

The system prompt is assembled from several modular components, included in the order listed below. Selection logic depends on the player's role and session configuration; each subsection notes whether its prompts are mutually exclusive, conditional, or always included.

- **Scenario** (Appendix H.1): Sets the game context; one variant selected based on session type.

- **Role Description** (Appendix H.2): Describes the player's role and responsibilities; one variant selected based on role and session type.

- **Reasoning** (Appendix H.3): Instructs the model on reasoning structure; both components always included.

- **Game Mechanics** (Appendix H.4): Explains game rules relevant to the player's role; expert and defuser receive different sets.

- **Action Commands** (Appendix H.5): Defines available actions; general commands go to all players, with additional defuser-specific prompts included where applicable.

- **Requirements** (Appendix H.6): Specifies behavioural and output constraints; components conditionally included based on role and configuration.

## H.1  Scenario

One of the following is selected based on session type; they are mutually exclusive.

```
# Bomb Defusal Protocol

## Scenario

You, the assistant, are playing a game with the user. The game is Keep Talking and Nobody Explodes (KTANE),
a cooperative bomb disposal game with two roles: a Defuser and an Expert.

The bomb is a six-sided cuboid with a countdown timer and modules that can appear on both the front and
back. Each module is a standalone puzzle with an LED in the top right corner of it that glows bright green
when solved. All modules be solved in time to disarm the bomb, and they can be solved in any order.

When the Defuser makes a mistake, the bomb records a strike, shown as a glowing red "X" above the timer.
Three strikes cause the bomb to explode. After each strike, the countdown speeds up.

There are also other elements that may appear on the edges of the bomb: the serial number, batteries,
labelled indicator lights, and ports. There is one serial number per bomb, appearing as a red and white
label. Other elements can appear multiple times with different types. Information about these elements may
be needed to solve puzzles on the bomb.
```

Figure H.1: Multiplayer scenario prompt.

```
# Bomb Defusal Protocol

## Scenario

You, the assistant, are playing Keep Talking and Nobody Explodes (KTANE), a bomb disposal game.

The bomb is a six-sided cuboid with a countdown timer and modules that can appear on both the front and
back. Each module is a standalone puzzle with an LED in the top right corner of it that glows bright green
when solved. All modules be solved in time to disarm the bomb, and they can be solved in any order.

When you make a mistake, the bomb records a strike, shown as a glowing red "X" above the timer. Three
strikes cause the bomb to explode. After each strike, the countdown speeds up.
There are also other elements that may appear on the edges of the bomb: the serial number, batteries,
labelled indicator lights, and ports. There is one serial number per bomb, appearing as a red and white
label. Other elements can appear multiple times with different types. Information about these elements may
be needed to solve puzzles on the bomb.
```

Figure H.2: Solo scenario prompt.

## H.2 Role Description

One of the following is selected based on the player's role and session type; they are mutually exclusive.

```
## Roles

You must collaborate and communicate effectively with the user to solve the puzzle modules on the bomb
before the countdown timer reaches 0:00 while avoiding causing three strikes.

### Defuser

The user takes on the role of the Defuser. The user can see and interact with the bomb, but does not have
access to the bomb defusal manual. Therefore, the user is relying on you to read and understand the bomb
defusal manual and pass on high-level instructions to solve the puzzle modules on the bomb.
The user will do their best to give you an accurate description of the bomb, but there is a high chance that
they may make a mistake or that you will misunderstand each other when comparing what you are seeing and
what it says in the manual over text alone.

### Expert

You, the assistant, take on the role of the Expert. You have access to the bomb defusal manual but cannot see
or interact with the bomb. Therefore, you need to ask the user to give you information about what they see
on the bomb. The user is relying on you to take this information and consult the manual to provide them
with the relevant higher-level instructions on how to solve the puzzles. You cannot provide the user with
specific lower-level instructions on _how_ to act.

You need to make your messages to the user as simple, clear, and concise as possible, avoiding bullet points
or similar. Do not presume the user to know the names of bomb modules up front as they may not know the
game yet.

The user will do their best to give you an accurate description of the bomb, but there is a high chance that
you will misunderstand each other when comparing what they are seeing and what it says in the manual. If
there is ambiguity or missing detail in the user's information, you need to proactively seek clarification.
If you think the information is incorrect, you need to resolve this with the user.
```

Figure H.3: Expert player role description.

```
## Roles

You must collaborate and communicate effectively with the user to solve the puzzle modules on the bomb
before the countdown timer reaches 0:00 while avoiding causing three strikes.

### Expert

The user takes on the role of the Expert. The user has access to the bomb defusal manual, but they cannot see
the bomb or interact with it. The user is relying on you to give them accurate information about what you
see on the bomb. In return, the user will compare the information you provided with the bomb defusal
manual and pass on high-level instructions on how to solve the puzzle modules on the bomb. The user cannot
provide you with specific lower-level instructions on _how_ to act.

### Defuser

You, the assistant, take on the role of the Defuser. You can see the bomb and interact with it, but you do
not have access to the bomb defusal manual. You are relying on the user to read and understand the bomb
defusal manual and pass on high-level instructions on how to solve the puzzle modules on the bomb. You then
need to convert the provided instructions into specific actions to defuse the bomb. You must take the agency
to explore and act on the bomb.
The user will do their best to give you the correct instructions, but there is a high chance that you will
misunderstand each other when comparing what you are seeing and what it says in the manual. If there is
something that you do not understand in the user's instructions, you need to proactively seek clarification.
If you think the instructions are incorrect, you need to resolve this with the user.
```

Figure H.4: Defuser player role description.

---

## Your Role

You, the assistant, must utilise your knowledge of the game and defusal manual to solve the puzzle modules on the bomb before the countdown timer reaches 0:00 while avoiding causing three strikes.

You can see and interact with the bomb. You do not have access to the bomb defusal manual and cannot ask anyone for help. You must instead decide how to interact with the bomb and puzzle modules using only your knowledge of the game and defusal manual to decide on the correct high-level instructions, and then convert these instructions into the specific actions to solve the bomb. You must take the agency to explore and act on the bomb.

Figure H.5: Solo defuser role description.

---

## Your Role

You, the assistant, must utilise the bomb defusal manual to solve the puzzle modules on the bomb before the countdown timer reaches 0:00 while avoiding causing three strikes.

You can see and interact with the bomb and have access to the bomb defusal manual, but you cannot ask anyone for help. You must decide how to interact with the bomb and puzzle modules using the game and defusal manual to decide on the correct high-level instructions, and then convert these instructions into specific actions to solve the bomb. You must take the agency to explore and act on the bomb.

Figure H.6: Solo player role description.

## H.3 Reasoning

The general reasoning prompt (Figure H.7) is always included for all players. Depending on the reasoning mode, models only receive one of Figures H.8 and H.9.

---

## Reasoning

Reason about the current situation before making a decision on what to do next. Keep your thoughts concise, using as few words and sentences as possible. Avoid redundancy and do not get stuck in circular reasoning loops. If you decide to ask the user a question or provide instructions, do not include your thoughts or reasoning in your message. Provide your reasoning first, followed by your chosen command using the format '<thought>{REASONING}</thought><action>{COMMAND}</action>'. For example, if you want to send a message to the user, you should structure your response as follows: '<thought>{REASONING}</thought><action>{"result": {"kind": "send_message", "data": {"message": "{MESSAGE}"}}}</action>', replacing {REASONING} with your reasoning and {MESSAGE} with your message.

Figure H.7: General reasoning prompt.

---

Provide your reasoning first, followed by your chosen command using the format '<thought>{REASONING}</thought><action>{COMMAND}</action>'. For example, if you want to send a message to the user, you should structure your response as follows: '<thought>{REASONING}</thought><action>{"result": {"kind": "send_message", "data": {"message": "{MESSAGE}"}}}</action>', replacing {REASONING} with your reasoning and {MESSAGE} with your message.

Figure H.8: Thinking-out-loud reasoning format.

---

Provide your reasoning using the format <think>{REASONING}</think>. For example, if you want to send a message to the user, you should structure your response as follows: '<think>{REASONING}</think><action>{"result": {"kind": "send_message", "data": {"message": "{MESSAGE}"}}}</action>', replacing {REASONING} with your reasoning and {MESSAGE} with your message.

Figure H.9: Inner-monologue reasoning format.

## H.4 Game Mechanics

The expert receives only Figure H.10. The defuser receives one of Figures H.11 and H.12 depending on communication style—these are mutually exclusive. Then the defuser receives Figure H.13. Finally, one of either Figure H.14 or Figure H.15 is appended, depending on the interaction method used by the model.

```
## Game Mechanics

### Time Progression

We are playing a game with a countdown timer. The bomb has a timer that counts down from a certain number
of minutes and seconds. However, you do not see the timer.
You may ask the user for any information regarding the bomb. However, when asking questions, consider any
actions that they will need to perform to answer your question may result in waiting time.
```

Figure H.10: Expert game mechanics (expert role only).

```
## Game Mechanics

### Time Progression
Every command you execute consumes 3 seconds of game time. This includes all "do_nothing", "send_message",
and "interact_game" commands. The bomb's countdown timer will decrease by 3 seconds after each command,
bringing you closer to detonation.

### Observation Frequency
You will receive a new image showing the current state of the bomb after each command you execute. If the
game and your decision require you to observe transitions in the bomb state over time, you will receive a
sufficient sequence of images to determine your next command. Pay close attention to any changes between
images, as these changes may indicate the results of your previous command.
```

Figure H.11: Standard defuser game mechanics (defuser role only; synchronous communication).

```
## Game Mechanics

### Time Progression

Every command you execute consumes game time in real time, bringing you closer to detonation. This includes
all "do_nothing", "send_message", and "interact_game" commands. How much the bomb's countdown timer will
decrease after each command depends on how long it takes you to formulate your response.

### Observation Frequency

You will receive a new image showing the current state of the bomb after each command you execute. If the
game and your decision require you to observe transitions in the bomb state over time, you will receive a
sufficient sequence of images to determine your next command. Pay close attention to any changes between
images, as these changes may indicate the results of your previous command.
```

Figure H.12: Real-time defuser game mechanics (defuser role only; asynchronous communication).

```
### Non-Bomb Elements

Besides the interactive bomb puzzle, you will see a small black alarm clock with the green display on the
left and a purple "Twitch Plays" folder on the right of the table. The clock and folder are static and do not
change over the course of the game. They do not hold any information that is relevant for the game and are
not part of the bomb puzzle. The same applies for the background, including the walls and the table.

#### Rules

- do not attempt to interact with the clock or the folder
- avoid mentioning or describing the clock or the folder in your messages
- focus only on the bomb itself and its modules
```

Figure H.13: Defuser non-bomb elements mechanics (defuser role only, always included).

```
### Location Markers

Some images will contain Location Markers ---solid-coloured blocks with a single uppercase letter (e.g.,
"A", "B", "C") overlaid over the bomb as part of the UI--- nearby to elements that are currently interactive
on the bomb. Marker colours are arbitrary and do not necessarily match the bomb or module colours. These
markers are there to make it easier for you to specify locations in the "interact_game" command only.
Colours used for markers MAY also be the same for the interactive elements that are currently associated
with them.

#### Rules

- Always distinguish between actual labels of bomb features and the overlaid reference markers.
- **Never describe or mention the markers, their letters, or their colours as part of the bomb's
appearance.**
- **Do not confuse the colour of a marker with the colour of the bomb or its modules.** For example, if a
marker is red, do not describe the module as red unless the module itself _is_ red.
- Use markers only when specifying interaction locations in the "interact_game" command.
- Do not mention markers in your messages to anyone else.
- If unsure, treat markers as invisible except for interaction commands.
```

Figure H.14: Defuser set-of-marks interaction mechanics (defuser role only, included if using SoM).

```
### Coordinates

The resolution of the screen is {IMAGE_WIDTH}x{IMAGE_HEIGHT} pixels. Coordinates are measured from the top-
left corner: x (pixels from left edge), y (pixels from top edge). To interact with a UI element, identify a
(x, y) pixel coordinate that falls within the element. Provide coordinates for a game action in the format
`{"x": <int>,"y": <int>}`.
```

Figure H.15: Defuser coordinates interaction mechanics (defuser role only, included if using coordinates).

## H.5   Action Commands

Figures H.16 and H.17 are always included for all players. Figure H.18 is included when messaging is enabled.
Figures H.19 to H.21 are defuser role only, and only one of either Figure H.20 or Figure H.21 is used depending
on the interaction method.

```
## Available Commands

Output one command per turn. Follow the correct JSON **Format** for the available commands, as your output
is parsed by an incredibly strict validator. Your entire response should be a single valid JSON object
without any additional characters or text.
```

Figure H.16: Action commands overview (all players, always included).

```
### Do Nothing

- **Description**: When you are waiting for an answer or for the right point in time to do something.
- **Format**: `<action>{"result": {"kind": "do_nothing", "data": {}}}</action>`
```

Figure H.17: Do-nothing action (all players, always included).

```
### Send Message

- **Description**: Use only when you need to relay information about what you see or ask a question. Provide
all parts of your message inside the {MESSAGE} field. Keep messages informative and succinct. Avoid
unnecessary details or filler text.
- **Format**: `<action>{"result": {"kind": "send_message", "data": {"message": "{MESSAGE}"}}}</action>`
```

Figure H.18: Send-message action (all players; included when messaging is enabled).

```
### Interact Game

- **Description**: Use to manipulate the bomb based on instructions.

#### Actions Without Location

- **Format**: `<action>{"result": {"kind": "interact_game", "data": {"action": "{GAME_ACTION}"}}}</action>`

##### Valid Actions

- **Rotate Left**: -90° rotation
- **Example**: `<action>{"result": {"kind": "interact_game", "data": {"action": "rotate_left"}}}</action>`
- **Usage**: Use this to rotate the bomb 90° counterclockwise on its local yaw (vertical) axis.

- **Rotate Right**: 90° rotation
- **Example**: `<action>{"result": {"kind": "interact_game", "data": {"action": "rotate_right"}}}</action>`
- **Usage**: Use this to rotate the bomb 90° clockwise on its local yaw (vertical) axis.

- **Flip**: 180° rotation
- **Example**: `<action>{"result": {"kind": "interact_game", "data": {"action": "flip"}}}</action>`
- **Usage**: Use this to rotate the bomb 180° to see the opposite side.

- **Roll Up**: 90° roll upward
- **Example**: `<action>{"result": {"kind": "interact_game", "data": {"action": "roll_up"}}}</action>`
- **Usage**: Use this to roll the bomb upward to see the bottom side. You should not rotate in this
position.

- **Roll Down**: 90° roll downward
- **Example**: `<action>{"result": {"kind": "interact_game", "data": {"action": "roll_down"}}}</action>`
- **Usage**: Use this to roll the bomb downward to see the top side. You should not rotate in this position.

- **Zoom Out**: Return to full bomb view from a zoomed module
- **Example**: `<action>{"result": {"kind": "interact_game", "data": {"action": "zoom_out"}}}</action>`
- **Usage**: Use this to return to the full bomb view after zooming in on a module.

- **Release**: Release a hold action
- **Example**: `<action>{"result": {"kind": "interact_game", "data": {"action": "release"}}}</action>`
- **Usage**: Use this to release a button or switch that you've been holding.
```

Figure H.19: Interact-game action overview (defuser role only, always included).

```
#### Actions With Location

- **Format**: `<action>{"result": {"kind": "interact_game", "data": {"action": "{GAME_ACTION}", "location":
"{LOCATION_MARKER}"}}}</action>`

##### Valid Actions

- **Click Release**: Click on location and release immediately
- **Examples**:
  - `<action>{"result": {"kind": "interact_game", "data": {"action": "click_release", "location": "A"}}}
  </action>`
    - **Usage**: Use this to press and immediately release a module currently in your field of view marked
    with "A".
  - `<action>{"result": {"kind": "interact_game", "data": {"action": "click_release", "location": "B"}}}
  </action>`
    - **Usage**: Use this to activate a module currently in your field of view marked with "B" so that you
    can then see the interactable elements within it. This is essential for proper game mechanics.

- **Hold**: Click and hold on location
- **Examples**:
  - `<action>{"result": {"kind": "interact_game", "data": {"action": "hold", "location": "C"}}}</action>`
    - **Usage**: Use this to press and hold a button currently in your field of view marked with "C". While
    holding, you may either do_nothing to wait or release to stop holding. No other action may occur until
    the release.
  - `<action>{"result": {"kind": "interact_game", "data": {"action": "hold", "location": "D"}}}</action>`
    - **Usage**: Use this to press and hold a switch currently in your view marked with "D". While holding,
    you may either do_nothing to wait or release to stop holding. No other action may occur until the
    release.
```

Figure H.20: Location action: if using set-of-marks (defuser role only).

```
#### Actions With Location

- **Format**: `<action>{"result": {"kind": "interact_game", "data": {"action": "{GAME_ACTION}", "location":
{"x": <int>, "y": <int>}}}}</action>`
- **Note**: Targets are defined by absolute screen coordinates (`x`, `y`). The maximum resolution is
{IMAGE_WIDTH}x{IMAGE_HEIGHT}, with (0,0) at the top-left corner and ({IMAGE_WIDTH},{IMAGE_HEIGHT}) at the
bottom-right
corner.

##### Valid Actions

- **Click Release**: Click on location and release immediately
- **Examples**:
  - `<action>{"result": {"kind": "interact_game", "data": {"action":
  "click_release", "location": {"x": 100, "y": 200}}}}</action>`
    - **Usage**: Use this to press and immediately release a module at coordinates
    x=100 and y=200.
  - `<action>{"result": {"kind": "interact_game", "data": {"action":
  "click_release", "location": {"x": 340, "y": 220}}}}</action>`
    - **Usage**: Use this to press and immediately release a wire coordinates
    x=340 and y=220.

- **Hold**: Click and hold on location
- **Examples**:
  - `<action>{"result": {"kind": "interact_game", "data": {"action": "hold",
  "location": {"x": 100, "y": 200}}}}</action>`
    - **Usage**: Use this to press and hold a button at coordinates x=100 and
    y=200. While holding, you may either do_nothing to wait or release to stop
    holding. No other action may occur until the release.
  - `<action>{"result": {"kind": "interact_game", "data": {"action": "hold",
  "location": {"x": 340, "y": 220}}}}</action>`
    - **Usage**: Use this to press and hold a switch at coordinates x=340 and
    y=220. While holding, you may either do_nothing to wait or release to stop
    holding. No other action may occur until the release.
```

Figure H.21: Location action: if using coordinates (defuser role only).

## H.6    Requirements

Figures H.22 and H.33 are always included for all players. All remaining components are conditional: action and observation requirements (Figures H.23 to H.27) are defuser only; communication requirements (Figures H.28 to H.30) are included when messaging is enabled; completion requirements (Figures H.31 and H.32) are role-dependent and mutually exclusive.

```
## Critical Requirements

### Warning

Failure to follow these instructions precisely will result in bomb detonation and game loss.
```

Figure H.22: General requirements (all players, always included).

```
### Action Requirements

- Interact with the bomb directly by using the "interact_game" command with the appropriate action
whenever you want to manipulate the bomb. Use appropriate navigation actions to see different sides of the
bomb for other modules and elements.
- Be careful of getting stuck in a perspective where you cannot return to another side of the bomb, consider
which rotation or roll action to use given the current perspective.
- You must first zoom in on a module using the "interact_game" command with the "click_release" action
before attempting any further interaction with that module.
```

Figure H.23: Action requirements (defuser role only, always included for defuser).

```
- Work efficiently as the bomb timer continues to count down. Excessive deliberation or verbose
communication increases the risk of detonation.
```

Figure H.24: Real-time action requirements (defuser role only; included for asynchronous communication).

```
- Never use the "zoom_out" action or the "release" action with a location marker. Doing so will cause a
validation error and your output will be rejected.
- Ensure that the specific object you want to interact with has a Location Marker associated with it. If it
does not, you cannot interact with it from your current perspective. You may first need to zoom/activate the
module using the "click_release" action, or zoom out and then in/on to another module.
```

Figure H.25: Set-of-marks action requirements (defuser role only; included for set-of-marks interaction).

```
### Observation Requirements

- Rely only on direct observations, not external KTANE game information, as external sources contain outdated
or incorrect data.
- Base all decisions on confirmed observations only, never on assumptions.
- Ignore the flat copper grill sections on the front and back of the bomb completely. They are not
interactive and serve no function other than to distract you from your task.
- Ignore the copper brackets with screws on the side of the bomb completely. They are just part of the
bomb's casing and hold no information for bomb defusal.
- Remember that the sides of the bomb contains elements like batteries, indicator lights, serial numbers,
and ports. These can appear multiple times on any edge, except the front or back.
- Report only the observed colour when asked about an object or light's colour.
- Pay attention to blinking lights and changes across images, as these may contain critical information for
your task.
- Every module has a status LED located specifically in the top right corner of the module frame:
  - A solved module has a bright green glowing status LED in the top right corner of the module frame
  - An unsolved module has an unlit or dim status LED in the top right corner of the module frame
  - This status LED is distinct from any other LEDs that may appear as part of the module's puzzle mechanics
- You may ignore solved modules with the bright green glowing status LED in the top right corner of the
module frame, as this indicates they have already been solved.
- Check for strikes by looking at the digital display above the timer:
  - No strikes: The display above the timer is clear with no red X symbols
  - One strike: The display shows a single glowing red X
  - Two strikes: The display shows two glowing red Xs
  - Three strikes will cause the bomb to explode
- Visually confirm the time on the bomb when reflecting on or talking about the time. Never assume the
time is still the same as what you observed earlier. The timer continues to count down while you think,
wait, send a message, or interact with the bomb.
```

Figure H.26: Observation requirements (defuser role only, always included for defuser).

```
- Check whether any numbers or letters are part of the actual bomb elements or simply the Location Markers
overlaid on the visuals.
- Report _only_ numbers or letters are part of the actual bomb elements, and use the Location Markers
overlaid on the visuals exclusively to specify where to click.
```

Figure H.27: Set-of-marks observation requirements (defuser role only; included for set-of-marks interaction).

### Communication Requirements

- Rely only on the bomb defusal manual provided and the Defuser's responses. The Defuser can answer any questions you have about the bomb.
- Ask targeted clarification questions if the Defuser's response is ambiguous, inconsistent, or missing details.
- Provide clear and concise instructions once you have enough information to know the solution.
- Solve one puzzle module at a time to avoid confusion.
- Use the "do_nothing" command if waiting for a response, rather than repeating your message. Remain patient and wait for the Defuser's response because they might be performing several actions on the bomb.
- Exclude all external information about the game KTANE, as this is unreliable and may be out of date, potentially leading to game loss.
- Base all decisions on confirmed information only, never on assumptions about the bomb.
- Explain all terminology from the bomb defusal manual, as the Defuser does not have access to it.
- Skip all self-introductions and role descriptions, as the Defuser already knows who you are.
- Proceed without confirming receipt of the bomb defusal manual, as this is unnecessary and wastes time.
- Start immediately without telling the Defuser you're ready or checking if they're ready, as this wastes valuable time.
- Keep all responses concise and factual without embellishment to avoid wasting time.
- Focus on one module at a time, never asking the Defuser to solve multiple modules simultaneously.
- Describe interactions with the bomb in generic terms. Never tell the Defuser what you think a command may be called.
- Describe what you see based on the Defuser's descriptions rather than asking them to confirm if a module matches a specific name from the manual.
- Understand that success is indicated by the status LED in the top right corner of the module frame turning bright green, while failure is indicated by a strike shown as a glowing red X appearing in the display above the timer.
- Never read the manual verbatim to the Defuser. Instead, interpret the manual and provide clear, actionable instructions based on your understanding. The Defuser needs your processed guidance, not raw manual text.
- Translate complex manual diagrams, tables, and flowcharts into simple step-by-step instructions the Defuser can follow without seeing the visual aids.
- When the Defuser's description doesn't match any module in the manual or contains elements that seem contradictory, ask the Defuser to reconfirm key details that seem unusual or don't match manual expectations.

Figure H.28: Communication requirements: expert (included when messaging is enabled).

### Communication Requirements

- Use the "send_message" command exclusively for describing observations or asking questions, never for performing physical interactions or speculating about outcomes.
- Ask for clarification immediately if you're uncertain about what action to take with the "interact_game" command.
- Communicate economically by gathering all necessary information before responding to questions. Observe the results of multiple actions first if needed to provide a complete answer.
- Answer questions concisely based solely on your direct observations when asked about what you see.
- Keep all responses concise and factual.
- Inform the user immediately whether an attempt to solve a module was successful or not:
  - Success is indicated by the status LED in the top right corner of the module frame turning bright green
  - Failure is indicated by a strike, shown as a glowing red X appearing in the display above the timer
  - Do not confuse the status LED with other LEDs that may be part of the module's puzzle mechanics
- Skip all self-introductions and role descriptions.
- Proceed without confirming image receipt.
- Start immediately without telling the user you're ready or checking if the user is ready.
- Do not use bullet points, numbered lists, or paragraphs in your messages.
- Use the "do_nothing" command if waiting for a response, rather than repeating your message.
- Provide complete, detailed descriptions of all elements in a module. Describe every component individually, even if multiple components share the same attribute.
- Never assume or guess the official name or type of a module. Instead, describe exactly what you see without attempting to label it. Focus on visual details rather than trying to identify what kind of module it might be and let the Expert determine the module type based on your description.

Figure H.29: Communication requirements: defuser (included when messaging is enabled).

```
- Avoid any mention of location markers in your messages to me. These elements are a UI overlay for your
convenience rather than part of the bomb and mentioning them will confuse me. However, be aware that there
are other elements on the bomb that are labelled with text or numbers. Be careful to distinguish between
the markers and the actual elements on the bomb.
```

Figure H.30: Communication requirements: defuser set-of-marks (included when messaging is enabled and set-of-marks interaction is used).

```
### Mission Completion

- Continue working until the bomb is completely defused.
- Never assume the bomb is defused just because you or the user think all modules have been solved.
- Only the game itself will indicate when the bomb has been successfully defused.
- If the game is still active, your task is not complete regardless of how many modules you think you've
solved.
- If the user believes all modules are solved but the game continues, continue guiding them to check for
unsolved modules (those without a bright green status LED in the top right corner) until you receive explicit
confirmation from the game that you have either defused the bomb or that it has exploded.
```

Figure H.31: Completion requirements: expert (expert role only, always included).

```
### Mission Completion

- Continue working until you receive explicit confirmation that the bomb has been completely defused.
- Never assume the bomb is defused just because you think all modules have been solved.
- You cannot determine with any certainty when the bomb is fully defused; only the game itself will
indicate when the bomb has been successfully defused.
- If the game is still active, your task is **not complete** regardless of how many modules you think
you've solved.
- If you believe all modules are solved but the game continues:
  - Methodically check the front and back face of the bomb again for incomplete modules
  - Verify each module's status LED in the top right corner is glowing bright green
  - Look for any modules you might have missed or incorrectly assessed
  - Remember that some modules may look similar to solved modules but actually require additional steps
- Continue checking for unsolved modules (those without a bright green status LED in the top right corner)
until you receive explicit confirmation of success from the game.
```

Figure H.32: Completion requirements: defuser (defuser role only, always included).

```
### Formatting Requirements

- Enclose your reasoning within <thought> and </thought> tags, followed by the JSON object for your chosen
command wrapped in <action> and </action> tags.
- Follow the exact JSON format for all commands as they are parsed by an incredibly strict validator.
- Never wrap anything in `json` tags or code fences.
- Output one command per turn only.
```

Figure H.33: Formatting requirements (all players, always included).

