# OpenReview forum: "GPTNT: Benchmarking Real-Time Collaboration Between Multimodal Agents on Keep Talking And Nobody Explodes"
_TMLR — Under review for TMLR_

### Review · Reviewer_mQNZ · 2026-06-30

**Summary Of Contributions:**

This paper introduces GPTNT, a benchmark for evaluating real-time collaboration between multimodal large language models, instantiated on the cooperative game Keep Talking and Nobody Explodes. The central contribution is the integration of four conditions that prior benchmarks have examined only in isolation: asymmetric information between agents, action under a continuous time constraint, a dynamic visual environment, and sustained multi-turn communication. Table 1 substantiates that no existing benchmark combines these properties. The principal empirical result is unambiguous: no evaluated model defuses a single bomb in the full setting, a threshold that nine of ten human pairs clear. The more substantive strength lies in the experimental decomposition supporting that result. The asynchronous vs synchronous comparison separates generation latency from capability, the single-versus-multi-module comparison separates task complexity from coordination demands and the offline probes and ablations attribute failures to specific constituent skills. The resulting diagnosis deficits in state tracking, ambiguity detection, and error recovery is persuasive because each weakness recurs across independent experiments rather than depending on any single table. The measures taken to prevent premature saturation, namely procedural generation and an extension path through the game's modding community, are a reasonable safeguard and contribute further to the work's value.

**Audience:**

Yes

**Audience Explanation:**

The benchmark addresses a timely and genuinely under-measured capability -- multi-agent collaboration under realistic constraints that bears directly on the agentic-AI research occupying a substantial portion of the TMLR readership.

**Broader Impact Concerns:**

None. The submission constitutes a diagnostic benchmark on a cooperative video game and presents no evident dual-use or ethical exposure. A Broader Impact Statement, in my assessment, is not required.

**Claims And Evidence:**

Yes

**Claims Explanation:**

The central claims are supported, and the weaknesses I identify call for attenuation of several finer-grained assertions rather than additional experimentation, which I regard as within the scope of an ordinary revision.

Strengths

1. The conclusions concerning where models fail are well substantiated, as they are corroborated across multiple independent tests. The state-tracking deficit is evident in the multi-module results, in the module-type breakdown and in the single-agent ablation. A finding that persists across this many partitions of the data warrants confidence and the principal diagnoses do.

2. The headline result is robust to the statistical concerns raised below. Because no model defuses a single bomb in the full setting, and the margin against the human baseline is large, the central claim does not depend on fine distinctions that single-run evaluation might distort.

3. The experimental design is well constructed for the purpose of attribution. The asynchronous-vs-synchronous and single-vs-multi-module comparisons isolate latency and complexity respectively, and the offline probes localize failures to specific capabilities. This lends the diagnostic claims a credibility that a single end-to-end score could not provide.

Weaknesses

1. Statistical rigor is a real concern here. Every configuration is run once at a single fixed temperature with no repeated trials and no confidence intervals. The paper actually admits this in Section 12, stating that testing reliability across repeated attempts is left for future work.
The authors do offer one justification, citing Bouthillier et al.: instead of repeating one mission many times, they sample many different missions and run each once, which gives a reasonable read on overall capability. That's a fair approach for the paper's headline result, since the gap there is large enough that single-run noise couldn't explain it.
But that justification doesn't cover Section 7.1, where two single-pass numbers are compared directly against each other. There, the "compatibility" claim rests on Sonnet-Defuser with Gemini-Expert scoring 40.9 percent versus Sonnet self-play scoring 39.1 percent, a gap of about two missions out of 110. From that small difference, the paper draws a specific conclusion: that compatibility between models, not just individual skill, drives performance. But nothing flags that this could simply be noise, even though the paper acknowledges that exact risk elsewhere.
The broad conclusions hold up fine, since their margins are too large to be chance. This narrower one doesn't get the same benefit. It needs either repeated trials or a clear pointer back to the limitation already disclosed in Section 12.

2. Regarding the human baseline, it is candidly characterized as a soft upper bound. However, the paper's signature framing -- a threshold that humans clear -- depends on an uncontrolled comparison between voice and text. Human participants benefited from the fluency of speech, while models benefited from access to the complete manual, the exact dialogue history, and faster text entry. These asymmetries operate in both directions and none is quantified, so the human gap is best read as suggestive rather than decisive.

3. The stated novelty is the asynchronous setting, yet nearly all of the interpretable analysis is conducted in synchronous mode, characterized by a frozen clock, enforced turn-taking, and an artificial three-second-per-action calibration. The most informative results therefore originate from the setting that brackets the very real-time pressure the benchmark was designed to introduce. I do not regard this as fatal, but the framing relies on the asynchronous contribution more heavily than the analysis itself does.

4. Text-only communication situates models in their strongest modality while diverging from the game's natural mode of play, producing acknowledged artifacts such as the Unicode keypad workaround; this carries somewhat greater weight than it first appears, as dialogue grounding is precisely the capability most affected by the modality choice.

**Requested Changes:**

1. (Critical) Section 12 already discloses that reliability across repeated attempts is left to future work. I am asking that this disclosure be connected explicitly to the compatibility claim in Section 7.1, which draws a qualitative conclusion from a two-mission difference in Table 6 without referencing it. Either run repeated trials for the affected pairings, or add a sentence at the point of the claim itself noting that it rests on a single pass and should be read accordingly.

2. (Critical) The asynchronous setting is presented as the benchmark's central novelty in the first contribution bullet in Section 1 and in the abstract's framing of the headline result, yet the large majority of the interpretable analysis is run in synchronous mode: Sections 5.2, 6.2, 7.1, 7.2, 8, and 10.1 all use the frozen-clock setting, while only Sections 5.1 and 6.1 report asynchronous results. I'd ask the authors to either extend more of the diagnostic analysis (Sections 7, 8, and 10.1 in particular) into the asynchronous setting, or revise the framing in the abstract and Section 1 so that asynchronicity reads as one condition studied among several, rather than as the lens through which most of the paper's findings should be read. To be clear, this is not a request to temper the headline result itself (no model solving a mission in the full setting), since that finding is asynchronous; it is the surrounding diagnostic claims in Sections 7–10 that should each be legible as to which mode produced them.

3. (Strengthening) Quantify the advantages embedded in the human baseline such as speech incrementality, typing speed, and access to the full history rather than asserting their direction, so that the human comparison may be interpreted with greater precision.

4. (Strengthening) The paper fixes temperature at 0.6 for all models, and Section 12 already justifies fixing it at one value across models, to avoid conflating sampling configuration with capability. What is missing is a justification for the value itself: why 0.6 rather than 0.2 or 1.0. A short decoding sweep, two or three temperature values run on a subset of single-module missions, would be enough to show the diagnostic conclusions are not an artifact of this particular choice.

5. (Strengthening) The Conclusion section already notes that the core missions span only difficulty levels 2–3 of 7. I would ask that this qualifier be attached explicitly to the "capability ceiling" language introduced in Section 5.2, since as currently written the term could be read as a ceiling on the game as a whole rather than on the regime actually examined.

---

> ### Author Response · Authors · 2026-07-03
>
> Thank you for your incredibly detailed and considered review. We are genuinely pleased (and relieved) that you recognised the strengths of our work, its positioning in the field, and we greatly appreciate the clear critique you provide in your response.
>
> We further discuss each in turn below and look forward to your response.
>
> **(1) Compatibility claims in §7.1 (also in response to W1)**
>
> We will modify the paragraph on compatibility to clarify that the analysis rests on a single pass, replacing it with the following:
>
> > \paragraph{Partner compatibility shapes performance.} Gemini illustrates compatibility sensitivity most evidently: its peak performance overall is achieved by Sonnet as Defuser paired with Gemini as Expert (40.9\%), a result not predicted by Gemini's overall Expert average. Yet with GPT as Expert, Gemini drops to 13.6\%, while with InternVL as Expert, it performs nearly as well as with itself, despite InternVL being among the weakest Experts. While the best other-play pairing exceeds Claude self-play by a narrow margin (two missions out of 110 under single-pass evaluation), the overall results suggest that collaborative performance is driven by compatibility rather than partner capability alone. One plausible cause is divergent communication styles shaped during post-training, though we leave this to future work.
>
>
> **(2) Claims comparing asynchronous vs synchronous (also in response to W3)**
>
> Sections 5.2 and 6.2 report synchronous results on the same missions as 5.1 and 6.1, respectively, to distinguish between failure stemming from the generation latency of models and other capabilities of models. The main reason we focus on the synchronous setting in the diagnostic sections 7, 8 and 10.1, is because they allow us to draw more nuanced conclusions on model behavior. The synchronous setting not only removes the confounding factor of the generation time, it also provides a more informative signal for interpreting model capabilities due to the wider spread in performance as success becomes more attainable for current agents. Therefore, we will update the framing around diagnostic claims in the contribution list in the introduction to more precisely reflect the setting they were produced in:
>
> **(3) The human baseline  (also in response to W2 and W4)**
>
> We understand how the term "baseline" can overstate the comparability we originally intended. Therefore, we will revise these claims as a "reference point" throughout (abstract, §1, and §10.2), to clarify that the comparison is deliberately, and inherently, not a like-for-like.
>
> Primarily, our motivation was to establish whether agents—humans or AI—can play the game in real time, and not whether they can play the game in the same way. We allow models and humans to engage with the task in the most natural way for them, with access to the conditions they are most suited to, respectively. Our aim is not to make a direct comparison, which we understand is a common precedent in many tasks. However, even enforcing the same input and output modality for both humans and AI would not guarantee experimental parity for our task. If you’ll forgive the analogy, we would be comparing the speed of a fish with the speed of a human: a human will win if they’re both on land, a fish will win if they’re in the water.
>
> For instance, even when both humans and AI play the game through text, their action spaces are fundamentally different. Where humans act on the game using keyboard and mouse, AI does so by generating JSON (and requiring the same from humans would make the task harder for them). Similarly, humans typing using a keyboard and models generating text are not the same: models emit text far faster than humans can type and can output unicode (circumventing language descriptions with quasi-image output), and enforcing that parity would give humans a decided disadvantage.
>
>
> **(4) Temperature**
>
> As discussed in the Limitation (Section 12) and Appendix E.2, we’ve set the temperature to 0.6 after conducting preliminary experiments with all models. We observed qualitatively that the determinism resulting from zero and near-zero (<0.2) temperatures led to models getting stuck in loops of repeated unproductive thoughts and actions. Therefore, we chose a moderate temperature with the objective to prevent this kind of behaviour. Interestingly, out of candidate temperatures in the moderate 0.2-0.8 range, we chose 0.6 in line with recommendations for encouraging reasoning in InternVL, the weakest model we tested. We will extend our discussion of temperature in the appendix, and will add the corresponding reference to the appropriate passage in the Limitations.

---

> > ### Author Response · Authors · 2026-07-03
> >
> > **(5) The “capability ceiling” of §5.2**
> >
> > We explain that missions correspond to difficulty levels 2-3 out of 7 in §4.3. We will reiterate this in the experimental setup (specifically in §5.1):
> >
> > > We first evaluate each model in asynchronous self-play---where time progresses in real time and without turn-taking constraints, and where both roles are filled by an instance of the same model---on ten missions with multiple puzzle modules. Evaluated missions are configured according to the game's difficulty levels 2-3 out of seven difficulty levels in the game, full details in \cref{app:ktane-difficulty}.
> >
> > And add a corresponding qualifier to the capability ceiling in §5.2:
> >
> > > We address this by running the same set of experiments in a \textit{synchronous mode}: between each Defuser turn, we freeze the game clock to give each model unlimited processing time. In this case, models take turns: the Defuser acts, the game advances, the Expert acts, and so on. From this, we establish a capability ceiling on the examined missions…

---

> > > ### Comment · Reviewer_mQNZ · 2026-07-13
> > >
> > > Thank you for the detailed and prompt response. I appreciate the care taken across all five points.
> > >
> > > Points (2), (3), (4), and (5) resolve my concerns as raised.
> > >
> > > On (1), compatibility claims in §7.1: the revised paragraph gives a clearer picture of the pairwise results, which is helpful. However, my original request was specifically to flag that the headline comparison (Sonnet-Defuser + Gemini-Expert at 40.9% vs. Sonnet self-play at 39.1%) rests on a single-pass difference of about two missions out of 110, and to tie that back to the reliability limitation already disclosed in Section 12. The new text doesn't yet include that caveat. Could you confirm whether you plan to add an explicit sentence noting this comparison is based on a single pass, alongside (or instead of) the additional pairwise detail?

---

> > > > ### Author Response · Authors · 2026-07-14
> > > >
> > > > We appreciate your ongoing engagement, and are glad that we were able to resolve most of the points you raised in your initial comments.
> > > >
> > > > We are keen to address your remaining concern and, after carefully cross-checking the wording in our paragraph against your comments, we believe that the current wording of the paragraph may not have brought our point across and gave rise to misunderstandings. Specifically, we intended to convey that we observed some deviations in the pairwise results that suggest that performance of different model pairings is not purely shaped by the capability of the respective models in their given role. We hypothesise that there may be an element of compatibility that shapes performance *in addition to the base capability*. This does not hinge on the performance of the Sonnet-Gemini pairing being better than Sonnet self-play, and we updated the title and wording of the paragraph to better reflect what we expect readers to take away from this instead. We also highlighted the sample size limitations more explicitly.
> > > >
> > > > >**Partner compatibility may shape performance beyond role capability**. Gemini illustrates compatibility sensitivity most evidently. As Defuser, Gemini scores 19.1% with InternVL as Expert but only 13.6% with GPT as Expert, despite GPT being the stronger Expert on average. Gemini is also the only Expert that does not negatively affect Sonnet’s performance as a Defuser compared with self-play (39.1%): when a Sonnet-Defuser is paired with another model as the Expert, performance drops to 20.9–32.7%, but remains comparable when paired with Gemini (40.9%). One hypothesis for this is that particular Defuser--Expert pairings coordinate better than their individual role skill would predict, with one plausible cause being divergent communication styles shaped during post-training. Given the limited sample size—discussed further in §12—we leave testing this hypothesis to future work.

---

### Review · Reviewer_ViuF · 2026-07-22

**Summary Of Contributions:**

The authors propose GPTNT, a benchmark built on the cooperative video game KTANE for evaluating real-time collaboration between two agents with distinct roles under time pressure, information asymmetry, and imperfect communication. An Expert who has access only to the instruction manual must work with a Defuser who cannot access the manual but can observe and manipulate the bomb. The two agents can communicate only through natural language. The authors study real-time collaboration in two modes: an asynchronous mode, in which the agents run through independent decision loops while the game clock continues, and a synchronous mode, in which the game is paused during decision-making and advances by an amount at every turn. The paper evaluates five LLMs through multi-module missions and simplified single-module missions, and provides a comprehensive analysis of agent behavior. It also reports a human-human baseline.

1. The design of the benchmark around KTANE is nice. I think the game is a very good fit for studying LLM-agent collaboration since information asymmetry and natural-language communication are intrinsic to the task. The real-time aspect of collaboration is also a nice and timely angle.
2. I like that the authors study collaboration through both asynchronous and synchronous modes.
3. The paper is very well written and easy to read. The benchmark, implementation, and experimental settings are described in great detail.
4. I think the single-agent ablations are well designed. Evaluating a single agent with access to the manual helps understand the effect of partner communication, and evaluating it without access to the manual is nice to learn about the model’s parametric knowledge of KTANE.

**Additional Comments:**

Minor Comments:

- Some other work on LLM-MAS collaboration might be worth mentioning. The following are several examples that came to mind: Real-Time Reasoning Gym (https://arxiv.org/pdf/2511.04898), Collab-Overcooked (https://arxiv.org/pdf/2502.20073), and CoELA (https://arxiv.org/pdf/2307.02485).
- I think studying real-time collaboration through two modes is a contribution of the paper and could be made more clear. The authors also mention asynchronous actions, which could also briefly distinguish it from conventional multi-agent action interfaces, such as the AEC and Parallel APIs in PettingZoo (https://pettingzoo.farama.org/api/aec/, https://pettingzoo.farama.org/api/parallel/).

Forward-looking:

- The game is designed for two agents. But could the setting be extended to support more agents? I would be interested to hear the authors’ thoughts on this. Larger teams would introduce more complex dynamics and could be quite interesting.

**Audience:**

Yes

**Audience Explanation:**

I think the benchmark and its findings will be interesting to people working on multimodal agents, multi-agent systems, and multi-agent communication under time constraints. Real-time collaboration between LLM agents is also a very timely topic, as more work attempts to deploy LLM agents in interactive environments, such as embodied environments, where real-time, reactive decision-making is important.

**Claims And Evidence:**

Yes

**Claims Explanation:**

The entire paper is well written. The language is clear, and the benchmark, methods, prompts, and experimental setup are documented in detail in the appendix. The paper looks highly reproducible.

**Requested Changes:**

Comments:

- The paper identifies time pressure, information asymmetry, and imperfect communication as the core dimensions of the benchmark. The authors do a great job of explaining how KTANE demands all three. However, the effects of these dimensions on agent behavior are measured less directly. Some more direct measurements could strengthen the paper. For example, do agents have a meaningful sense of time or adapt their behavior as the deadline approaches? Do they communicate more concisely, reduce their thinking time, inspect fewer parts of the bomb, or take riskier actions when little time remains? Some discussion of how GPTNT could help with understanding these behaviors, even quantitatively, would be very helpful.
- While I appreciate that the paper provides very clear and detailed language descriptions throughout Sections 3 and 4, I think some compact formalization would really help readers establish a precise understanding of the benchmark (e.g., game state space, role-specific observation and action spaces, message space, and message queues). It would also be helpful to formalize the dynamics of the two modes.
- The statement that no evaluated model solves any multi-module mission in the asynchronous setting feels a little strong under the current pass@1 setting. The authors sample ten distinct missions and attempt each mission once to get a sense of agent capabilities across different configurations at a manageable compute cost. However, LLMs can be non-deterministic, so one attempt per mission provides limited evidence to support the claim. I do not think the authors need to repeat the entire experimental suite. Repeating a representative subset would already help. For example, the authors could select one or two missions and run them several times with locally served open-source models to reduce the additional cost, and perhaps report the variation in outcomes.
- The authors use a consistent ReAct-style agent harness. However, GPTNT requires visual processing, communication, message-queue management, etc., all of which can be substantially affected by the agent architecture (e.g., memory and specialized visual-processing tools). The current results demonstrate only that GPTNT is difficult for the evaluated agent architecture; they do not necessarily mean that other agent architectures built around the same models cannot solve the missions. I think the authors could compare against other architectures, or discuss this as a limitation and adjust the scope of the claims.

---

### Review · Reviewer_Dp7o · 2026-07-22

**Summary Of Contributions:**

This paper introduces GPTNT, a benchmark for evaluating real-time collaboration between multimodal agents, built on a game called “Keep Talking and Nobody Explodes”. The task assigns two roles for each agent. The Defuser agent can observe and manipulate the bomb but has no access to the manual, and the Expert agent has access to the manual but cannot observe the bomb. The two agents must communicate under a countdown to perform visual understanding, reasoning, and coordination.

The main contributions are:
- A benchmark that combines async actions, dynamic visual environments, and multi-turn communication.
- Evaluation of five multimodal models across sync/async settings, self-play/other-play, etc.
- Evaluation of offline VQA, localization, single-agent, and no-manual tasks designed to analyze specific failure modes.

Strengths
- The paper is well-written with clear figures and examples. The task design is novel and relevant to agent research.
- The engineering implementation, the experimental setup, and prompts are documented in detail.
- The evaluation goes beyond success rates to analyze perception, reasoning, communication, etc. The paper also illustrates failures such as hallucination and poor error recovery.
- The paper explicitly investigates contamination and the use of parametric knowledge.

Weaknesses
- The claim that the proposed diagnostics separate collaboration from memorization is stronger than what the experiments establish.
- The connection between offline diagnostic performance and actual gameplay outcomes is not sufficiently direct.
- The benchmark’s long-term robustness currently depends mainly on procedural generation and future community modules.

**Additional Comments:**

This is a valuable and technically sound benchmark paper. The remaining issues mainly concern claim calibration and diagnostic validation rather than the core contribution.

**Audience:**

Yes

**Audience Explanation:**

The benchmark and findings are relevant to multimodal agents, multi-agent collaboration, human-AI interaction, real-time decision-making, and benchmark design.

**Broader Impact Concerns:**

There is no meaningful real-world weapons concern because the task uses fictional game rules.

**Claims And Evidence:**

Yes

**Claims Explanation:**

Mostly yes. The experiments convincingly show that current multimodal models struggle with real-time collaboration, including state tracking, ambiguity handling, communication, and error recovery. However:
- The no-manual and single-agent experiments only probe, rather than fully separate, collaboration and memorization.
- Offline diagnostic scores do not consistently predict gameplay performance, so stronger causal conclusions are not yet supported.

**Requested Changes:**

Important

- Clearly distinguish empirical findings from speculative mechanism explanations (e.g., partner incompatibility, disengagement, or context corruption).
- Add stronger links between offline diagnostics and gameplay, ideally using oracle or scripted partners.
- Report more validation details for the diagnostic datasets.
- Clarify benchmark versioning, hidden tests, and protection against overfitting.
- Add ethics, consent, compensation, and anonymization details for the human study.

Optional

- Test new or rule-remapped modules to reduce contamination.
- Analyze communication content, not only communication frequency.